# A Class of Short-term Recurrence Anderson Mixing Methods and Their Applications

**Fuchao Wei** [1]**, Chenglong Bao** [3,4*]**, Yang Liu** [1,2]

[1]Department of Computer Science and Technology, Tsinghua University
[2]Institute for AI Industry Research, Tsinghua University
[3]Yau Mathematical Sciences Center, Tsinghua University
[4]Yanqi Lake Beijing Institute of Mathematical Sciences and Applications
`wfc16@mails.tsinghua.edu.cn`, `{clbao,liuyang2011}@tsinghua.edu.cn`

## Abstract

Anderson mixing (AM) is a powerful acceleration method for fixed-point iterations, but its computation requires storing many historical iterations. The extra memory footprint can be prohibitive when solving high-dimensional problems in a resource-limited machine. To reduce the memory overhead, we propose a novel class of short-term recurrence AM methods (ST-AM). The ST-AM methods only store two previous iterations with cheap corrections. We prove that the basic version of ST-AM is equivalent to the full-memory AM in strongly convex quadratic optimization, and with minor changes it has local linear convergence for solving general nonlinear fixed-point problems. We further analyze the convergence properties of the regularized ST-AM for nonconvex (stochastic) optimization. Finally, we apply ST-AM to several applications including solving root-finding problems and training neural networks. Experimental results show that ST-AM is competitive with the long-memory AM and outperforms many existing optimizers.

## 1 Introduction

Anderson mixing (AM) (Anderson, 1965) is a powerful sequence acceleration method (Brezinski et al., 2018) for fixed-point iterations and has been widely used in scientific computing (Lin et al., 2019; Fu et al., 2020; An et al., 2017), e.g., the self-consistent field iterations in electronic structure computations (Garza & Scuseria, 2012; Arora et al., 2017). Specifically, we consider a fixed-point iteration $x_{k+1} = g(x_k), k = 0, 1, \ldots$, where $g : \mathbb{R}^d \mapsto \mathbb{R}^d$ is the fixed-point map. By using $m$ historical iterations, AM($m$) aims to extrapolate a new iterate that satisfies certain optimality property. When the function evaluation is costly, the reduction of the number of iterations brought by AM can save a large amount of computation (Fang & Saad, 2009).

AM can be used as a method for solving nonlinear equations (Kelley, 2018) as the fixed-point problem $x = g(x)$ is equivalent to $h(x) := x - g(x) = 0$. In practice, since computing the Jacobian of $h(x)$ is commonly difficult or even unavailable (Nocedal & Wright, 2006), AM can be seen as a practical alternate for Newton's method (An et al., 2017). Also, compared with the classical iterative methods such as the nonlinear conjugate gradient (CG) method (Hager & Zhang, 2006), no line-search or trust-region technique is used in AM, which is preferable for large-scale unconstrained optimization. Empirically, it is observable that AM can largely accelerate convergence, though its theoretical analysis is still under-explored. It turns out that in the linear case (Walker & Ni, 2011; Potra & Engler, 2013), the full-memory AM ($m = k$) is essentially equivalent to GMRES (Saad & Schultz, 1986), a powerful Krylov subspace method that can exhibit superlinear convergence behaviour in solving linear systems (Van der Vorst & Vuik, 1993). For general nonlinear problems, AM is recognized as a multisecant quasi-Newton method (Fang & Saad, 2009; Brezinski et al., 2018). As far as we know, only local linear convergence has been obtained for the limited-memory AM ($m < k$) in general (Toth & Kelley, 2015; Evans et al., 2020; De Sterck & He, 2021).

For the application of AM, one of the major concerns is the historical length $m$, a critical factor related to the efficiency of AM (Walker & Ni, 2011). A larger $m$ can incorporate more historical

---

*Corresponding author.

information into one extrapolation, but it incurs heavier memory overhead since $2m$ vectors of dimension $d$ need to be stored in AM($m$). The additional memory footprint can be prohibitive for solving high-dimensional problems in a resource-limited machine (Deng, 2019). Using small $m$ can alleviate the memory overhead but may deteriorate the efficacy of AM since much historical information is omitted in the extrapolation (Walker & Ni, 2011; Evans et al., 2020).

To address the memory issue of AM, we deeply investigate the properties of the historical iterations produced by AM and leverage them to develop the short-term recurrence variant, namely ST-AM. The basic version of ST-AM imposes some *orthogonality* property on the historical sequence, which is inspired by the CG method (Hestenes & Stiefel, 1952) that enjoys a three-term recurrence. Furthermore, to better suit the more difficult nonconvex optimization, a regularized short-term form is introduced. We highlight the main contributions of our work as follows.

1. We develop a novel class of short-term recurrence AM methods (ST-AM), including the basic ST-AM, the modified ST-AM (MST-AM), and the regularized ST-AM (RST-AM). The basic ST-AM is applicable for linear systems; MST-AM can solve general fixed-point problems; RST-AM aims for solving stochastic optimization. An important feature of ST-AM is that all methods only need to store two previous iterations with cheap corrections, which significantly reduces the memory requirement compared with the classical AM.

2. A complete theoretical analysis of the ST-AM methods is given. When solving strongly convex quadratic optimization, we prove that the basic ST-AM is equivalent to the full-memory AM and the convergence rate is similar to that of the CG method. We also prove that MST-AM has improved local linear convergence for solving fixed-point problems. Besides, we establish the global convergence property and complexity analysis for RST-AM when solving stochastic optimization problems.

3. The numerical results on solving (non)linear equations and cubic-regularized quadratic optimization are consistent with the theoretical results for the basic ST-AM and MST-AM. Furthermore, extensive experiments on training neural networks for image classification and language modeling show that RST-AM is competitive with the long-memory AM and outperforms many existing optimizers such as SGD and Adam.

## 2 RELATED WORK

AM is also known as an *extrapolation algorithm* in scientific computing (Anderson, 2019). A parallel method is Shanks transformation (Shanks, 1955) which transforms an existing sequence to a new sequence for faster convergence. Related classical algorithms include Minimal Polynomial Extrapolation (Cabay & Jackson, 1976) and Reduced Rank Extrapolation (Eddy, 1979), and a framework of these extrapolation algorithms including AM is given in (Brezinski et al., 2018). Note that an elegant recursive algorithm named $\epsilon$-algorithm had been discovered for Shanks transformation for scalar sequence (Wynn, 1956), and was later generalized as the vector $\epsilon$-algorithm (Wynn, 1962) to handle vector sequences, but this short-term recurrence form is not equivalent to the original Shanks transformation in general (Brezinski & Redivo-Zaglia, 2017). Since AM is closely related to quasi-Newton methods (Fang & Saad, 2009), there are also some works trying to derive equivalent forms of the full-memory quasi-Newton methods using limited memory (Kolda et al., 1998; Berahas et al., 2021), while no short-term recurrence is available. To the best of our knowledge, ST-AM is the first attempt to short-term recurrence quasi-Newton methods.

Recently, there have been growing demands for solving large-scale and high-dimensional fixed-point problems in scientific computing (Lin et al., 2019) and machine learning (Bottou et al., 2018). For these applications, Newton-like methods (Byrd et al., 2016; Wang et al., 2017; Mokhtari et al., 2018) are less appealing due to the heavy memory and computational cost, especially in nonconvex stochastic optimization, where only sublinear convergence can be expected if only stochastic gradients can be accessed (Nemirovski & Yudin, 1983). On the other side, first-order methods (Necoara et al., 2019) stand out for their low per-iteration cost, though the convergence can be slow in practice. When training neural networks, SGD with momentum (SGDM) (Qian, 1999), and adaptive learning rate methods, e.g. AdaGrad (Duchi et al., 2011), RMSprop (Tieleman & Hinton, 2012), Adam (Kingma & Ba, 2014), are very popular optimizers. Our methods have the nature of quasi-Newton methods while the memory footprint is largely reduced to be close to first-order methods. Thus, ST-AM can be a competitive optimizer from both theoretical and practical perspectives.

## 3 METHODOLOGY

In this section, we give the details of the proposed ST-AM. We always assume the objective function as $f : \mathbb{R}^d \to \mathbb{R}$, the fixed-point map $g : \mathbb{R}^d \mapsto \mathbb{R}^d$. Moreover, we do not distinguish $r_k = -\nabla f(x_k)$ and $r_k = g(x_k) - x_k$ in our discussion as $\nabla f(x) = 0$ is equivalent to $g(x) = x - \nabla f(x) = x$.

### 3.1 ANDERSON MIXING

The AM finds the fixed point of $g$ via maintaining two sequences of length $m$ ($m \leq k$):

$$X_k = [\Delta x_{k-m}, \Delta x_{k-m+1}, \cdots, \Delta x_{k-1}], R_k = [\Delta r_{k-m}, \Delta r_{k-m+1}, \cdots, \Delta r_{k-1}] \in \mathbb{R}^{d \times m}, \quad (1)$$

where the operator $\Delta$ denotes the forward difference, e.g. $\Delta x_k = x_{k+1} - x_k$. Each update of AM can be decoupled into two steps, namely the *projection step* and the *mixing step*:

$$\bar{x}_k = x_k - X_k \Gamma_k, \quad \text{(Projection step)}, \quad x_{k+1} = \bar{x}_k + \beta_k \bar{r}_k, \quad \text{(Mixing step)}, \quad (2)$$

where $\bar{r}_k := r_k - R_k \Gamma_k$ and $\beta_k > 0$ is the mixing parameter. The $\Gamma_k$ is determined by

$$\Gamma_k = \arg \min_{\Gamma \in \mathbb{R}^m} \|r_k - R_k \Gamma\|_2. \quad (3)$$

Thus, the full form of AM (Fang & Saad, 2009; Walker & Ni, 2011) is

$$x_{k+1} = x_k + \beta_k r_k - (X_k + \beta_k R_k) \Gamma_k. \quad (4)$$

**Remark 1.** *To see the rationality of AM, assume $g$ is continuously differentiable, then we have $h(x_j) - h(x_{j-1}) \approx h'(x_k)(x_j - x_{j-1})$ around $x_k$, where $h'(x_k)$ is the Jacobian of $h(x) := x - g(x)$. So, it is reasonable to assume $R_k \approx -h'(x_k)X_k$, and we see $\|r_k - R_k \Gamma\|_2 \approx \|r_k + h'(x_k)X_k\Gamma\|_2$. Thus, we can recognize (3) as solving $h'(x_k)d_k = h(x_k)$ in a least-squares sense, where $d_k = X_k\Gamma_k$. The mixing step incorporates $r_k$ into the new update $x_{k+1}$ if $\beta_k > 0$. Otherwise, if $\beta_k = 0$, then $x_{k+1} = \bar{x}_k$ is an interpolation of the previous iterates, leading to a stagnation.*

### 3.2 THE BASIC SHORT-TERM RECURRENCE ANDERSON MIXING

The basic ST-AM is to solve the strongly convex quadratic optimization:

$$\min_{x \in \mathbb{R}^d} f(x) := \frac{1}{2} x^{\mathrm{T}} A x - b^{\mathrm{T}} x, \quad (5)$$

where $A \succ 0$. Let $p_{-1} = q_{-1} = p_0 = q_0 = \mathbf{0} \in \mathbb{R}^d$. At the $k$-th iteration, given the two matrices $P_{k-1} = (p_{k-2}, p_{k-1}) \in \mathbb{R}^{d \times 2}, Q_{k-1} = (q_{k-2}, q_{k-1}) \in \mathbb{R}^{d \times 2}$ and defining $p = x_k - x_{k-1}$ and $q = r_k - r_{k-1}$, the basic ST-AM constructs

$$\tilde{p} = p - P_{k-1}(Q_{k-1}^{\mathrm{T}} q), \quad \tilde{q} = q - Q_{k-1}(Q_{k-1}^{\mathrm{T}} q), \quad (6a)$$

$$p_k = \tilde{p} / \|\tilde{q}\|_2, \quad q_k = \tilde{q} / \|\tilde{q}\|_2. \quad (6b)$$

Then, we update $P_k = (p_{k-1}, p_k), Q_k = (q_{k-1}, q_k) \in \mathbb{R}^{d \times 2}$. Such construction ensures $Q_k^{\mathrm{T}} Q_k = I_2$ for $k \geq 2$ and the storage of $P_k$ and $Q_k$ is equal to AM(2). With the corrected $P_k$ and $Q_k$, the ST-AM method modifies the projection step and the mixing step accordingly, that is,

$$\bar{x}_k = x_k - P_k \Gamma_k, \quad \text{(Projection step)}, \quad x_{k+1} = \bar{x}_k + \beta_k \bar{r}_k, \quad \text{(Mixing step)}, \quad (7)$$

where $\Gamma_k = \arg \min \|r_k - Q_k \Gamma\|_2 = Q_k^{\mathrm{T}} r_k$ and $\bar{r}_k = r_k - Q_k \Gamma_k$. Thus, the ST-AM replaces $X_k$ and $R_k$ in (1) by $P_k$ and $Q_k$ respectively and imposes the orthogonality condition on $Q_k$. The details of basic ST-AM are given in Algorithm 2 in Appendix C.1. Define $\bar{P}_k = (p_1, p_2, \ldots, p_k), \bar{Q}_k = (q_1, q_2, \ldots, q_k)$, the Krylov subspace $\mathcal{K}_m(A, v) \equiv \mathrm{span}\{v, Av, A^2 v, \ldots, A^{m-1} v\}$, the range of $X$ as $\mathrm{range}(X)$. We give the properties of the basic ST-AM in Theorem 1.

**Theorem 1.** *Let $\{x_k\}$ be the sequence generated by the basic ST-AM. The following relations hold:*
*(i) $\|\tilde{q}\|_2 > 0, \mathrm{range}(\bar{P}_k) = \mathrm{range}(X_k) = \mathcal{K}_k(A, r_0), \mathrm{range}(\bar{Q}_k) = \mathrm{range}(R_k) = A\mathcal{K}_k(A, r_0)$;*
*(ii) $\bar{Q}_k = -A\bar{P}_k, \bar{Q}_k^{\mathrm{T}} \bar{Q}_k = I_k$;*
*(iii) $\bar{r}_k \perp \mathrm{range}(\bar{Q}_k)$ and $\bar{x}_k = x_0 + z_k$, where $z_k = \arg \min_{z \in \mathcal{K}_k(A, r_0)} \|r_0 - Az\|_2$.*
*If $\|\bar{r}_k\|_2 = 0$, then $x_{k+1}$ is the exact solution.*

The proof is in Appendix C.1. Note that the property (iii) in Theorem 1 exactly describes the relation $\bar{x}_k = x_k^G$, where $x_k^G$ is the output of the $k$-th iteration of GMRES (Saad & Schultz, 1986). Moreover, let $\bar{x}_k^{AM}$ be the $k$-th intermediate iterate in the full-memory AM. It holds that $\bar{x}_k^{AM} = x_k^G$ (See Proposition 1 in Appendix C.1.), which induces that $\bar{x}_k = \bar{x}_k^{AM} = x_k^G$. This equivalence indicates that ST-AM is more efficient than AM and GMRES since only two historical iterations need to be stored. Moreover, by directly applying the convergence analysis of GMRES (Corollary 6.33 in (Saad, 2003)), we obtain the convergence rate of the basic ST-AM for solving (5):

**Corollary 1.** *Suppose the eigenvalues of $A$ lie in $[\mu, L]$ with $\mu > 0$, and let $\{x_k\}$ be the sequence generated by the basic ST-AM, then the $k$-th intermediate residual $\bar{r}_k$ satisfies $\|\bar{r}_k\|_2 \leq 2\left(\frac{\sqrt{L/\mu}-1}{\sqrt{L/\mu}+1}\right)^k \|r_0\|_2$. Moreover, the algorithm finds the exact solution in at most $(d+1)$ iterations.*

**Remark 2.** *The GMRES can be simplified to an elegant three-term recurrence algorithm called the conjugate residual (CR) method (Algorithm 6.20 in (Saad, 2003)) when solving (5). Thus, a similar simplification for AM is expected to exist. Like CG and Chebyshev acceleration (Algorithm 12.1 in (Saad, 2003)), the convergence rate of ST-AM has the optimal dependence on the condition number, while ST-AM does not form the Hessian-vector products explicitly.*

### 3.3 THE MODIFIED SHORT-TERM RECURRENCE ANDERSON MIXING

For general nonlinear fixed-point problems, global convergence may be unavailable for the basic ST-AM, as a counter-example exists for AM (Mai & Johansson, 2020). Thus, we propose a modified version of the basic ST-AM (MST-AM) and prove the local linear convergence rate under similar conditions used in (Toth & Kelley, 2015; Evans et al., 2020). Concretely, the MST-AM makes three main changes to the basic ST-AM.

**Change 1**: Instead of applying the normalization (6b), the MST-AM constructs $p_k$ and $q_k$ via

$$\zeta_k = (Q_{k-1}^T Q_{k-1})^\dagger Q_{k-1}^T q, \quad p_k = p - P_{k-1}\zeta_k, \quad q_k = q - Q_{k-1}\zeta_k, \tag{8}$$

where "$\dagger$" is the Moore-Penrose inverse. Accordingly, we choose $\Gamma_k = \arg\min \|r_k - Q_k\Gamma\|_2 = (Q_k^T Q_k)^\dagger Q_k^T r_k$. This change relaxes the orthonormality for $Q_k$ ($k \geq 2$), but keeps the orthogonality condition: $Q_{k-1}^T q_k = 0$. In fact, $\bar{Q}_{k-1}^T q_k = 0$ in the case of solving (5).

**Change 2**: MST-AM imposes the boundedness constraints on $P_{k-1}\zeta_k$ and $Q_{k-1}\zeta_k$: If $\|P_{k-1}\zeta_k\|_2 > c_p\|p\|_2$ or $\|Q_{k-1}\zeta_k\|_2 > c_q\|q\|_2$, then $P_k = P_{k-1}, Q_k = Q_{k-1}$, where $c_p > 0, c_q \in (0,1)$ are predefined constants. It is worth mentioning that adding some boundedness condition is common in the analysis of AM (Toth & Kelley, 2015; Evans et al., 2020).

**Change 3**: MST-AM restarts, i.e. setting $P_k = Q_k = \mathbf{0} \in \mathbb{R}^{d \times 2}$ every $m$ iterations. This restart operation is to limit the number of higher-order terms appeared in the residual expansion in our analysis and we can set $m$ to be a large number in practice.

The detailed description of MST-AM is given in Appendix C.2. In the next theorem, we establish the convergence rate analysis for the MST-AM.

**Theorem 2.** *Let $\{x_k\}$ be the sequence generated by MST-AM, $x^* \in \mathbb{R}^d$ be a fixed point of $g$ and $m$ be the restarting period for MST-AM. Suppose that in the ball $\mathcal{B}(\rho) := \{x \in \mathbb{R}^d | \|x - x^*\|_2 < \rho\}$ for some $\rho > 0$, $g$ is Lipschitz continuously differentiable and there are constants $\kappa \in (0,1)$ and $\hat{\kappa} > 0$ with (i) $\|g(y) - g(x)\|_2 \leq \kappa\|y - x\|_2$ for every $x, y \in \mathcal{B}(\rho)$, and (ii) $\|g'(y) - g'(x)\|_2 \leq \hat{\kappa}\|y - x\|_2$ for every $x, y \in \mathcal{B}(\rho)$, where $g'$ is the Jacobian of $g$. Assume $|1 - \beta_k| + \kappa\beta_k \leq \kappa_0$ for a constant $\kappa_0 \in (0,1)$. If $x_0$ is sufficiently close to $x^*$, then for $r_k := g(x_k) - x_k$, the following bound holds:*

$$\|r_{k+1}\|_2 \leq \theta_k(|1 - \beta_k| + \kappa\beta_k)\|r_k\|_2 + \hat{\kappa}\sum_{j=0}^{m_k} \mathcal{O}\left(\|r_{k-j}\|_2^2\right), \tag{9}$$

*where $\theta_k = \|\bar{r}_k\|_2/\|r_k\|_2 \leq 1$ and $m_k = k \mod m$. Thus, the residuals $\{r_k\}$ converge Q-linearly, and the errors $\{\|x_k - x^*\|_2\}$ converge R-linearly.*

**Remark 3.** *In a local region around $x^*$, the convergence rate is determined by the first-order term $\theta_k(|1 - \beta_k| + \kappa\beta_k)\|r_k\|_2$. We can choose $\beta_k = 1$ such that $|1 - \beta_k| + \kappa\beta_k = \kappa < 1$. Since $\bar{r}_k$ is the orthogonal projection of $r_k$ onto the subspace $\text{range}(Q_k)^\perp$, $\theta_k$ has the interpretation of*

---

**Algorithm 1** RST-AM for stochastic programming

---

**Input**: $x_0 \in \mathbb{R}^d, \beta_k > 0, \alpha_k \in [0, 1], \delta_k^{(1)} > 0, \delta_k^{(2)} > 0$.
**Output**: $x \in \mathbb{R}^d$
  1:  $P_0, Q_0 = \mathbf{0} \in \mathbb{R}^{d \times 2}, p_0, q_0 = \mathbf{0} \in \mathbb{R}^d$
  2: **for** $k = 0, 1, \ldots$, until convergence, **do**
  3:    $r_k = -\nabla f_{S_k}(x_k)$
  4:    **if** $k > 0$ **then**
  5:       $p = x_k - x_{k-1}, q = r_k - r_{k-1}$
  6:       $\zeta_k = (Q_{k-1}^{\mathrm{T}} Q_{k-1} + \delta_k^{(1)} P_{k-1}^{\mathrm{T}} P_{k-1})^\dagger Q_{k-1}^{\mathrm{T}} q$
  7:       $q_k = q - Q_{k-1}\zeta_k, p_k = p - P_{k-1}\zeta_k$
  8:       $P_k = [p_{k-1}, p_k], Q_k = [q_{k-1}, q_k]$
  9:    **end if**
10:    Check Condition (13) and use smaller $\alpha_k$ if (13) is violated
11:    $\Gamma_k = (Q_k^{\mathrm{T}} Q_k + \delta_k^{(2)} P_k^{\mathrm{T}} P_k)^\dagger Q_k^{\mathrm{T}} r_k$
12:    $\bar{x}_k = x_k - \alpha_k P_k \Gamma_k, \bar{r}_k = r_k - \alpha_k Q_k \Gamma_k$
13:    $x_{k+1} = \bar{x}_k + \beta_k \bar{r}_k$
14:    Apply learning rate schedule of $\alpha_k, \beta_k$
15: **end for**
16: **return** $x_k$

---

*the direction-sine between $r_k$ and the subspace* $\mathrm{range}(Q_k)$. *When $\theta_k$ is small, e.g., $r_k$ nearly lies in* $\mathrm{range}(Q_k)$, *the acceleration by MST-AM is significant. Compared to AM(m), MST-AM incorporates historical information with orthogonalization. In the SPD linear case and without restart, the global orthogonality property holds, i.e. $\bar{r}_k \perp range(\bar{Q}_k)$, which means there is no loss of historical information, while AM(m) (Evans et al., 2020) does not have such property in this ideal case.*

### 3.4 THE REGULARIZED SHORT-TERM RECURRENCE ANDERSON MIXING

Inspired by the recent work on stochastic Anderson mixing (SAM) method (Wei et al., 2021), we develop a regularized ST-AM (RST-AM) for solving nonconvex stochastic optimization problems.

Consider the nonconvex optimization problem $\min_{x \in \mathbb{R}^d} f(x) := \frac{1}{T} \sum_{i=1}^{T} f_{\xi_i}(x)$, where $f_{\xi_i} : \mathbb{R}^d \to \mathbb{R}$ is the loss function corresponding to $i$-th data sample and $T$ is the number of data samples. In mini-batch training, the gradient is evaluated for $f_{S_k}(x_k) := \frac{1}{n_k} \sum_{i \in S_k} f_{\xi_i}(x_k)$, where $S_k \subseteq [T] := \{1, 2, \ldots, T\}$ is the sampled mini-batch, and $n_k := |S_k|$ is the batch size. In this case, we set $r_k = -\nabla f_{S_k}(x_k)$ (Line 3 in Algorithm 1), which is an unbiased estimate of the negative gradient.

Recalling from (8), $\zeta_k = (Q_{k-1}^{\mathrm{T}} Q_{k-1})^\dagger Q_{k-1}^{\mathrm{T}} \Delta r_{k-1} = \arg\min \|\Delta r_{k-1} - Q_{k-1}\zeta\|_2$ as $q = \Delta r_{k-1}$ by definition. Since ST-AM is based on a local quadratic approximation (5) in a small region around $x_k$, a large magnitude of $\|P_{k-1}\zeta_k\|_2$ tends to make the change from $\Delta x_{k-1}$ to $p_k = \Delta x_{k-1} - P_{k-1}\zeta_k$ too aggressively, which may lead to instability. Consequently, we add a penalty term in the above least squares problem, i.e.

$$\zeta_k = \arg\min \|\Delta r_{k-1} - Q_{k-1}\zeta\|_2^2 + \delta_k^{(1)}\|P_{k-1}\zeta\|_2^2, \tag{10}$$

where $\delta_k^{(1)} > 0$. The same as SAM (Wei et al., 2021), we also add a regularization term for computing $\Gamma_k$ via

$$\Gamma_k = \arg\min \|r_k - Q_k\Gamma\|_2^2 + \delta_k^{(2)}\|P_k\Gamma\|_2^2, \tag{11}$$

where $\delta_k^{(2)} > 0$, and a damping term $\alpha_k$ is used as shown in Line 12 in Algorithm 1. In practice, we choose the two regularization parameters as

$$\delta_k^{(1)} = \frac{c_1\|r_k\|_2^2}{\|\Delta x_{k-1}\|_2^2 + \epsilon_0}, \quad \delta_k^{(2)} = \max\left\{\frac{c_2\|r_k\|_2^2}{\|p_k\|_2^2 + \epsilon_0}, C\beta_k^{-2}\right\}, \tag{12}$$

where $c_1, c_2, C > 0$ are constants, and $\epsilon_0 > 0$ is a small constant to bound the denominators away from zero. Assuming $\|p_{k-1}\|_2 \approx \|p_k\|_2 = \mathcal{O}(\|\Delta x_{k-1}\|_2)$, the choices of (12) make $\|\delta_k^{(1)} P_{k-1}^{\mathrm{T}} P_{k-1}\|_2 \approx \mathcal{O}(\|r_k\|_2^2)$ and $\|\delta_k^{(2)} P_k^{\mathrm{T}} P_k\|_2 \approx \mathcal{O}(\|r_k\|_2^2)$ aware of the change of the local curvature: large (small) $\|r_k\|_2$ tends to lead to a large (small) regularization.

**Remark 4.** *One update of $x_k$ given by Line 11-13 in Algorithm 1 can be formulated as $x_{k+1} = x_k + H_k r_k$, where $H_k = \beta_k I - \alpha_k Y_k Z_k^{\dagger} Q_k^{\mathrm{T}}$, $Y_k = P_k + \beta_k Q_k$, $Z_k = Q_k^{\mathrm{T}} Q_k + \delta_k^{(2)} P_k^{\mathrm{T}} P_k$. To guarantee the positive definiteness of $H_k$, we follow the same procedure in SAM. Let $\lambda_k$ be the largest eigenvalue of $Y_k Z_k^{\dagger} Q_k^{\mathrm{T}} + Q_k Z_k^{\dagger} Y_k^{\mathrm{T}}$. If $\alpha_k$ satisfies*

$$\alpha_k \lambda_k \leq 2\beta_k(1 - \mu), \tag{13}$$

*then $s_k^{\mathrm{T}} H_k s_k \geq \beta_k \mu \|s_k\|_2^2, \forall s_k \in \mathbb{R}^d$, where $\mu \in (0,1)$ is a constant. Note that $\lambda_k$ can be cheaply obtained by computing the largest eigenvalue of a matrix of $\mathbb{R}^{4 \times 4}$ (see Appendix C.3.1).*

We summarize the RST-AM in Algorithm 1 and establish its convergence properties here. First, we impose the same assumptions on the objective function $f$ as those in (Wei et al., 2021).

**Assumption 1.** *$f : \mathbb{R}^d \to \mathbb{R}$ is continuously differentiable. $f(x) \geq f^{low} > -\infty$ for any $x \in \mathbb{R}^d$. $\nabla f$ is globally $L$-Lipschitz continuous; namely $\|\nabla f(x) - \nabla f(y)\|_2 \leq L\|x - y\|_2$ for any $x, y \in \mathbb{R}^d$.*

**Assumption 2.** *For any iteration $k$, the stochastic gradient $\nabla f_{\xi_k}(x_k)$ satisfies $\mathbb{E}_{\xi_k}[\nabla f_{\xi_k}(x_k)] = \nabla f(x_k)$, $\mathbb{E}_{\xi_k}[\|\nabla f_{\xi_k}(x_k) - \nabla f(x_k)\|_2^2] \leq \sigma^2$, where $\sigma > 0$, and $\xi_k, k = 0, 1, \ldots$ are independent samples that are independent of $\{x_j\}_{j=0}^k$.*

The diminishing condition about $\beta_k$ is

$$\sum_{k=0}^{+\infty} \beta_k = +\infty, \qquad \sum_{k=0}^{+\infty} \beta_k^2 < +\infty. \tag{14}$$

We give the convergence properties of RST-AM in nonconvex (stochastic) optimization and proofs are deferred to Appendix C.3.2.

**Theorem 3.** *Suppose Assumption 1 hold and $\{x_k\}$ is the sequence generated by full-batch RST-AM, i.e. $n_k = T$. Let $\beta_k = \beta \in (0, \frac{\mu}{2L(1+C^{-1})}]$ be a constant, $\alpha_k \in [0, 1]$ and satisfies (13), then*

$$\frac{1}{N} \sum_{k=0}^{N-1} \|\nabla f(x_k)\|_2^2 \leq \frac{2(f(x_0) - f^{low})}{N\mu\beta}, \tag{15}$$

*in the $N$ iterations. To ensure $\frac{1}{N}\sum_{k=0}^{N-1}\|\nabla f(x_k)\|_2^2 < \epsilon$, the number of iterations is $\mathcal{O}(1/\epsilon)$.*

**Theorem 4.** *Suppose Assumptions 1 and 2 hold and $\{x_k\}$ is the sequence generated by RST-AM with batch size $n_k = n \leq T$. If $\beta_k \in (0, \frac{\mu}{4L(1+C^{-1})}]$ and satisfies (14), $\alpha_k \in [0, \min\{1, \beta_k^{\frac{1}{2}}\}]$ and satisfies (13), then*

$$\liminf_{k \to \infty} \|\nabla f(x_k)\|_2 = 0 \text{ with probability } 1 \quad and \quad \exists M_f > 0 \to \mathbb{E}[f(x_k)] \leq M_f, \forall k. \tag{16}$$

*If $\mathbb{E}_{\xi_k}[\|\nabla f_{\xi_k}(x_k)\|_2^2] \leq M_g, \forall k$, where $M_g > 0$ is a constant, we have*

$$\lim_{k \to \infty} \|\nabla f(x_k)\|_2 = 0 \text{ with probability } 1. \tag{17}$$

**Theorem 5.** *Suppose Assumptions 1 and 2 hold and $\{x_k\}_{k=0}^{N-1}$ is the first $N$ iterations generated by RST-AM with fixed batch size $n_k = n$. Let $\beta_k = \min\{\frac{\mu}{4L(1+C^{-1})}, \frac{\tilde{D}}{\sigma\sqrt{N}}\}$, where $\tilde{D}$ is a problem-independent constant; $\alpha_k \in [0, \min\{1, \beta_k^{\frac{1}{2}}\}]$ and satisfies (13). Let $R$ be a random variable following $P_R(k) := Prob\{R = k\} = 1/N$, then*

$$\mathbb{E}[\|\nabla f(x_R)\|_2^2] \leq \frac{16 D_f L(1 + C^{-1})}{N\mu^2} + \frac{\sigma}{\mu\sqrt{N}}\left(\frac{4D_f}{\tilde{D}} + \frac{4(L + \mu^{-1})(1 + C^{-1})\tilde{D}}{n}\right), \tag{18}$$

*where $D_f := f(x_0) - f^{low}$ and the expectation is taken with respect to $R$ and $\{S_j\}_{j=0}^{N-1}$. To ensure $\mathbb{E}[\|\nabla f(x_R)\|_2^2] \leq \epsilon$, the number of iterations is $\mathcal{O}(1/\epsilon^2)$.*

**Remark 5.** *The proofs of Theorem 4 and 5 are based on the analysis of SAM (Wei et al., 2021). The theorems show that the convergence of RST-AM is no worse than SGD (Robbins & Monro, 1951). There are two key differences between RST-AM and SAM: RST-AM is based on short-term recurrences while SAM usually maintains longer historical sequences to ensure effectiveness; RST-AM uses additional correction and regularization terms (Line 5-8 in Algorithm 1) to incorporate historical information while SAM simply discards the oldest iteration to make space for $\Delta x_{k-1}$ and $\Delta r_{k-1}$. The reduced memory requirement in RST-AM makes it applicable for solving more challenging problems in machine learning.*

## 4 EXPERIMENTS

We validated the effectiveness of our proposed ST-AM methods in various applications in fixed-point iterations and nonconvex optimization, including linear and nonlinear problems, deterministic and stochastic optimization. Specifically, we first tested ST-AM in linear problems, cubic-regularized quadratic minimization (Carmon & Duchi, 2020) and a multiscale deep equilibrium (MDEQ) model (Bai et al., 2020). Then we applied RST-AM to train neural networks and compared them with several first-order and second-order optimizers. Experimental details are in Appendix D.

### 4.1 EXPERIMENTS ABOUT THE BASIC ST-AM AND MST-AM

We verified the properties of ST-AM declared in Theorem 1 and 2 by solving four problems (details are in Appendix D.1): (I) strongly convex quadratic optimization (corresponding to Theorem 1); (II) solving a nonsymmetric linear system $Ax = b$ (corresponding to $\hat{\kappa} = 0$ in Theorem 2); (III) cubic-regularized quadratic minimization $\min_{x \in \mathbb{R}^d} f(x) := \|Ax - b\|_2^2 + \frac{M}{3}\|x\|_2^3$ (corresponding to $\hat{\kappa} > 0$ in Theorem 2); (IV) root-finding problems in MDEQ on CIFAR-10 (Krizhevsky et al., 2009).

The compared methods were gradient descent (GD), fixed-point iteration (FP), conjugate residual method (CR) (Saad, 2003), BFGS (Nocedal & Wright, 2006), Broyden's method (Broyden, 1965), and the full-memory AM (AM). We used the basic ST-AM to solve Problem I and II, and MST-AM ($c_p = c_q = 1$) to solve Problem III and IV.

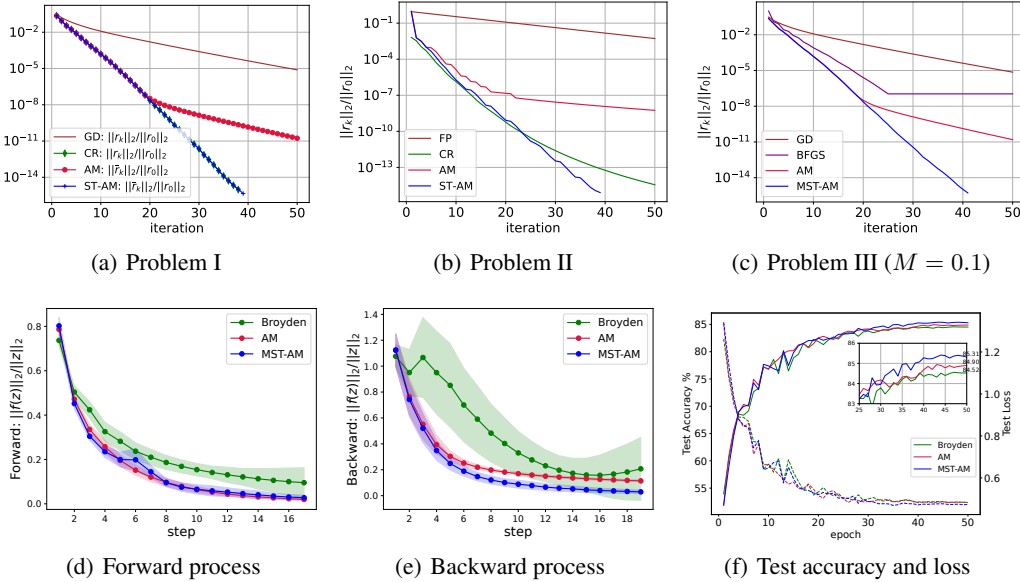

(a) Problem I       (b) Problem II       (c) Problem III ($M = 0.1$)

(d) Forward process       (e) Backward process       (f) Test accuracy and loss

Figure 1: (a) $\|r_k\|_2/\|r_0\|_2$ of GD and CR, and $\|\bar{r}_k\|_2/\|r_0\|_2$ of AM and ST-AM for solving Problem I; (b) $\|r_k\|_2/\|r_0\|_2$ for solving Problem II; (c) $\|r_k\|_2/\|r_0\|_2$ for solving Problem III ($M = 0.1$); (d)(e) relative residuals of the forward and backward root-finding processes in MDEQ, and shaded areas correspond to the standard deviations; (f) test accuracy and loss in MDEQ/CIFAR-10.

The numerical results shown in Figure 1 demonstrate the power of ST-AM as a variant of Krylov subspace methods. It significantly accelerates the slow convergence of the GD or FP method, and can outperform AM. Figure 1(a) clearly verifies the correctness of Theorem 1: within the machine precision, the intermediate residual $\bar{r}_k$ of ST-AM coincides with the residual $r_k$ of CR. Note that AM fails to coincide with CR and ST-AM due to the intrinsic numerical weakness to solve (3), as also pointed out in (Walker & Ni, 2011). Figure 1(b) shows that ST-AM can outperform CR, though both methods enjoy short-term recurrences and are equivalent for solving SPD linear systems. Figure 1(c) also shows MST-AM surpasses BFGS in solving cubic-regularized problems. The tests in MDEQ/CIFAR-10 indicate that MST-AM is comparable to the full-memory methods in the forward root-finding process and converges faster in the backward process. The accuracy is also comparable.

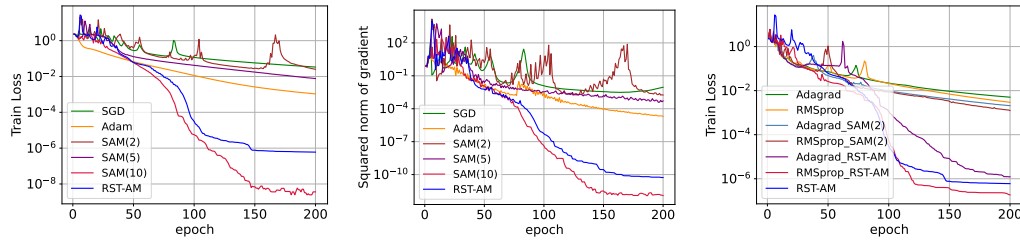

(a) Train loss (w/o preconditioning)     (b) SNG (w/o preconditioning)     (c) Train loss (w/ preconditioning)

Figure 2: Experiments on MNIST. (a)(b) Training loss and the squared norm of gradient (SNG) (w/o preconditioning for SAM, RST-AM); (c) Training loss (w/ preconditioning for SAM, RST-AM).

## 4.2 EXPERIMENTS ABOUT RST-AM

We applied RST-AM to train neural networks, with full-batch training on MNIST (LeCun et al., 1998), and mini-batch training on CIFAR-10/CIFAR-100 and Penn Treebank (Marcus et al., 1993).

**Experiments on MNIST**. We trained a convolutional neural network (CNN) on MNIST to see the convergence behaviour of RST-AM in nonconvex optimization (cf. Theorem 3), for which we were only concerned about the training loss. Figure 2(a)(b) show that the short-memory SAMs ($m = 2, 5$) hardly show any improvement over the first-order optimizers SGD and Adam, while RST-AM can close the gap of the long-memory ($m = 10$) and the short-memory methods. We also considered the effect of preconditioning on RST-AM (see Appendix A.3). The notation "A_B" means B method preconditioned by A method. Figure 2(c) indicates that preconditioning also works much better for RST-AM than SAM(2), and RMSprop_RST-AM can outperform the non-preconditioned RST-AM.

Table 1: Experiments on CIFAR10/CIFAR100. "-" means failing to complete the test in our device due to memory limit. "*" indicates numbers published in (Wei et al., 2021).

(a) Final TOP1 test accuracy (mean ± standard deviation) (%) for training 160 epochs.

| Method | Test accuracy on CIFAR10 | | | | | | Test accuracy on CIFAR100 | | |
|---|---|---|---|---|---|---|---|---|---|
| | ResNet18 | ResNet20 | ResNet32 | ResNet44 | ResNet56 | WRN16-4 | ResNet18 | ResNeXt | DenseNet |
| SGDM* | 94.82±.15 | 92.03±.16 | 92.86±.15 | 93.10±.23 | 93.47±.28 | 94.90±.09 | 77.27±.09 | 78.41±.54 | 78.49±.12 |
| Adam* | 93.03±.07 | 91.17±.13 | 92.03±.28 | 92.28±.62 | 92.39±.23 | 92.45±.11 | 72.41±.17 | 73.57±.17 | 70.80±.23 |
| AdaBound | 94.25±.31 | 90.77±.08 | 91.73±.06 | 92.00±.18 | 92.44±.04 | 93.50±.12 | 75.07±.14 | 75.74±.20 | 76.06±.13 |
| AdaBelief* | 94.65±.13 | 91.15±.21 | 92.15±.17 | 92.79±.24 | 93.30±.07 | 94.46±.13 | 76.25±.06 | 78.27±.16 | 78.83±.15 |
| Lookahead* | 94.92±.33 | 92.07±.04 | 92.86±.15 | 93.26±.24 | 93.36±.13 | 94.90±.15 | 77.63±.35 | 78.93±.12 | 79.37±.16 |
| AdaHessian* | 94.36±.09 | 91.92±.32 | 92.18±.18 | 92.74±.11 | 92.40±.06 | 94.04±.12 | 76.59±.42 | - | - |
| SAM(2) | 95.07±.04 | 92.14±.33 | 93.04±.23 | 93.46±.09 | 93.66±.06 | 95.07±.16 | 77.51±.24 | 79.02±.21 | 80.00±.23 |
| SAM(10)* | 95.17±.10 | **92.43±.19** | 93.22±.32 | **93.57±.14** | **93.77±.12** | **95.23±.07** | **78.13±.14** | 79.31±.27 | 80.09±.52 |
| RST-AM | **95.27±.04** | 92.39±.11 | **93.24±.36** | 93.52±.02 | 93.69±.18 | 95.21±.09 | 77.91±.22 | **79.53±.34** | **80.36±.25** |

(b) The memory and computation cost compared with SGDM. The notations "m","t/e" and "t" are abbreviations of memory, per-epoch time and total running time, respectively.

| Cost (× SGDM) | CIFAR10/ResNet18 | | | CIFAR10/WRN16-4 | | | CIFAR100/ResNeXt50 | | | CIFAR100/DenseNet121 | | |
|---|---|---|---|---|---|---|---|---|---|---|---|---|
| | m | t/e | t | m | t/e | t | m | t/e | t | m | t/e | t |
| SGDM* | 1.00 | 1.00 | 1.00 | 1.00 | 1.00 | 1.00 | 1.00 | 1.00 | 1.00 | 1.00 | 1.00 | 1.00 |
| SAM(10)* | 1.73 | 1.78 | 1.00 | 1.26 | 1.28 | 0.80 | 1.30 | 1.16 | 0.58 | 1.16 | 1.19 | 0.60 |
| RST-AM | 1.05 | 1.46 | 0.82 | 1.03 | 1.14 | 0.71 | 1.04 | 1.07 | 0.54 | 1.01 | 1.11 | 0.55 |

**Experiments on CIFAR**. We trained ResNet18/20/32/44/56 (He et al., 2016), WideResNet16-4 (Zagoruyko & Komodakis, 2016) (abbr. WRN16-4) on CIFAR-10, and ResNet18, ResNeXt50 (Xie et al., 2017), DenseNet121 (Huang et al., 2017) on CIFAR-100. The baseline optimizers were SGDM, Adam, AdaBound (Luo et al., 2018), AdaBelief (Zhuang et al., 2020), Lookahead (Zhang et al., 2019), AdaHessian (Yao et al., 2021) and SAM. Here, some results of the baselines in (Wei et al., 2021) were used for reference since the experimental settings were the same. Table 1(a) shows RST-AM improves SAM(2) and has comparable test accuracy to SAM(10). RST-AM also outperforms other baseline optimizers. Table 1(b) reports the memory and computation cost, where we

used SGDM as the baseline and other optimizers were terminated when achieving a comparable or better test accuracy than SGDM. It indicates that RST-AM introduces $\leq 5\%$ extra memory overhead compared with SGDM, and significantly reduces the memory footprint of AM. Since RST-AM needs fewer training epochs, the total running time is less than SGDM.

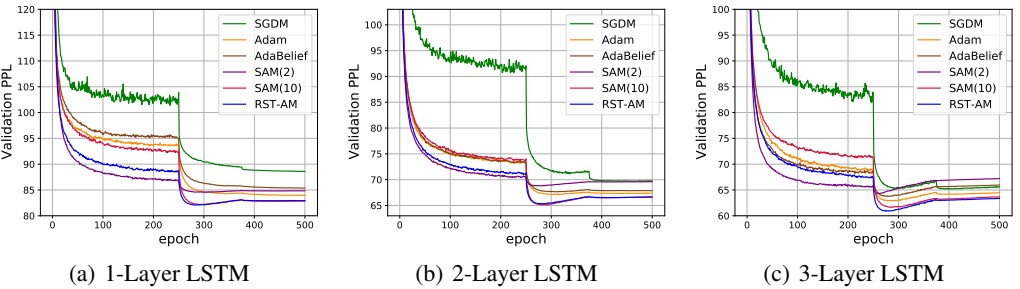

(a) 1-Layer LSTM  (b) 2-Layer LSTM  (c) 3-Layer LSTM

Figure 3: Validation perplexity of training 1,2,3-layer LSTM on Penn Treebank.

**Experiments on Penn Treebank**. We trained LSTMs with 1-3 layer(s) on Penn Treebank and report the validation perplexity in Figure 3 and test perplexity in Table 2 (lower is better). The results suggest that RST-AM is comparable to or even better than SAM(10). The improvement of RST-AM over other optimizers is also significant. We report the computation and memory cost in Appendix D.2.4. RST-AM can still surpass Adam while using much fewer epochs, thus reducing the total running time.

Table 2: Test perplexity of training 1,2,3-layer LSTM on Penn Treebank. Lower is better.

| Method | 1-Layer | 2-Layer | 3-Layer |
|---|---|---|---|
| SGDM | 83.48±.03 | 65.89±.18 | 61.88±.23 |
| Adam | 80.33±.15 | 64.32±.06 | 59.72±.13 |
| AdaBelief | 81.29±.35 | 64.68±.10 | 60.46±.07 |
| SAM(2) | 80.79±.19 | 65.52±.29 | 61.13±.12 |
| SAM(10) | 78.78±.14 | 62.46±.11 | 58.93±.09 |
| RST-AM | **78.41±.18** | **62.46±.08** | **58.31±.23** |

Table 3: Test accuracy (%) for adversarial training.

| Optimizer | CIFAR10/ResNet18 | | | | CIFAR100/DenseNet121 | | | |
|---|---|---|---|---|---|---|---|---|
| | Clean | FGSM | PGD-20 | C&W$_\infty$ | Clean | FGSM | PGD-20 | C&W$_\infty$ |
| SGD | 82.16 | 63.23 | 51.91 | 50.22 | 59.45 | 39.76 | 30.92 | 29.00 |
| RST-AM | **82.53** | **63.78** | **52.43** | **50.52** | **60.48** | **40.41** | **31.20** | **29.52** |

Table 4: FID score for SN-GAN.

| Method | Adam | AdaBelief | RST-AM |
|---|---|---|---|
| Best FID | 13.07±.18 | 12.80±.09 | **12.05±.15** |
| Final FID | 13.34±.14 | 13.59±.21 | **12.50±.29** |

**Adversarial training**. We applied RST-AM to adversarial training (Madry et al., 2018) as the outer-optimizer and compared it with SGD by the clean test accuracy and robust test accuracy. The results on CIFAR10/ResNet18 and CIFAR100/DenseNet121 are reported in Table 3. It can be seen that RST-AM can achieve both higher clean test accuracy and higher robust test accuracy. More results can be found in Appendix D.2.5.

**Generative adversarial network (GAN)**. We tested RST-AM by training a GAN equipped with spectral normalization (SN-GAN) (Miyato et al., 2018), where the generator and discriminator networks were ResNets and the dataset was CIFAR-10. Table 4 shows that RST-AM can achieve lower FID score (better accuracy) than Adam and AdaBelief.

## 5 CONCLUSION

In this paper, to address the memory issue of Anderson mixing (AM), we develop a novel class of short-term recurrence AM methods (ST-AM) and test it in various applications, including solving linear and nonlinear problems, deterministic and stochastic optimization. We give a complete theoretical analysis of the proposed methods. We prove that the basic ST-AM is equivalent to the full-memory AM in strongly convex quadratic optimization. With some minor changes, it has local linear convergence for solving general fixed-point problems under some common assumptions. We also introduce the regularized form of ST-AM and analyze its convergence properties. The numerical results show that the ST-AM methods are comparable to or even better than the long-memory AM while consuming less memory. The regularized ST-AM also outperforms many existing optimizers in training neural networks in various tasks.

ACKNOWLEDGMENTS

This work was supported by the National Key R&D Program of China (No. 2021YFA1001300), National Natural Science Foundation of China (No.61925601), Tsinghua University Initiative Scientific Research Program, National Natural Science Foundation of China (No.11901338), and Huawei Noah's Ark Lab. We thank all anonymous reviewers for their valuable comments and suggestions on this work.

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

## A  ADDITIONAL PRELIMINARIES

We provide some additional preliminaries in this section for readers that are not familiar with Anderson mixing, fixed-point iterations and some techniques mentioned in the main paper.

### A.1  FIXED-POINT ITERATION

The fixed-point problem and the optimization problem are the main application scenarios of our methods. It is worth pointing out that there are some minor differences between these two problems that make algorithm designs different. The key difference is that for a fixed-point problem, the Jacobian (if exists) is generally not symmetric while for optimization, the Hessian is naturally symmetric. In principle, a fixed-point solver can also be applicable for an optimization problem since the first-order necessary condition of $\min_{x \in \mathbb{R}^d} f(x)$, where $f : \mathbb{R}^d \to \mathbb{R}$, is $\nabla f(x) = 0$.

Consider a contraction mapping $g : \mathbb{R}^d \mapsto \mathbb{R}^d$, i.e. for some $\kappa < 1$, $\|g(x) - g(y)\|_2 \leq \kappa\|x - y\|_2, \forall x, y \in \mathbb{R}^d$. According to the contraction mapping theorem, a unique fixed point $x^*$ exists for $g$ and the iterates generated by the iteration $x_{k+1} = g(x_k)$ converge to $x^*$, starting from any $x_0 \in \mathbb{R}^d$. In practice, the damped fixed-point iteration is also commonly used: given the $\beta_k \in (0, 2)$, the update is $x_{k+1} = (1 - \beta_k)x_k + \beta_k g(x_k) = x_k + \beta_k r_k$, where $r_k := g(x_k) - x_k$ is called the *residual*.

The fixed-point iteration, also known as Picard iteration in some areas, can converge very slowly in practice. Anderson mixing is a method to improve the convergence.

### A.2  ANOTHER FORM OF ANDERSON MIXING

The derivation of Anderson mixing (AM) in Section 3.1 explicitly interprets AM as a two-step procedure. In the literature, there is another equivalent form of AM.

Let the projection coefficients $\Gamma_k = (\gamma_k^{(1)}, \ldots, \gamma_k^{(m)})^{\mathrm{T}} \in \mathbb{R}^m$. Define the auxiliary coefficients $\{\theta_k^{(j)}\}_{j=0}^m$ as $\theta_k^{(0)} = \gamma_k^{(1)}, \theta_k^{(j)} = \Delta\gamma_k^{(j)} (j = 1, \ldots, m - 1), \theta_k^{(m)} = 1 - \gamma_k^{(m)}$, then $\sum_{j=0}^m \theta_k^{(j)} = 1$ and $\bar{r}_k = \sum_{j=0}^m \theta_k^{(j)} r_{k-m+j}$. Hence the least squares problem (3) can be reformulated as a constrained problem $\min_{\{\theta_k^{(j)}\}_{j=0}^m} \|\sum_{j=0}^m \theta_k^{(j)} r_{k-m+j}\|_2$ s.t. $\sum_{j=0}^m \theta_k^{(j)} = 1$, which indicates that as a linear combination of the historical residuals $\{r_j\}_{j=k-m}^m$, $\bar{r}_k$ is minimal in terms of the $L_2$-norm. Also, the projection step and the mixing step (2) can be reformulated as $\bar{x}_k = \sum_{j=0}^m \theta_k^{(j)} x_{k-m+j}$ and $x_{k+1} = (1-\beta_k)\sum_{j=0}^m \theta_k^{(j)} x_{k-m+j} + \beta_k \sum_{j=0}^m \theta_k^{(j)} g(x_{k-m+j})$, respectively. Such formulation is also adopted in the literature (Toth & Kelley, 2015; Mai & Johansson, 2020; Scieur et al., 2020).

Let $H_k$ be the solution to the constrained optimization problem (Fang & Saad, 2009)

$$\min_{H_k} \|H_k - \beta_k I\|_F \text{ subject to } H_k R_k = -X_k, \tag{19}$$

then the update (4) is $x_{k+1} = x_k + H_k r_k$. It suggests that AM is a multisecant quasi-Newton method (Fang & Saad, 2009).

### A.3  PRECONDITIONED ANDERSON MIXING

Like preconditioning for Krylov subspace methods (Saad, 2003), preconditioning can also be incorporated into Anderson mixing to mitigate the ill-conditioning of the original problem (Wei et al., 2021). The idea is to replace the mixing step in (2) via a preconditioned mixing.

Suppose that there is a basic solver $preconditioner(x_k, s_k)$, which works as a black-box procedure that updates $x_k$ given the residual $s_k$, i.e. $x_{k+1} = preconditioner(x_k, s_k)$, then the preconditioned mixing of RST-AM is

$$x_{k+1} = preconditioner(\bar{x}_k, \bar{r}_k).$$

which substitutes for the Line 13 in Algorithm 1. The simple mixing (Line 13 in Algorithm 1 can be seen as a special case by defining $preconditioner(x_k, s_k) = x_k + \beta_k s_k$, i.e. preconditioned by a damped fixed-point iteration. Moreover, if we write the preconditioning operation as

$preconditioner(x_k, s_k) := x_k + G_k s_k$, where $G_k$ is the matrix to approximate the inverse Jacobian, then the preconditioned AM is

$$x_{k+1} = x_k + G_k r_k - (X_k + G_k R_k)(R_k^{\mathrm{T}} R_k)^{\dagger} R_k^{\mathrm{T}} r_k \qquad (20)$$

(cf. the definitions of $X_k, R_k$ in Section 3.1). The matrix $H_k = G_k - (X_k + G_k R_k)(R_k^{\mathrm{T}} R_k)^{\dagger} R_k^{\mathrm{T}}$ forms a low-rank updated approximation to the inverse Jacobian: it solves

$$\min_{H_k} \|H_k - G_k\|_F \text{ subject to } H_k R_k = -X_k,$$

which is a direct extension of (19).

For the stochastic Anderson mixing, using damped projection can be helpful, i.e. $\bar{x}_k = (1-\alpha_k)x_k + \alpha_k(x_k - X_k\Gamma_k) = x_k - \alpha_k X_k\Gamma_k$ and $\bar{r}_k = r_k - \alpha_k R_k\Gamma_k$ correspondingly. Then the preconditioned AM with damped projection is

$$x_{k+1} = x_k + G_k r_k - \alpha_k(X_k + G_k R_k)\Gamma_k. \qquad (21)$$

## B ADDITIONAL DISCUSSION

### B.1 THE MEMORY AND COMPUTATIONAL EFFICIENCY

Since the ST-AM methods only need to store two previous iterations, the memory and computational cost can be reduced to be close to first-order methods.

**Memory cost.** Assume that the iteration number is $k$ and the model parameter size is $d$. The full-memory AM stores all previous iterations, thus the additional memory is $2kd$. To reduce the memory overhead, the limited-memory AM($m$) only maintains the most recent $m$ iterations while discarding the older historical information (cf. (1)). Hence the additional memory of AM($m$) is $2md$. Choosing a suitable $m$ can be problem-dependent (Walker & Ni, 2011), and it is often suggested that $m \geq 5$ (Mai & Johansson, 2020; Fu et al., 2020) to avoid too much historical information being discarded. There is no equivalence between AM($m$) and the full-memory AM. On the other side, our ST-AM methods have the same memory footprint as that of AM(2), i.e. only introducing $4d$ additional memory. For a large model that contains millions or even billions of parameters, such reduction in memory requirement can be significant.

**Computational cost.** For ST-AM, besides the gradient evaluations, the main additional computational cost comes from vector corrections, the projection and mixing step. These operations are very cheap: matrix multiplications of $\mathbb{R}^{2\times d} \times \mathbb{R}^{d\times 2}$, $\mathbb{R}^{2\times d} \times \mathbb{R}^{d\times 1}$, and $\mathbb{R}^{d\times 2} \times \mathbb{R}^{2\times 1}$; the pseudo-inverse can be exactly solved as the size of the matrix is $\mathbb{R}^{2\times 2}$. Also, the $Q_{k-1}^{\mathrm{T}} Q_{k-1}$ and $P_{k-1}^{\mathrm{T}} P_{k-1}$ in Line 6 in Algorithm 1 can reuse the results in the previous iteration (cf. Line 11). So the total additional computational cost of RST-AM is $\mathcal{O}(d)$.

### B.2 APPLICABILITY

In the main paper, we develop a class of short-term recurrence Anderson mixing. Among them, the basic ST-AM can serve as a new linear solver for linear systems; MST-AM can be used as a new fixed-point solver for nonlinear equations; RST-AM is a new method for optimization. These methods are based on Anderson mixing and have close relationship between each other. The theoretical analysis and numerical results show the great potential of ST-AM methods for various applications. Here, we give some comparisons between the ST-AM methods and some other classical methods.

**ST-AM versus current solvers for linear systems.** ST-AM is an iterative solver. It is equivalent to the full-memory AM and GMRES in strongly convex quadratic optimization (or SPD linear systems), and can also have linear convergence rate for general nonsymmetric linear systems. Since it is based on short-term recurrences, like the CG method, the memory and per-iteration cost of ST-AM is economical. On the other side, the LU factorization based methods (Golub & Van Loan, 2013) are direct solvers that can incur overwhelming memory and computation cost for large sparse linear systems. Although sparse direct solvers (Duff et al., 2017) can alleviate overhead, they are often more complicated to implement and difficult for parallel computing. Moreover, iterative solvers can benefit from the preconditioning technique that improves convergence (Saad, 2003). Also, ST-AM

has advantages over other iterative methods since it does not need to directly access the matrix and only the residual is required. In the case that explicitly accessing the matrix is difficult, this property is appealing. Hence, unlike nonlinear CG that relies on line search, it is direct to extend ST-AM to unconstrained optimization where the gradient is commonly available while the Hessian can be too costly to obtain. Moreover, ST-AM is very flexible and any iterative solver can be viewed as a black-box iterative process to be accelerated by ST-AM.

**MST-AM versus other quasi-Newton methods for nonlinear equations.** MST-AM has the nature of quasi-Newton methods since it is built upon AM that is recognized as a multisecant quasi-Newton method (Fang & Saad, 2009). In scientific computing, AM has been successfully applied to solve many difficult nonlinear problems arising from many areas (An et al., 2017; Lin et al., 2019; Fu et al., 2020; Yang, 2021). Since AM only manipulates residuals and does not use line-search or trust-region technique, it is efficient to apply AM to accelerate a slowly convergent black-box iterative process. A comprehensive discussion about the applicability of AM for nonlinear problems and the relation between AM and Broyden's methods can be found in (Fang & Saad, 2009).

One of the biggest issues of AM and other quasi-Newton methods is the additional memory overhead, because they need to store historical iterations to form the secant equations. To make a compromise, limited-memory quasi-Newton methods such as L-BFGS (Liu & Nocedal, 1989) are proposed in which only limited number of historical iterations are stored. In each update, the oldest iteration is discarded to make space for the up-to-date secant pair. As a result, the limited-memory quasi-Newton methods can lose the local superlinear convergence properties achieved by the full-memory schemes (Berahas et al., 2021). Also, as stated by Berahas et al. (2021), the choice of the number of historical iterations (i.e. historical length) is problem-dependent, and one does not know *a priori* the best choice when solving a particular problem.

Unlike the limited-memory quasi-Newton methods whose performance can be sensitive to the historical length, our MST-AM method only needs to store two corrected historical iterations. MST-AM carefully incorporates historical information through *orthogonalization*. In the ideal case, i.e. strongly convex quadratic optimization (or SPD linear systems), it is equivalent to the full-memory AM which means there is no loss of historical information. Our experiments verify the theoretical properties of MST-AM and indicate MST-AM can be a competitive method for nonlinear equations.

**RST-AM versus first-order methods for stochastic optimization.** For stochastic optimization such as training deep neural networks in machine learning, the first-order methods have come to dominate the field due to the low memory and per-iteration cost. The RST-AM is an extension of ST-AM and MST-AM to tackle this challenging problem. RST-AM has theoretical guarantees in deterministic/stochastic optimization and also inherits several advantages of AM and ST-AM:

- Fast convergence in quadratic optimization. It is important to possess such property for an optimizer since a smooth function can be approximated by a quadratic function in the local region around the optima and many techniques such as trust region (Nocedal & Wright, 2006) rely on this local approximation. Adaptive learning rate methods such as AdaGrad (Duchi et al., 2011), Adam (Kingma & Ba, 2014) use a diagonal approximation of the *Fisher information* matrix that is an approximation of the Hessian. However, in the simple quadratic case, these methods can only roughly match the performance of the Jacobi method for solving linear systems (Saad, 2003), which is better than gradient descent but far inferior to the powerful Krylov subspace methods. The momentum method mimics CG method by incorporating a historical iteration into the search direction. However, the choice of momentum and stepsize can be an art (Sutskever et al., 2013). For RST-AM, there is no need to determine the stepsize and the fast convergence rate of ST-AM can be recovered by simply setting $\alpha_k = 1$ and $\delta_k^{(1)} = \delta_k^{(2)} = 0$. For more difficult functions, the damping and regularization in RST-AM can be enabled to improve stability.

- Theoretical guarantee in stochastic optimization. If only first-order information can be accessed, the SGD (Ghadimi & Lan, 2013) achieves an optimal convergence rate $\mathcal{O}(1/\epsilon^2)$ to obtain an $\epsilon$-accurate solution (Nemirovski & Yudin, 1983). In such case, it seems to be a big mismatch for current second-order methods, because there is no theoretical improvement albeit with more memory and computation resource. Such mismatch may also account for the popularity of first-order methods. Since RST-AM has very limited additional memory

overhead, and also achieves the $\mathcal{O}(1/\epsilon^2)$ complexity, it can be applied to many applications that are dominated by first-order methods.

- Flexibility in use. The application of RST-AM can be very flexible. In principle, RST-AM can be applied to improve any slowly convergent black-box iterative process by viewing the latter as a fixed-point iteration. For example, consider accelerating a solver of a commercial software, where we have no access to the underlying codes to provide our custom implementation, or some case rewriting the codes is too cumbersome. Moreover, RST-AM can efficiently incorporate the *preconditioning* technique. Hence any optimizer, even an optimizer built upon neural networks (Andrychowicz et al., 2016), can be used as a preconditioner for RST-AM. Preconditioning largely enhances the applicability of RST-AM for various applications. For example, RST-AM can be preconditioned by Adam for the language task and SGDM for image classification. So any fine-tuned first-order optimizer can be combined with RST-AM to achieve an overall improvement. As RST-AM is light-weight, this additional cost is marginal and can be largely counteracted by the actual improvement. So in some sense, the purpose of RST-AM is not to totally replace current off-the-shelf optimizers but to achieve collaborative effectiveness: RST-AM is aware of the second-order information while the first-order method can mitigate the ill-conditioning of the problem.

Overall, the ST-AM methods have wide applicability and can be competent methods from both theoretical and practical perspectives.

## C    PROOFS

We give more details about ST-AM and the proofs of the theorems in the main paper.

### C.1    THE BASIC ST-AM FOR STRONGLY CONVEX QUADRATIC OPTIMIZATION

Recall that the strongly convex quadratic optimization is formulated as

$$\min_{x\in\mathbb{R}^d} f(x) := \frac{1}{2}x^{\mathrm{T}}Ax - b^{\mathrm{T}}x, \tag{22}$$

where $A \in \mathbb{R}^{d\times d}$ is SPD, $b \in \mathbb{R}^d$. Solving (22) is equivalent to solving the SPD linear system

$$Ax = b. \tag{23}$$

The detail of the basic ST-AM is given in Algorithm 2.

We first state the relationship of AM with GMRES in the following proposition. Similar results can also be found in (Walker & Ni, 2011; Wei et al., 2021).

Let $x_k^{\mathrm{G}}, r_k^{\mathrm{G}} := b - Ax_k^{\mathrm{G}}$ denote the $k$-th GMRES iterate and residual, respectively, and $\mathcal{K}_k(A,v) :=$ span$\{v, Av, \ldots, A^{k-1}v\}$ denotes the $k$-th Krylov subspace generated by $A$ and $v$. Define $e^j :=$ $(1, 1, \ldots, 1)^{\mathrm{T}} \in \mathbb{R}^j$ for $j \geq 1$. Let range$(X)$ denote the linear space spanned by the columns of $X$. The main results of the full-memory AM are stated in Proposition 1 and Proposition 2. The Proposition 1 is the same as the Proposition 2 in (Wei et al., 2021), and we restate it here for completeness. The Proposition 2 is new as far as we know.

**Proposition 1** (General linear system)**.** *For solving a general linear system $Ax = b$ with the full-memory AM ($m = k$), suppose that $\beta_k > 0$ and the fixed-point map is $g(x) = (I - A)x + b$. If the initial point of AM is $x_0 = x_0^{\mathrm{G}}$ and* rank$(R_k) = m$, *then the intermediate iterate $\bar{x}_k$ satisfies $\bar{x}_k = x_k^{\mathrm{G}}$.*

*Proof.* The definition of the fixed-point map suggests that the residual $r_k = g(x_k) - x_k = b - Ax_k$.

Since $R_k = -AX_k$ and $A$ is nonsingular, we have rank$(X_k) = m$. We first show

$$\text{range}(X_k) = \mathcal{K}_k(A, r_0^{\mathrm{G}}) \tag{24}$$

by induction. We abbreviate $\mathcal{K}_k(A, r_0^{\mathrm{G}})$ as $\mathcal{K}_k$ in this proof.

---

**Algorithm 2** ST-AM for strongly convex quadratic optimization

---

**Input**: $x_0 \in \mathbb{R}^d, \beta_k > 0, 0 < max\_iter \leq d$.
**Output**: $x \in \mathbb{R}^d$
 1: $P_0, Q_0 = \mathbf{0} \in \mathbb{R}^{d \times 2}, p_0, q_0 = \mathbf{0} \in \mathbb{R}^d$
 2: **for** $k = 0, 1, \ldots, max\_iter$ **do**
 3:     $r_k = -\nabla f(x_k)$
 4:     **if** $k > 0$ **then**
 5:        $p = x_k - x_{k-1}, q = r_k - r_{k-1}$         (Compute $\Delta x_{k-1}, \Delta r_{k-1}$)
 6:        $\tilde{q} = q - Q_{k-1}(Q_{k-1}^{\mathrm{T}}q), \tilde{p} = p - P_{k-1}(Q_{k-1}^{\mathrm{T}}q)$    ($\tilde{q} \perp Q_{k-1}$)
 7:        $p_k = \tilde{p}/\|\tilde{q}\|_2, q_k = \tilde{q}/\|\tilde{q}\|_2$              ($\|q_k\|_2 = 1$)
 8:        $P_k = [p_{k-1}, p_k], Q_k = [q_{k-1}, q_k]$           ($Q_k^{\mathrm{T}}Q_k = I_2$)
 9:     **end if**
10:     $\Gamma_k = Q_k^{\mathrm{T}}r_k$
11:     $\bar{x}_k = x_k - P_k\Gamma_k, \bar{r}_k = r_k - Q_k\Gamma_k$         (Projection step: $\bar{r}_k \perp Q_k$)
12:     $x_{k+1} = \bar{x}_k + \beta_k\bar{r}_k$                   (Mixing step)
13:     **if** $\|\bar{r}_k\|_2 = 0$ **then**
14:        break
15:     **end if**
16: **end for**
17: **return** $x_k$

---

First, $\Delta x_0 = \beta_0 r_0 = \beta_0 r_0^{\mathrm{G}}$ since $x_1 = x_0 + \beta_0 r_0$. If $k = 1$, then the proof is complete. Then, suppose that $k > 1$ and, as an inductive hypothesis, that $\text{range}(X_{k-1}) = \mathcal{K}_{k-1}$. With (4) we have

$$
\begin{aligned}
\Delta x_{k-1} &= x_k - x_{k-1} \\
&= \beta_{k-1}r_{k-1} - (X_{k-1} + \beta_{k-1}R_{k-1})\Gamma_{k-1} \\
&= \beta_{k-1}(b - Ax_{k-1}) - (X_{k-1} - \beta_{k-1}AX_{k-1})\Gamma_{k-1} \\
&= \beta_{k-1}b - \beta_{k-1}A(x_0 + \Delta x_0 + \cdots + \Delta x_{k-2}) - (X_{k-1} - \beta_{k-1}AX_{k-1})\Gamma_{k-1} \\
&= \beta_{k-1}r_0 - \beta_{k-1}AX_{k-1}e^{k-1} - (X_{k-1} - \beta_{k-1}AX_{k-1})\Gamma_{k-1}.
\end{aligned}
\tag{25}
$$

Since $r_0 \in \mathcal{K}_{k-1}$, and by the inductive hypothesis $\text{range}(X_{k-1}) \subseteq \mathcal{K}_{k-1}$ which also implies $\text{range}(AX_{k-1}) \subseteq \mathcal{K}_k$, we know $\Delta x_{k-1} \in \mathcal{K}_k$, which implies $\text{range}(X_k) \subseteq \mathcal{K}_k$. Since we assume $\text{rank}(X_k) = m = k$ which implies $\dim(\text{range}(X_k)) = \dim(\mathcal{K}_k)$, we have $\text{range}(X_k) = \mathcal{K}_k$, thus completing the induction. As a result, we also have

$$
\text{range}(R_k) = \text{range}(AX_k) = A\mathcal{K}_k(A, r_0^{\mathrm{G}}).
\tag{26}
$$

Recalling that to determine $\Gamma_k$, we solve the least squares problem (3) and $R_k = -AX_k$. We have

$$
\Gamma_k = \arg\min_{\Gamma \in \mathbb{R}^m} \|r_k + AX_k\Gamma\|_2.
\tag{27}
$$

Since $\text{rank}(AX_k) = \text{rank}(X_k) = m$, (27) has a unique solution. Also, since $r_k = b - Ax_k = b - A(x_0 + X_ke^k) = r_0 - AX_ke^k$, we have $r_k + AX_k\Gamma = r_0 - AX_ke^k + AX_k\Gamma = r_0 - AX_k\tilde{\Gamma}$, where $\tilde{\Gamma} = e^k - \Gamma$. So $\Gamma_k$ solves (27) if and only if $\tilde{\Gamma}_k = e^k - \Gamma_k$ solves

$$
\min_{\tilde{\Gamma} \in \mathbb{R}^m} \|r_0 - AX_k\tilde{\Gamma}\|_2,
\tag{28}
$$

According to (24), (28) is equal to $\min_{z \in \mathcal{K}_m(A, r_0^{\mathrm{G}})} \|r_0 - Az\|_2$ which is the GMRES minimization problem. Since the solution of (28) is also unique, we have

$$
\bar{x}_k = x_k - X_k\Gamma_k = x_k - X_k(e^k - \tilde{\Gamma}_k) = x_0 + X_k\tilde{\Gamma}_k = x_k^{\mathrm{G}}.
$$

$\square$

In Proposition 1, the assumption that $R_k$ has full column rank is critical to ensure no stagnation occurs in AM for solving a general linear system. In fact, for SPD linear systems (23) or strongly convex quadratic optimization (22), when AM breaks down, i.e. $R_k$ is rank deficient, AM obtains the exact solution, as shown in the next proposition.

**Proposition 2** (SPD). *For applying the full-memory AM to minimize a strongly convex quadratic problem (22), or equivalently, solve a SPD linear system (23), suppose that $\beta_k > 0$ and the fixed-point map is $g(x) = (I - A)x + b$. If $\mathrm{rank}(R_k) = k$ holds for $1 \leq k < s$ while failing to hold for $k = s$, where $s \geq 1$, then the residual of AM satisfies $r_s = \bar{r}_{s-1} = 0$.*

*Proof.* The definition of $g$ suggests that the residual $r_k = g(x_k) - x_k = b - Ax_k$. The relation $R_k = -AX_k$ holds during the iterations and the nonsingularity of $A$ implies $\mathrm{rank}(X_k) = \mathrm{rank}(R_k)$.

For $s = 1$, since the first step of AM is $x_1 = x_0 + \beta_0 r_0$, the assumption $\mathrm{rank}(R_1) = 0$ implies that $\mathrm{rank}(r_0) = \mathrm{rank}(X_1) = 0$, i.e. $r_1 = \bar{r}_0 := 0$.

For $s > 1$, because $\Delta x_{s-1} = x_s - x_{s-1} = -X_{s-1}\Gamma_{s-1} + \beta_{s-1}\bar{r}_{s-1}$, the rank deficiency of $X_s$ implies $\Delta x_{s-1} \in \mathrm{range}(X_{s-1})$, which further implies $\bar{r}_{s-1} \in \mathrm{range}(X_{s-1})$. So there exists $\zeta \in \mathbb{R}^{s-1}$, such that $\bar{r}_{s-1} = X_{s-1}\zeta$. Note that according to (3), $\bar{r}_{s-1} \perp R_{s-1} = -AX_{s-1}$, so we have

$$0 = \bar{r}_{s-1}^{\mathrm{T}} AX_{s-1} = (X_{s-1}\zeta)^{\mathrm{T}} AX_{s-1} = \zeta^{\mathrm{T}} X_{s-1}^{\mathrm{T}} AX_{s-1}. \tag{29}$$

Because $\mathrm{rank}(X_{s-1}) = s - 1$ and $A$ is SPD, we know $X_{s-1}^{\mathrm{T}} AX_{s-1}$ is also SPD. So $\zeta = 0$, which implies $\bar{r}_{s-1} = 0$. Hence $x_s = \bar{x}_{s-1}$ and $r_s = \bar{r}_{s-1} = 0$. □

Now we give the proof of Theorem 1.

***Proof of Theorem 1***. Besides relations (i)-(iii), we add an auxiliary relation here:
(iv) $r_k = r_0 + \bar{Q}_k \bar{\Gamma}_k \in \mathcal{K}_{k+1}(A, r_0)$, where $\bar{\Gamma}_k \in \mathbb{R}^k$.
We prove the relations (i)-(iv) by induction.

For $k = 1$, since $\bar{r}_0 \neq 0$, according to Proposition 2, $\mathrm{rank}(\Delta x_0) = \mathrm{rank}(X_1) = 1, \mathrm{rank}(\Delta r_0) = \mathrm{rank}(R_1) = 1$, so $\tilde{q} \neq 0$, which implies Line 7 in Algorithm 2 is well-defined. The relation (i) holds. Since $\tilde{q} = q = \Delta x_0, \tilde{p} = p = \Delta r_0$, and $\Delta r_0 = -A\Delta x_0$, the equality $\bar{Q}_1 = -A\bar{P}_1$ also holds. Due to the normalization in Line 7, $\bar{Q}_1^{\mathrm{T}} \bar{Q}_1 = 1$. Since $r_1 = r_0 - \beta_0 Ar_0$ and $\mathrm{range}(\bar{Q}_1) = \mathrm{range}(Ar_0)$, it is clear that $r_1 = r_0 - \bar{Q}_1 \bar{\Gamma}_1 \in \mathcal{K}_2(A, r_0)$, namely relation (iv). Due to the projection step Line 11, $\bar{r}_1 \perp \mathrm{range}(Q_1) = \mathrm{range}(\bar{Q}_1)$. Also, $\bar{r}_1 = r_1 - Q_1\Gamma_1 = r_0 - \beta_0 Ar_0 - Q_1\Gamma_1 = r_0 - \bar{Q}_1\eta_1$, where the last equality is due to $\mathrm{span}\{Ar_0\} = \mathrm{range}(Q_1) = \mathrm{range}(\bar{Q}_1)$. For $r_1^{\mathrm{G}} = r_0 - Az_1$, where $z_1 = \arg\min_{z \in \mathcal{K}_1(A, r_0)} \|r_0 - Az\|_2$, it holds $r_1^{\mathrm{G}} \perp A\mathcal{K}_1(A, r_0) = \mathrm{range}(\bar{Q}_1)$. As a result, both $\bar{r}_1$ and $r_1^{\mathrm{G}}$ are the orthogonal projections of $r_0$ onto the subspace $\mathrm{range}(\bar{Q}_1)^{\perp}$, which implies $\bar{r}_1 = r_1^{\mathrm{G}}$. So $\bar{x}_1 = x_1^{\mathrm{G}} = x_0 + z_1$ because their residuals are equal and $A$ is nonsingular. Hence relation (iii) holds.

Suppose that $k > 1$, and as an inductive hypothesis, the relations (i)-(iv) hold for $j = 1, \ldots, k - 1$. Consider the $k$-th iteration. From Line 6 in Algorithm 2, $\tilde{q} \in \mathrm{range}(\Delta r_{k-1}, Q_{k-1})$, and $\tilde{p} \in \mathrm{range}(\Delta x_{k-1}, P_{k-1})$. We first prove that $\tilde{q} \neq 0$ by contradiction.

If $\tilde{q} = 0$, then from Line 6 in Algorithm 2, $\Delta r_{k-1} \in \mathrm{range}(Q_{k-1}) \subseteq \mathrm{range}(\bar{Q}_{k-1})$, which implies $\Delta x_{k-1} \in \mathrm{range}(P_{k-1}) \subseteq \mathrm{range}(\bar{P}_{k-1})$ as $\Delta r_{k-1} = -A\Delta x_{k-1}$ and $\bar{Q}_{k-1} = -A\bar{P}_{k-1}$ and $A$ is nonsingular. From Line 11 and Line 12, we have

$$\Delta x_{k-1} = x_k - x_{k-1} = -P_{k-1}\Gamma_{k-1} + \beta_{k-1}\bar{r}_{k-1}. \tag{30}$$

So $\bar{r}_{k-1} \in \mathrm{range}(P_{k-1}) \subseteq \mathrm{range}(\bar{P}_{k-1})$ since $\Delta x_{k-1} \in \mathrm{range}(P_{k-1})$. Hence there exists $\zeta \in \mathbb{R}^{k-1}$, such that $\bar{r}_{k-1} = \bar{P}_{k-1}\zeta$. From the inductive hypothesis, we know $\bar{r}_{k-1} \perp \bar{Q}_{k-1} = -A\bar{P}_{k-1}$, so we have

$$0 = \bar{r}_{k-1}^{\mathrm{T}} A\bar{P}_{k-1} = (\bar{P}_{k-1}\zeta)^{\mathrm{T}} A\bar{P}_{k-1} = \zeta^{\mathrm{T}} \bar{P}_{k-1}^{\mathrm{T}} A\bar{P}_{k-1}.$$

Since $\bar{Q}_{k-1}^{\mathrm{T}} \bar{Q}_{k-1} = I_{k-1}$, we know $\mathrm{rank}(\bar{Q}_{k-1}) = k - 1$, which implies $\mathrm{rank}(\bar{P}_{k-1}) = k - 1$ due to $\bar{Q}_{k-1} = -A\bar{P}_{k-1}$. Hence $\bar{P}_{k-1}^{\mathrm{T}} A\bar{P}_{k-1}$ is also SPD. Then $\zeta = 0$ which implies $\bar{r}_{k-1} = 0$. It is impossible otherwise Algorithm 2 has terminated in the $(k-1)$-th iteration. So $\tilde{q} \neq 0$ and Line 7 is well-defined.

Since $\bar{r}_{k-1} = r_{k-1} - Q_{k-1}\Gamma_{k-1}$, and $r_{k-1} \in \mathcal{K}_k(A, r_0), \mathrm{range}(Q_{k-1}) \subseteq \mathrm{range}(\bar{Q}_{k-1}) = A\mathcal{K}_{k-1}(A, r_0)$ as the inductive hypothesis, we have $\bar{r}_{k-1} \in \mathcal{K}_k(A, r_0)$, which together with (30)

and $\text{range}(P_{k-1}) \subseteq \text{range}(\bar{P}_{k-1}) = \mathcal{K}_{k-1}(A, r_0)$ infers $\Delta x_{k-1} \in \mathcal{K}_k(A, r_0)$. Hence $\Delta r_{k-1} = -A\Delta x_{k-1} \in A\mathcal{K}_k(A, r_0)$. As a result, $q_k = \tilde{q}/\|\tilde{q}\|_2 \in \text{range}(\Delta r_{k-1}, Q_{k-1}) \subseteq A\mathcal{K}_k(A, r_0)$. So $\text{range}(\bar{Q}_k) = \text{range}(\bar{Q}_{k-1}, q_k) \subseteq A\mathcal{K}_k(A, r_0)$. Moreover, $q_k \notin \text{range}(\bar{Q}_{k-1})$, otherwise $\tilde{q} \in \text{range}(\bar{Q}_{k-1})$ that implies $\Delta r_{k-1} = q \in \text{range}(\bar{Q}_{k-1})$, which is impossible following the former proof of $\tilde{q} \neq 0$. So we have $\text{range}(\bar{Q}_k) = A\mathcal{K}_k(A, r_0)$.

Because $\Delta r_{k-1} = -A\Delta x_{k-1}$ and $Q_{k-1} = -AP_{k-1}$ due to $\bar{Q}_{k-1} = -A\bar{P}_{k-1}$, Line 6 in Algorithm 2 infers $\tilde{q} = -A\tilde{p}$, which implies $q_k = -Ap_k$. So $\bar{Q}_k = -A\bar{P}_k$. Since $A$ is nonsingular and $\text{range}(\bar{Q}_k) = A\mathcal{K}_k(A, r_0)$, we have $\text{range}(\bar{P}_k) = \mathcal{K}_k(A, r_0)$.

As $\text{range}(X_k) = \mathcal{K}_k(A, r_0)$ and $\text{range}(R_k) = A\mathcal{K}_k(A, r_0)$ has been proved in Proposition 1, the relation (i) holds for the $k$-th iteration.

To prove $\bar{Q}_k^{\mathrm{T}} \bar{Q}_k = I_k$, it suffices to show $q_k \perp \bar{Q}_{k-1}$, as the equalities $\bar{Q}_{k-1}^{\mathrm{T}} \bar{Q}_{k-1} = I_{k-1}$ and $\|q_k\|_2 = 1$ has already held. It is equivalent to prove $\tilde{q} \perp \bar{Q}_{k-1}$. From the construction of $\tilde{q}$ in Line 6 in Algorithm 2, we know $Q_{k-1}^{\mathrm{T}} \tilde{q} = 0$, so $\tilde{q} \perp \text{span}(q_{k-2}, q_{k-1})$ (for $k = 2$, $\tilde{q} \perp q_0 = \mathbf{0}$ clearly holds). To further prove $\tilde{q} \perp \text{range}(\bar{Q}_{k-3})(k \geq 4)$, note that

$$\Delta r_{k-1} = -A\Delta x_{k-1} = AP_{k-1}\Gamma_{k-1} - \beta_{k-1}A\bar{r}_{k-1} = -Q_{k-1}\Gamma_{k-1} - \beta_{k-1}A\bar{r}_{k-1},$$

where the second equality is a direct substitution with (30). Therefore,

$$\bar{Q}_{k-3}^{\mathrm{T}}\Delta r_{k-1} = -\bar{Q}_{k-3}^{\mathrm{T}}Q_{k-1}\Gamma_{k-1} - \beta_{k-1}\bar{Q}_{k-3}^{\mathrm{T}}A\bar{r}_{k-1} = 0 - \beta_{k-1}(A\bar{Q}_{k-3})^{\mathrm{T}}\bar{r}_{k-1} = 0, \quad (31)$$

where the second equality is due to $\text{range}(Q_{k-1}) = \text{span}(q_{k-2}, q_{k-1}) \perp \text{range}(\bar{Q}_{k-3})$ and $A$ is SPD, the third equality is due to $\bar{r}_{k-1} \perp \text{range}(\bar{Q}_{k-1}) = A\mathcal{K}_{k-1}(A, r_0)$ and $\text{range}(A\bar{Q}_{k-3}) = A^2\mathcal{K}_{k-3}(A, r_0) \subseteq A\mathcal{K}_{k-1}(A, r_0)$. As a result, noting that $q = \Delta r_{k-1}$, we obtain

$$\bar{Q}_{k-3}^{\mathrm{T}}\tilde{q} = \bar{Q}_{k-3}^{\mathrm{T}}q - \bar{Q}_{k-3}^{\mathrm{T}}Q_{k-1}(Q_{k-1}^{\mathrm{T}}q) = 0,$$

which is due to (31) and $\text{range}(Q_{k-1}) = \text{span}\{q_{k-2}, q_{k-1}\} \perp \text{range}(\bar{Q}_{k-3})$. Therefore, we show that $\bar{Q}_k^{\mathrm{T}}\bar{Q} = I_k$, which along with $\bar{Q}_k = -A\bar{P}_k$ proves relation (ii) in the $k$-th iteration.

Next, we prove the relation (iv). We have

$$
\begin{aligned}
r_k &= \bar{r}_{k-1} - \beta_{k-1}A\bar{r}_{k-1} \\
&= r_{k-1} - Q_{k-1}\Gamma_{k-1} - \beta_{k-1}A(r_{k-1} - Q_{k-1}\Gamma_{k-1}) \\
&= r_{k-1} - Q_{k-1}\Gamma_{k-1} - \beta_{k-1}Ar_{k-1} + \beta_{k-1}AQ_{k-1}\Gamma_{k-1} \\
&= r_0 + \bar{Q}_{k-1}\bar{\Gamma}_{k-1} - Q_{k-1}\Gamma_{k-1} - \beta_{k-1}(Ar_0 + A\bar{Q}_{k-1}\bar{\Gamma}_{k-1}) + \beta_{k-1}AQ_{k-1}\Gamma_{k-1},
\end{aligned}
$$

where the last equality is due to $r_{k-1} = r_0 + \bar{Q}_{k-1}\bar{\Gamma}_{k-1}$ by the inductive hypothesis. Since $\text{range}(\bar{Q}_{k-1}) = A\mathcal{K}_{k-1}(A, r_0) \subseteq A\mathcal{K}_k(A, r_0), \text{range}(Q_{k-1}) \subseteq \text{range}(\bar{Q}_k), \text{span}\{Ar_0\} \subseteq A\mathcal{K}_k(A, r_0), \text{range}(AQ_{k-1}) \subseteq \text{range}(A\bar{Q}_{k-1}) \subseteq A^2\mathcal{K}_{k-1}(A, r_0) \subseteq A\mathcal{K}_k(A, r_0)$, and $\text{range}(\bar{Q}_k) = A\mathcal{K}_k(A, r_0)$, it is clear that $r_k = r_0 + \bar{Q}_k\bar{\Gamma}_k \in \mathcal{K}_{k+1}(A, r_0)$ for some $\bar{\Gamma}_k \in \mathbb{R}^k$. The relation (iv) is proved.

Finally, we prove the relation (iii). For proving $\bar{r}_k \perp \text{range}(\bar{Q}_k)$, note that $\bar{r}_k \perp \text{span}\{q_{k-1}, q_k\} = \text{range}(Q_k)$ already holds due to the projection step (Line 10 and Line 11 in Algorithm 2). It suffices to prove $\bar{r}_k \perp \text{range}(\bar{Q}_{k-2})$. In fact, since we have $\bar{r}_k = r_k - Q_k\Gamma_k$, we can prove that $r_k \perp \text{range}(\bar{Q}_{k-2})$ and $Q_k\Gamma_k \perp \text{range}(\bar{Q}_{k-2})$:

Since $\text{range}(Q_k) = \text{span}\{q_{k-1}, q_k\} \perp \text{range}(\bar{Q}_{k-2})$ as induced from $\bar{Q}_k^{\mathrm{T}}\bar{Q}_k = I_k$, it is clear that $Q_k\Gamma_k \perp \text{range}(\bar{Q}_{k-2})$. For $r_k$, according to Line 12 in Algorithm 2, we have $r_k = \bar{r}_{k-1} - \beta_{k-1}A\bar{r}_{k-1}$. We have $\bar{r}_{k-1} \perp \text{range}(\bar{Q}_{k-1}) \supseteq \text{range}(\bar{Q}_{k-2})$ by the inductive hypothesis. Also, $\bar{Q}_{k-2}^{\mathrm{T}}A\bar{r}_{k-1} = (A\bar{Q}_{k-2})^{\mathrm{T}}\bar{r}_{k-1} = 0$ due to $\text{range}(A\bar{Q}_{k-2}) = A^2\mathcal{K}_{k-2}(A, r_0) \subseteq A\mathcal{K}_{k-1}(A, r_0) = \text{range}(\bar{Q}_{k-1})$.

Therefore, we obtain $\bar{r}_k \perp \text{range}(\bar{Q}_{k-2})$, which along with $\bar{r}_k \perp \text{span}\{q_{k-1}, q_k\}$ implies $\bar{r}_k \perp \text{range}(\bar{Q}_k)$.

To prove $\bar{x}_k = x_k^{\mathrm{G}} := x_0 + z_k$, where $z_k = \arg\min_{z \in \mathcal{K}_k(A, r_0)} \|r_0 - Az\|_2$, first we have $r_k = r_0 + \bar{Q}_k\bar{\Gamma}_k$, where $\bar{\Gamma}_k \in \mathbb{R}^k$. Hence $\bar{r}_k = r_k - Q_k\Gamma_k = r_0 + \bar{Q}_k\bar{\Gamma}_k - Q_k\Gamma_k = r_0 - \bar{Q}_k\eta_k$, where $\eta_k \in \mathbb{R}^k$. Since $\bar{r}_k \perp \bar{Q}_k$, $\bar{r}_k$ is the orthogonal projection of $r_0$ onto the subspace $\text{range}(\bar{Q}_k)^{\perp}$. On the other side, for GMRES, $r_k^{\mathrm{G}} = r_0 - Az_k \perp A\mathcal{K}_k(A, r_0) = \text{range}(\bar{Q}_k)$, so $r_k^{\mathrm{G}}$ is also the

orthogonal projection of $r_0$ onto the subspace $\text{range}(\bar{Q}_k)^\perp$. So $\bar{r}_k = r_k^\text{G}$, which further indicates $\bar{x}_k = x_k^\text{G}$. Hence, the relation (iii) holds.

With relations (i)-(iv) being proved in the $k$-th iteration, we complete the induction. $\square$

## C.2 MODIFIED ST-AM FOR GENERAL FIXED-POINT ITERATIONS

Algorithm 2 is suitable for analysis and implementation in the linear case. For general nonlinear fixed-point iterations, we adopt an alternative form as described in Algorithm 3 which discards the normalization of $\tilde{q}$ in each iteration (Line 7 in Algorithm 2). In Line 7 in Algorithm 3, the orthogonal projection of $\Delta r_{k-1}$ is checked to ensure $\Delta r_{k-1}$ is "less linearly dependent" on $\text{range}(Q_k)$, which ensures $\|q_k\|_2$ is bounded away from zero; the check of $\|P_{k-1}\zeta_k\|_2$ ensures that $\|p_k - \Delta x_{k-1}\|_2 \le c_p\|\Delta x_{k-1}\|_2$, which is also important since a large deviation from $\Delta x_{k-1}$ can make $\|p_k\|_2 > \rho$ ($\rho$ is the radius introduced in Theorem 2). When this condition cannot be satisfied, the algorithm simply reuses the old $P_{k-1}, Q_{k-1}$. The main procedure of MST-AM restarts every $m$ iterations, i.e. $P_k, Q_k = \mathbf{0}$. Such restart mechanism is to restrict the higher-order terms in the residual expansion, as shown in (9). Also, restart can flush out the outdated historical information that may weaken the quality of $P_k$ and $Q_k$ that are used to pursue a local first-order approximation of $g$ in MST-AM.

---

**Algorithm 3** MST-AM for nonlinear fixed-point problems

---

**Input**: $x_0 \in \mathbb{R}^d, \beta_k \in (0, 1], c_p > 0, c_q \in (0, 1), m > 0$.
**Output**: $x \in \mathbb{R}^d$
1: $P_0, Q_0 = \mathbf{0} \in \mathbb{R}^{d \times 2}, p_0, q_0 = \mathbf{0} \in \mathbb{R}^d$
2: **for** $k = 0, 1, \ldots$, until convergence **do**
3:     $r_k = g(x_k) - x_k$
4:     **if** $k \mod m \ne 0$ **then**
5:         $p = x_k - x_{k-1}, q = r_k - r_{k-1}$
6:         $\zeta_k = (Q_{k-1}^\text{T}Q_{k-1})^\dagger Q_{k-1}^\text{T}q$
7:         **if** $\|P_{k-1}\zeta_k\|_2 \le c_p\|p\|_2$ and $\|Q_{k-1}\zeta_k\|_2 \le c_q\|q\|_2$ **then**
8:             $p_k = p - P_{k-1}\zeta_k, q_k = q - Q_{k-1}\zeta_k$
9:             $P_k = [p_{k-1}, p_k], Q_k = [q_{k-1}, q_k]$             $(q_k \perp Q_{k-1})$
10:       **else**
11:            $P_k = P_{k-1}, Q_k = Q_{k-1}$
12:       **end if**
13:     **else**
14:         $P_k, Q_k = \mathbf{0} \in \mathbb{R}^{d \times 2}, p_k, q_k = \mathbf{0} \in \mathbb{R}^d$
15:     **end if**
16:     $\Gamma_k = (Q_k^\text{T}Q_k)^\dagger Q_k^\text{T}r_k$
17:     $\bar{x}_k = x_k - P_k\Gamma_k, \bar{r}_k = r_k - Q_k\Gamma_k$         (Projection step: $\bar{r}_k \perp Q_k$)
18:     $x_{k+1} = \bar{x}_k + \beta_k\bar{r}_k$                  (Mixing step)
19: **end for**
20: **return** $x_k$
    (The notation "$\dagger$" is the Moore-Penrose pseudoinverse.)

---

In the linear case, Algorithm 2 and Algorithm 3 (with $m = \infty$) are equivalent. Similar to Algorithm 2, we have the following properties held for MST-AM:

**Claim 1.** *In the $k$-th iteration ($k > 0$) of Algorithm 1 applied to minimize a strongly convex quadratic problem (5), assuming $c_p = \infty, c_q = 1, m = \infty$, the following relations hold:*
*(i)* $\|q_k\|_2 > 0, \text{range}(\bar{P}_k) = \text{range}(X_k) = \mathcal{K}_k(A, r_0), \text{range}(\bar{Q}_k) = \text{range}(R_k) = A\mathcal{K}_k(A, r_0)$;
*(ii)* $\bar{Q}_k = -A\bar{P}_k, q_i \perp q_j (1 \le i \ne j \le k)$;
*(iii)* $\bar{r}_k \perp \text{range}(\bar{Q}_k)$ *and* $\bar{x}_k = x_0 + z_k$, *where* $z_k = \arg\min_{z \in \mathcal{K}_k(A, r_0)} \|r_0 - Az\|_2$.
*If* $\|\bar{r}_k\|_2 = 0$, *then* $x_{k+1}$ *is the exact solution.*

The proof of Claim 1 is essentially the same as the proof of Theorem 1, with a special care that $\bar{Q}_k^\text{T}\bar{Q}_k = I_k$ is replaced by the relation that columns of $\bar{Q}_k$ are orthogonal to each other, in other words, $\bar{Q}_k^\text{T}\bar{Q}_k = \text{diag}\{\|q_1\|_2^2, \ldots, \|q_k\|_2^2\}$.

For minimizing nonlinear functions or accelerating nonlinear fixed-point iterations, the long-term relation that $\bar{Q}_k^{\mathrm{T}} \bar{Q}_k = \mathrm{diag}\{\|q_1\|_2^2, \ldots, \|q_k\|_2^2\}$ generally cannot hold, while the orthogonalization procedure in Line 6 still leads to a short-term orthogonality relation: $q_i \perp q_j$ for $|i - j| \leq 2$. Hence, $Q_k^{\mathrm{T}} Q_k = \mathrm{diag}\{\|q_{k-1}\|_2^2, \|q_k\|_2^2\}$ for $k \geq 1$. Thus the pseudoinverse $(Q_k^{\mathrm{T}} Q_k)^\dagger = \mathrm{diag}\{\mathcal{I}(\|q_{k-1}\|_2 \neq 0)1/\|q_{k-1}\|_2^2, \mathcal{I}(\|q_k\|_2 \neq 0)1/\|q_k\|_2^2\}$, where $\mathcal{I}(\cdot)$ is the indicator function that

$$\mathcal{I}(x) = \left\{ \begin{array}{ll} 1 & x \text{ is true,} \\ 0 & x \text{ is false.} \end{array} \right.$$

Now, we give the proof of Theorem 2.

***Proof of Theorem 2.*** For convenience, we restate the main assumptions of $g$ here:

$$\text{(i) } \|g(y) - g(x)\|_2 \leq \kappa\|y - x\|_2, \ \kappa \in (0, 1), \text{ for } \forall x, y \in \mathcal{B}(\rho), \tag{32a}$$

$$\text{(ii) } \|g'(y) - g'(x)\|_2 \leq \hat{\kappa}\|y - x\|_2, \ \hat{\kappa} > 0, \text{ for } \forall x, y \in \mathcal{B}(\rho). \tag{32b}$$

Also, since $|1 - \beta_k| + \kappa\beta_k < 1$, we know $\beta_k > 0$ is bounded, i.e. $\beta_k \leq \beta$ where $\beta > 0$ is a constant.

The proof is based on the two lemmas given in Lemma 1 and Lemma 2. Besides (9), we also prove that $\|r_k\|_2 \leq \|r_0\|_2$ by induction.

For $k = 0$, $\bar{x}_0 = x_0 \in \mathcal{B}(\rho)$, and due to (48b), $\|x_1 - x^*\|_2 \leq \|x_0 - x^*\|_2 + \beta_0\|r_0\|_2 \leq \|x_0 - x^*\|_2 + \beta(1 + \kappa)\|x_0 - x^*\|_2 \leq \rho$ provided $\|x_0 - x^*\|_2 \leq \rho/(1 + \beta(1 + \kappa))$. Since

$$r_1 = g(x_1) - x_1 = g(x_1) - (x_0 + \beta_0 r_0) = g(x_1) - (x_0 + r_0) + (1 - \beta_0)r_0$$
$$= g(x_1) - g(x_0) + (1 - \beta_0)r_0,$$

it follows that

$$\|r_1\|_2 \leq \|g(x_1) - g(x_0)\|_2 + |1 - \beta_0|\|r_0\|_2$$
$$\leq \kappa\beta_0\|r_0\|_2 + |1 - \beta_0|\|r_0\|_2 = (\kappa\beta_0 + |1 - \beta_0|)\|r_0\|_2.$$

Also note that $\theta_0 = \|\bar{r}_0\|_2/\|r_0\|_2 = 1$. Thus (9) holds. Because $\kappa\beta_0 + |1 - \beta_0| \leq \kappa_0 < 1$, we have $\|r_1\|_2 < \|r_0\|_2$.

Now, suppose that (9) and $\|r_k\|_2 \leq \|r_0\|_2$ hold for $k \geq 0$. We establish the results for $k + 1$.

Let $\Gamma_k = (\gamma_k^{(1)}, \gamma_k^{(2)})^{\mathrm{T}} \in \mathbb{R}^2$. Since $\Gamma_k = (Q_k^{\mathrm{T}} Q_k)^\dagger Q_k^{\mathrm{T}} r_k$, $Q_k^{\mathrm{T}} Q_k = \mathrm{diag}\{\|q_{k-1}\|_2^2, \|q_k\|_2^2\}$, it follows that

$$\gamma_k^{(1)} = \mathcal{I}(q_{k-1} \neq 0) \frac{r_k^{\mathrm{T}} q_{k-1}}{q_{k-1}^{\mathrm{T}} q_{k-1}}, \ \gamma_k^{(2)} = \mathcal{I}(q_k \neq 0) \frac{r_k^{\mathrm{T}} q_k}{q_k^{\mathrm{T}} q_k}. \tag{33}$$

Therefore,

$$|\gamma_k^{(1)}| \leq \mathcal{I}(q_{k-1} \neq 0) \frac{\|r_k\|_2}{\|q_{k-1}\|_2}, \ |1 - \gamma_k^{(1)}| \leq \max\left\{ \mathcal{I}(q_{k-1} = 0), \mathcal{I}(q_{k-1} \neq 0) \frac{\|r_k - q_{k-1}\|_2}{\|q_{k-1}\|_2} \right\},$$

$$|\gamma_k^{(2)}| \leq \mathcal{I}(q_k \neq 0) \frac{\|r_k\|_2}{\|q_k\|_2}, \ |1 - \gamma_k^{(2)}| \leq \max\left\{ \mathcal{I}(q_k = 0), \mathcal{I}(q_k \neq 0) \frac{\|r_k - q_k\|_2}{\|q_k\|_2} \right\}. \tag{34}$$

Define $c = \frac{1 + c_p}{(1 - \kappa)(1 - c_q)}$. We have

$$\|P_k \Gamma_k\|_2 = \|p_k \gamma_k^{(2)} + p_{k-1} \gamma_{k-1}^{(1)}\|_2 \leq \|p_k \gamma_k^{(2)}\|_2 + \|p_{k-1} \gamma_{k-1}^{(1)}\|_2$$

$$\leq \mathcal{I}(q_k \neq 0)\|p_k\|_2 \frac{\|r_k\|_2}{\|q_k\|_2} + \mathcal{I}(q_{k-1} \neq 0)\|p_{k-1}\|_2 \frac{\|r_k\|_2}{\|q_{k-1}\|_2} \leq 2c\|r_k\|_2, \tag{35}$$

where the second inequality is due to (34) and the third inequality is due to (49). Then

$$\|\bar{x}_k - x^*\|_2 = \|x_k - P_k \Gamma_k - x^*\|_2 \leq \|x_k - x^*\|_2 + \|P_k \Gamma_k\|_2$$

$$\leq \frac{1}{1 - \kappa}\|r_k\|_2 + 2c\|r_k\|_2 = (\frac{1}{1 - \kappa} + 2c)\|r_k\|_2,$$

and due to $\|\bar{r}_k\|_2 \le \|r_k\|_2$, it holds that

$$\|x_{k+1} - x^*\|_2 = \|\bar{x}_k + \beta_k \bar{r}_k - x^*\|_2 \le \|\bar{x}_k - x^*\|_2 + \beta_k \|\bar{r}_k\|_2$$

$$\le \|\bar{x}_k - x^*\|_2 + \beta \|r_k\|_2 = (\frac{1}{1-\kappa} + 2c + \beta)\|r_k\|_2.$$

By the inductive hypothesis that $\|r_k\|_2 \le \|r_0\|_2$, and (48b), it has

$$\|\bar{x}_k - x^*\|_2 \le (\frac{1}{1-\kappa} + 2c)\|r_0\|_2 \le \left(\frac{1}{1-\kappa} + 2c\right)(1+\kappa)\|x_0 - x^*\|_2, \qquad (36)$$

and

$$\|x_{k+1} - x^*\|_2 \le \left(\frac{1}{1-\kappa} + 2c + \beta\right)\|r_0\|_2 \le \left(\frac{1}{1-\kappa} + 2c + \beta\right)(1+\kappa)\|x_0 - x^*\|_2. \qquad (37)$$

As a result, we can choose $\|x_0 - x^*\|_2$ sufficiently small to ensure $\bar{x}_k \in \mathcal{B}(\rho)$ and $x_{k+1} \in \mathcal{B}(\rho)$, which ensure the $g(\bar{x}_k)$ and $g(x_{k+1})$ are well defined.

At the end of the $k$-th iteration of Algorithm 3, we have

$$\begin{aligned} r_{k+1} &= g(x_{k+1}) - x_{k+1} \\ &= g(x_{k+1}) - g(\bar{x}_k) + g(\bar{x}_k) - (\bar{x}_k + \beta_k \bar{r}_k) \\ &= (g(x_{k+1}) - g(\bar{x}_k)) + (g(\bar{x}_k) - \bar{x}_k - \bar{r}_k) + (1 - \beta_k)\bar{r}_k. \end{aligned} \qquad (38)$$

Let $\mathcal{L}_k := g(x_{k+1}) - g(\bar{x}_k) + (1 - \beta_k)\bar{r}_k$, $\mathcal{H}_k := g(\bar{x}_k) - \bar{x}_k - \bar{r}_k$, then

$$\begin{aligned} \|\mathcal{L}_k\|_2 &\le \kappa \|x_{k+1} - \bar{x}_k\|_2 + |1 - \beta_k|\|\bar{r}_k\|_2 \\ &= \kappa\beta_k \|\bar{r}_k\|_2 + |1 - \beta_k|\|\bar{r}_k\|_2 \\ &= \theta_k(\kappa\beta_k + |1 - \beta_k|)\|r_k\|_2, \end{aligned} \qquad (39)$$

which bounds the linear part of the residual $r_{k+1}$.

For the higher-order terms $\mathcal{H}_k$, we have

$$\begin{aligned} \mathcal{H}_k &= g(\bar{x}_k) - (x_k - P_k\Gamma_k + r_k - Q_k\Gamma_k) \\ &= g(\bar{x}_k) - g(x_k) + (P_k + Q_k)\Gamma_k. \\ &= g(\bar{x}_k) - g(x_k) + (p_k + q_k)\gamma_k^{(2)} + (p_{k-1} + q_{k-1})\gamma_k^{(1)}. \end{aligned} \qquad (40)$$

According to the formula $\int_0^1 g'(x + t(y-x))(y-x)dt = g(y) - g(x)$, we have

$$\begin{aligned} g(\bar{x}_k) - g(x_k) &= g(\bar{x}_k) - g(x_k - p_k\gamma_k^{(2)}) + g(x_k - p_k\gamma_k^{(2)}) - g(x_k) \\ &= \int_0^1 -g'(x_k - p_k\gamma_k^{(2)} - tp_{k-1}\gamma_k^{(1)})p_{k-1}\gamma_k^{(1)}dt \\ &\quad + \int_0^1 -g'(x_k - tp_k\gamma_k^{(2)})p_k\gamma_k^{(2)}dt. \end{aligned} \qquad (41)$$

Also, by Lemma 2, we have

$$p_k + q_k = \int_0^1 g'(x_k - tp_k)p_k dt + \hat{\kappa}\|\Delta r_{\pi(k)-1}\|_2 \sum_{j=k-m_k}^{k-1} \mathcal{O}(\|\Delta r_j\|_2),$$

$$p_{k-1} + q_{k-1} = \int_0^1 g'(x_{k-1} - tp_{k-1})p_{k-1}dt + \hat{\kappa}\|\Delta r_{\upsilon(k)-1}\|_2 \sum_{j=k-m_k}^{k-2} \mathcal{O}(\|\Delta r_j\|_2),$$

where $\pi(k)$ denotes that the latest update of $q_k$ by Line 8 occurred in the $\pi(k)$-th iteration and $\upsilon(k) = \pi(\pi(k)-1)$ marks that $q_{k-1}$ records $q_{\upsilon(k)}$ that is the penultimate update by Line 8 occurring in the $\upsilon(k)$-th iteration.

By substituting these relations to (40), it follows that

$$
\begin{aligned}
\mathcal{H}_k =& \left( \int_0^1 g'(x_k - tp_k)p_k dt + \hat{\kappa}\|\Delta r_{\pi(k)-1}\|_2 \sum_{j=k-m_k}^{k-1} \mathcal{O}(\|\Delta r_j\|_2) \right) \gamma_k^{(2)} \\
&+ \left( \int_0^1 g'(x_{k-1} - tp_{k-1})p_{k-1} dt + \hat{\kappa}\|\Delta r_{\upsilon(k)-1}\|_2 \sum_{j=k-m_k}^{k-2} \mathcal{O}(\|\Delta r_j\|_2) \right) \gamma_k^{(1)} \\
&- \int_0^1 g'(x_k - p_k\gamma_k^{(2)} - tp_{k-1}\gamma_k^{(1)})p_{k-1}\gamma_k^{(1)} dt - \int_0^1 g'(x_k - tp_k\gamma_k^{(2)})p_k\gamma_k^{(2)} dt \\
=& \int_0^1 (g'(x_k - tp_k) - g'(x_k - tp_k\gamma_k^{(2)}))p_k\gamma_k^{(2)} dt \\
&+ \int_0^1 (g'(x_{k-1} - tp_{k-1}) - g'(x_k - p_k\gamma_k^{(2)} - tp_{k-1}\gamma_k^{(1)}))p_{k-1}\gamma_k^{(1)} dt \\
&+ \hat{\kappa}\|\Delta r_{\pi(k)-1}\|_2 \sum_{j=k-m_k}^{k-1} \mathcal{O}(\|\Delta r_j\|_2)\gamma_k^{(2)} \\
&+ \hat{\kappa}\|\Delta r_{\upsilon(k)-1}\|_2 \sum_{j=k-m_k}^{k-2} \mathcal{O}(\|\Delta r_j\|_2)\gamma_k^{(1)} \\
=& A_k + B_k + C_k + D_k. \tag{42}
\end{aligned}
$$

Then we can bound each terms of $\mathcal{H}_k$ as follows (Here we assume $q_k \neq 0$ and $q_{k-1} \neq 0$ as $q_k = 0$ leads to $\gamma_k^{(2)} = 0$ and $q_{k-1} = 0$ leads to $\gamma_k^{(1)} = 0$ where the result is trivial.):

$$
\begin{aligned}
\|A_k\|_2 &= \left\| \int_0^1 (g'(x_k - tp_k) - g'(x_k - tp_k\gamma_k^{(2)}))p_k\gamma_k^{(2)} dt \right\|_2 \\
&\leq \int_0^1 \|g'(x_k - tp_k) - g'(x_k - tp_k\gamma_k^{(2)})\|_2 \|p_k\gamma_k^{(2)}\|_2 dt \\
&\leq \hat{\kappa} \int_0^1 \|tp_k - tp_k\gamma_k^{(2)}\|_2 \|p_k\|_2 |\gamma_k^{(2)}| dt \\
&= \frac{\hat{\kappa}}{2} \|p_k\|_2^2 |1 - \gamma_k^{(2)}||\gamma_k^{(2)}| \\
&= \frac{\hat{\kappa}}{2} \|p_k\|_2^2 \frac{\mathcal{O}(\|r_k\|_2\|r_k - q_k\|_2)}{\|q_k\|_2^2} = \hat{\kappa} \sum_{j=k-m_k}^{k} \mathcal{O}(\|r_j\|_2^2), \tag{43}
\end{aligned}
$$

where the third equality is due to (34), and the last equality is due to (49) and (50b);

$$
\begin{aligned}
\|B_k\|_2 &= \left\| \int_0^1 (g'(x_{k-1} - tp_{k-1}) - g'(x_k - p_k\gamma_k^{(2)} - tp_{k-1}\gamma_k^{(1)}))p_{k-1}\gamma_k^{(1)} dt \right\|_2 \\
&\leq \int_0^1 \|g'(x_{k-1} - tp_{k-1}) - g'(x_k - p_k\gamma_k^{(2)} - tp_{k-1}\gamma_k^{(1)})\|_2 \|p_{k-1}\gamma_k^{(1)}\|_2 dt \\
&\leq \hat{\kappa} \int_0^1 \|\Delta x_{k-1} - p_k\gamma_k^{(2)} - tp_{k-1}\gamma_k^{(1)} + tp_{k-1}\|_2 \|p_{k-1}\|_2 |\gamma_k^{(1)}| dt \\
&\leq \hat{\kappa} \left( \|\Delta x_{k-1}\|_2 + \|p_k\|_2 |\gamma_k^{(2)}| + \frac{1}{2}\|p_{k-1}\|_2 |1 - \gamma_k^{(1)}| \right) \|p_{k-1}\|_2 |\gamma_k^{(1)}| \\
&\leq \hat{\kappa} \left( \|\Delta x_{k-1}\|_2 + \|p_k\|_2 \frac{\|r_k\|_2}{\|q_k\|_2} + \frac{1}{2}\|p_{k-1}\|_2 \frac{\|r_k - q_{k-1}\|_2}{\|q_{k-1}\|_2} \right) \|p_{k-1}\|_2 \frac{\|r_k\|_2}{\|q_{k-1}\|_2} \\
&= \hat{\kappa}\|r_k\|_2 (\mathcal{O}(\|\Delta r_{k-1}\|_2) + \mathcal{O}(\|r_k\|_2) + \mathcal{O}(\|r_k - q_{k-1}\|_2) \\
&= \hat{\kappa} \sum_{j=k-m_k}^{k} \mathcal{O}(\|r_j\|_2^2), \tag{44}
\end{aligned}
$$

where the last inequality is due to (34), and the second equality is due to (48a), (49), (50b);

$$\|C_k\|_2 = \hat{\kappa}\|\Delta r_{\pi(k)-1}\|_2 \sum_{j=k-m_k}^{k-1} \mathcal{O}(\|\Delta r_j\|_2)\gamma_k^{(2)}$$

$$\leq \hat{\kappa}\|\Delta r_{\pi(k)-1}\|_2 \frac{\|r_k\|_2}{\|q_k\|_2} \sum_{j=k-m_k}^{k-1} \mathcal{O}(\|\Delta r_j\|_2)$$

$$= \hat{\kappa}\|\Delta r_{\pi(k)-1}\|_2 \frac{\|r_k\|_2}{\|q_{\pi(k)}\|_2} \sum_{j=k-m_k}^{k-1} \mathcal{O}(\|\Delta r_j\|_2)$$

$$= \hat{\kappa} \sum_{j=k-m_k}^{k} \mathcal{O}(\|r_j\|_2^2), \tag{45}$$

where the first inequality is from (34), and the second equality is due to (50b);

$$\|D_k\|_2 = \hat{\kappa}\|\Delta r_{\upsilon(k)-1}\|_2 \sum_{j=k-m_k}^{k-2} \mathcal{O}(\|\Delta r_j\|_2)\gamma_k^{(1)}$$

$$\leq \hat{\kappa}\|\Delta r_{\upsilon(k)-1}\|_2 \frac{\|r_k\|_2}{\|q_{k-1}\|_2} \sum_{j=k-m_k}^{k-2} \mathcal{O}(\|\Delta r_j\|_2)$$

$$= \hat{\kappa}\|\Delta r_{\upsilon(k)-1}\|_2 \frac{\|r_k\|_2}{\|q_{\upsilon(k)}\|_2} \sum_{j=k-m_k}^{k-2} \mathcal{O}(\|\Delta r_j\|_2)$$

$$= \hat{\kappa} \sum_{j=k-m_k}^{k} \mathcal{O}(\|r_j\|_2^2), \tag{46}$$

where the first inequality is from (34), and the second equality is due to (50b). Then with the bounds (43), (44), (45) and (46), we obtain

$$\|\mathcal{H}_k\|_2 = \hat{\kappa} \sum_{j=k-m_k}^{k} \mathcal{O}(\|r_j\|_2^2). \tag{47}$$

Combining (39) and (47) to (38), we obtain

$$\|r_{k+1}\| \leq \|\mathcal{L}_k\|_2 + \|\mathcal{H}_k\|_2 \leq \theta_k(\kappa\beta_k + |1-\beta_k|)\|r_k\|_2 + \hat{\kappa} \sum_{j=k-m_k}^{k} \mathcal{O}(\|r_j\|_2^2),$$

as desired. Since $m_k \leq m$, the higher-order terms are limited. Note that

$$\kappa\beta_k + |1-\beta_k| \leq \kappa_0 < 1$$

by assumption. Then, for $\|x_0 - x^*\|_2$ sufficiently small, the residuals $\{r_k\}$ are $Q$-linearly convergent, which infers $\|r_k\|_2 \leq \|r_0\|_2$. Therefore, we complete the induction. □

**Remark 6.** *There may be some concern about whether $\hat{\kappa}$ can be counteracted by the constant hidden in the Big-$\mathcal{O}$ notation. In fact, since $\hat{\kappa}$ is the Lipschitz constant of $g'$ and the constants in $\mathcal{O}(\cdot)$ are composed of $\kappa, c_p, c_q$, it follows that $\hat{\kappa}$ is unrelated to $\mathcal{O}(\cdot)$. Hence a small $\hat{\kappa}$ can lead to a small uniform boundedness of the higher-order terms. In the extreme case where $\hat{\kappa} = 0$, e.g. $g$ is a linear map, the residual only consists of the first-order term $\mathcal{L}_k$.*

**Lemma 1.** *Under the same assumptions of Theorem 2, for $k \geq 1$, we have the following bounds:*

$$(1-\kappa)\|\Delta x_{k-1}\|_2 \leq \|\Delta r_{k-1}\|_2 \leq (1+\kappa)\|\Delta x_{k-1}\|_2, \tag{48a}$$

$$(1-\kappa)\|x_k - x^*\|_2 \leq \|r_k\|_2 \leq (1+\kappa)\|x_k - x^*\|_2, \tag{48b}$$

*If $q_k \neq 0$, then*

$$\frac{\|p_k\|_2}{\|q_k\|_2} \leq \frac{1+c_p}{(1-\kappa)(1-c_q)} \tag{49}$$

*If the condition in Line 7 is true, then*

$$\|p_k\|_2 \le (1 + c_p)\|\Delta x_{k-1}\|_2, \tag{50a}$$
$$q_k \ne 0, \ (1 - c_q)\|\Delta r_{k-1}\| \le \|q_k\|_2 \le \|\Delta r_{k-1}\|_2 \tag{50b}$$

*Proof.* From the assumption (32a) of $g$, we have

$$
\begin{aligned}
(1 - \kappa)\|\Delta x_{k-1}\|_2 &\le \|x_k - x_{k-1}\|_2 - \|g(x_k) - g(x_{k-1})\|_2 \\
&\le \|g(x_k) - g(x_{k-1}) - (x_k - x_{k-1})\|_2 \\
&= \|r_k - r_{k-1}\|_2 = \|\Delta r_{k-1}\|_2 \\
&\le \|g(x_k) - g(x_{k-1})\|_2 + \|x_k - x_{k-1}\| \\
&\le (1 + \kappa)\|x_k - x_{k-1}\|_2,
\end{aligned}
$$

$$
\begin{aligned}
(1 - \kappa)\|x_k - x^*\|_2 &\le \|x_k - x^*\|_2 - \|g(x_k) - g(x^*)\|_2 \\
&\le \|g(x_k) - g(x^*) - (x_k - x^*)\|_2 = \|r_k\|_2 \\
&\le \|g(x_k) - g(x^*)\|_2 + \|x_k - x^*\|_2 \\
&\le (1 + \kappa)\|x_k - x^*\|_2.
\end{aligned}
$$

If the condition in Line 7 is true, i.e., $\|P_{k-1}\zeta_k\|_2 \le c_p\|\Delta x_{k-1}\|_2, \|Q_{k-1}\zeta_k\|_2 \le c_q\|\Delta r_{k-1}\|_2$, we have

$$\|p_k\|_2 = \|\Delta x_{k-1} - P_{k-1}\zeta_k\|_2 \le \|\Delta x_{k-1}\|_2 + \|P_{k-1}\zeta_k\|_2 \le (1 + c_p)\|\Delta x_{k-1}\|_2,$$

$$\|q_k\|_2 = \|\Delta r_{k-1} - Q_{k-1}\zeta_k\|_2 \ge \|\Delta r_{k-1}\|_2 - \|Q_{k-1}\zeta_k\|_2 \ge (1 - c_q)\|\Delta r_{k-1}\|_2.$$

The inequality $\|q_k\|_2 \le \|\Delta r_{k-1}\|$ is due to the fact that $q_k$ is the orthogonal projection of $\Delta r_{k-1}$ onto $\text{range}(Q_{k-1})^\perp$. Also, $q_k \ne 0$ must hold otherwise $q = Q_{k-1}\zeta_k$ which violates the condition $\|Q_{k-1}\zeta_k\|_2 \le c_q\|\Delta r_{k-1}\|_2$ as $c_q \in (0, 1)$.

If $q_k \ne 0$, then $q_k$ must be updated by Line 8 in some previous iteration. We assume the latest update by Line 8 occurred in the $j$-th iteration, i.e., $q_k = q_j, p_k = p_j$, hence

$$\frac{\|p_k\|_2}{\|q_k\|_2} = \frac{\|p_j\|_2}{\|q_j\|_2} \le \frac{(1 + c_p)\|\Delta x_{j-1}\|_2}{(1 - c_q)\|\Delta r_{j-1}\|_2} \le \frac{1 + c_p}{(1 - c_q)(1 - \kappa)}.$$

where the first inequality is due to (50a) and (50b) and the second inequality is due to (48a). $\qquad \square$

**Lemma 2.** *Under the same assumptions of Theorem 2, in the $k$-th iteration ($k \ge 0$) of the restarted MST-AM (Algorithm 3), we have*

$$p_k + q_k = \int_0^1 g'(x_k - tp_k)p_k dt + \hat{\kappa}\|\Delta r_{\pi(k)-1}\|_2 \sum_{j=k-m_k}^{k-1} \mathcal{O}(\|\Delta r_j\|_2), \tag{51}$$

*where $m_k = k \mod m$, $\Delta r_{k-m_k-1} := r_{k-m_k}$, $\pi(k)$ denotes that the latest update of $q_k$ by Line 8 occurred in the $\pi(k)$-th iteration and $\pi(k) = k - m_k$ if Line 8 is never executed up to the $k$-iteration.*

*Proof.* We prove (51) by induction. Denote $\zeta_k = (\zeta_k^{(1)}, \zeta_k^{(2)})^\mathrm{T}$.

For $k = 0$, the relation trivially holds.

For $k = 1$, $p_1 = \Delta x_0, q_1 = \Delta r_0$. It follows that

$$p_1 + q_1 = g(x_1) - g(x_0) = \int_0^1 g'(x_1 - tp_1)p_1 dt.$$

So (51) holds for $k = 1$.

Suppose that (51) holds for $0 \le j \le k - 1$, where $k \ge 2$. For $k \ge 2$, if $m_k := k \mod m = 0$ or 1, (51) holds because it is the same as the case of $k = 0$ or $k = 1$. Now consider the nontrivial cases.

For $m_k \neq 0$ and $m_k \neq 1$, if the condition in Line 7 in Algorithm 3 is true, then $\pi(k) = k$. With the convention that $\Delta r_{k-m_k-1} = r_{k-m_k}$, we have

$$
\begin{aligned}
p_k + q_k &= \Delta x_{k-1} - P_{k-1}\zeta_k + \Delta r_{k-1} - Q_{k-1}\zeta_k \\
&= g(x_k) - g(x_{k-1}) - (p_{k-1} + q_{k-1})\zeta_k^{(2)} - (p_{k-2} + q_{k-2})\zeta_k^{(1)} \\
&= \int_0^1 g'(x_k - t\Delta x_{k-1})\Delta x_{k-1} dt \\
&\quad - \int_0^1 g'(x_{k-1} - tp_{k-1})p_{k-1}\zeta_k^{(2)} dt + \hat{\kappa}\|\Delta r_{\pi(k-1)-1}\|_2 \sum_{j=k-1-m_{k-1}}^{k-2} \mathcal{O}(\|\Delta r_j\|_2)\zeta_k^{(2)} \\
&\quad - \int_0^1 g'(x_{k-2} - tp_{k-2})p_{k-2}\zeta_k^{(1)} dt + \hat{\kappa}\|\Delta r_{\upsilon(k-1)-1}\|_2 \sum_{j=k-2-m_{k-2}}^{k-3} \mathcal{O}(\|\Delta r_j\|_2)\zeta_k^{(1)},
\end{aligned}
\tag{52}
$$

where $\upsilon(k-1) = \pi(\pi(k-1)-1)$ denotes that $q_{k-2}$ records $q_{\upsilon(k-1)}$ that is the penultimate update by Line 8 occurring in the $\upsilon(k-1)$-th iteration. Considering the terms in $p_k + q_k$, we know

$$
\begin{aligned}
&g'(x_k - t\Delta x_{k-1})\Delta x_{k-1} - g'(x_{k-1} - tp_{k-1})p_{k-1}\zeta_k^{(2)} - g'(x_{k-2} - tp_{k-2})p_{k-2}\zeta_k^{(1)} \\
&= g'(x_k - t\Delta x_{k-1})(\Delta x_{k-1} - p_{k-1}\zeta_k^{(2)} - p_{k-2}\zeta_k^{(1)}) \\
&\quad + (g'(x_k - t\Delta x_{k-1}) - g'(x_{k-1} - tp_{k-1}))p_{k-1}\zeta_k^{(2)} \\
&\quad + (g'(x_k - t\Delta x_{k-1}) - g'(x_{k-2} - tp_{k-2}))p_{k-2}\zeta_k^{(1)}.
\end{aligned}
\tag{53}
$$

For $\zeta_k = (Q_{k-1}^T Q_{k-1})^\dagger Q_{k-1}^T \Delta r_{k-1}$, because $Q_{k-1}^T Q_{k-1} = \text{diag}\{\|q_{k-2}\|_2^2, \|q_{k-1}\|_2^2\}$, it follows that

$$
\zeta_k^{(1)} = \mathcal{I}(q_{k-2} \neq 0)\frac{\Delta r_{k-1}^T q_{k-2}}{q_{k-2}^T q_{k-2}}, \quad \zeta_k^{(2)} = \mathcal{I}(q_{k-1} \neq 0)\frac{\Delta r_{k-1}^T q_{k-1}}{q_{k-1}^T q_{k-1}}.
$$

Therefore,

$$
|\zeta_k^{(1)}| \leq \mathcal{I}(q_{k-2} \neq 0)\frac{\|\Delta r_{k-1}\|_2}{\|q_{k-2}\|_2}, \quad |\zeta_k^{(2)}| \leq \mathcal{I}(q_{k-1} \neq 0)\frac{\|\Delta r_{k-1}\|_2}{\|q_{k-1}\|_2}.
\tag{54}
$$

Now, to bound the terms in (53), we have

$$
\begin{aligned}
&\|(g'(x_k - t\Delta x_{k-1}) - g'(x_{k-1} - tp_{k-1}))p_{k-1}\zeta_k^{(2)}\|_2 \\
&\leq \hat{\kappa}\|x_k - x_{k-1} - t(\Delta x_{k-1} - p_{k-1})\|_2\|p_{k-1}\zeta_k^{(2)}\|_2 \\
&\leq \hat{\kappa}((1-t)\|\Delta x_{k-1}\|_2 + t\|p_{k-1}\|_2)\|p_{k-1}\|_2|\zeta_k^{(2)}| \\
&\leq \hat{\kappa}((1-t)\|\Delta x_{k-1}\|_2 + t\|p_{k-1}\|_2)\|p_{k-1}\|_2\mathcal{I}(q_{k-1} \neq 0)\frac{\|\Delta r_{k-1}\|_2}{\|q_{k-1}\|_2} \\
&\leq \hat{\kappa}\|\Delta r_{k-1}\|_2 \sum_{j=k-1-m_{k-1}}^{k-1} \mathcal{O}(\|\Delta r_j\|_2) = \hat{\kappa}\|\Delta r_{k-1}\|_2 \sum_{j=k-m_k}^{k-1} \mathcal{O}(\|\Delta r_j\|_2),
\end{aligned}
\tag{55}
$$

where the third inequality is due to (54), and the last inequality is due to (48a), (50a) and (49), where the constants are absorbed into the big-$\mathcal{O}$ notation. Similarly, it follows that

$$
\begin{aligned}
&\|(g'(x_k - t\Delta x_{k-1}) - g'(x_{k-2} - tp_{k-2}))p_{k-2}\zeta_k^{(1)}\|_2 \\
&\leq \hat{\kappa}\|\Delta x_{k-1} + \Delta x_{k-2} - t\Delta x_{k-1} + tp_{k-2}\|_2\|p_{k-2}\zeta_k^{(1)}\|_2 \\
&\leq \hat{\kappa}((1-t)\|\Delta x_{k-1}\|_2 + \|\Delta x_{k-2}\|_2 + t\|p_{k-2}\|_2)\|p_{k-2}\|_2|\zeta_k^{(1)}| \\
&\leq \hat{\kappa}((1-t)\|\Delta x_{k-1}\|_2 + \|\Delta x_{k-2}\|_2 + t\|p_{k-2}\|_2)\|p_{k-2}\|_2\mathcal{I}(q_{k-2} \neq 0)\frac{\|\Delta r_{k-1}\|_2}{\|q_{k-2}\|_2} \\
&\leq \hat{\kappa}\|\Delta r_{k-1}\|_2 \sum_{j=k-2-m_{k-2}}^{k-1} \mathcal{O}(\|\Delta r_j\|_2) = \hat{\kappa}\|\Delta r_{k-1}\|_2 \sum_{j=k-m_k}^{k-1} \mathcal{O}(\|\Delta r_j\|_2).
\end{aligned}
\tag{56}
$$

For the remaining terms in (52), we have

$$\hat{\kappa}\|\Delta r_{\pi(k-1)-1}\|_2 \sum_{j=k-1-m_{k-1}}^{k-2} \mathcal{O}(\|\Delta r_j\|_2)\zeta_k^{(2)} + \hat{\kappa}\|\Delta r_{\upsilon(k-1)-1}\|_2 \sum_{k=k-2-m_{k-2}}^{k-3} \mathcal{O}(\|\Delta r_j\|_2)\zeta_k^{(1)}$$

$$\leq \hat{\kappa}\|\Delta r_{\pi(k-1)-1}\|_2 \mathcal{I}(q_{k-1} \neq 0)\frac{\|\Delta r_{k-1}\|_2}{\|q_{k-1}\|_2} \sum_{j=k-1-m_{k-1}}^{k-2} \mathcal{O}(\|\Delta r_j\|_2)$$

$$+ \hat{\kappa}\|\Delta r_{\upsilon(k-1)-1}\|_2 \mathcal{I}(q_{k-2} \neq 0)\frac{\|\Delta r_{k-1}\|_2}{\|q_{k-2}\|_2} \sum_{k=k-2-m_{k-2}}^{k-3} \mathcal{O}(\|\Delta r_j\|_2)$$

$$\leq \hat{\kappa}\|\Delta r_{k-1}\|_2 \sum_{k=k-m_k}^{k-2} \mathcal{O}(\|\Delta r_j\|_2), \tag{57}$$

where the last inequality is due to (50b) and that $q_{k-1} = q_{\pi(k-1)}, q_{k-2} = q_{\upsilon(k-1)}$.

With relations (52), (53), (55), (56) and (57), and noting that $p_k = \Delta x_{k-1} - p_{k-1}\zeta_k^{(2)} - p_{k-2}\zeta_k^{(1)}$, and

$$\left\|\int_0^1 (g'(x_k - t\Delta x_{k-1})\Delta x_{k-1} - g'(x_{k-1} - tp_{k-1})p_{k-1}\zeta_k^{(2)} - g'(x_{k-2} - tp_{k-2})p_{k-2}\zeta_k^{(1)})dt\right\|_2$$

$$\leq \int_0^1 \|g'(x_k - t\Delta x_{k-1})\Delta x_{k-1} - g'(x_{k-1} - tp_{k-1})p_{k-1}\zeta_k^{(2)} - g'(x_{k-2} - tp_{k-2})p_{k-2}\zeta_k^{(1)}\|_2 dt,$$

we can estimate $p_k + q_k$ as

$$p_k + q_k = \int_0^1 g'(x_k - t\Delta x_{k-1})p_k dt + \hat{\kappa}\|\Delta r_{k-1}\|_2 \sum_{j=k-m_k}^{k-1} \mathcal{O}(\|\Delta r_j\|_2). \tag{58}$$

To further obtain (51), notice that the difference

$$\left\|\int_0^1 g'(x_k - tp_k)p_k dt - \int_0^1 g'(x_k - t\Delta x_{k-1})p_k dt\right\|_2$$

$$= \left\|\int_0^1 (g'(x_k - tp_k) - g'(x_k - t\Delta x_{k-1}))p_k dt\right\|_2$$

$$\leq \int_0^1 \|g'(x_k - tp_k) - g'(x_k - t\Delta x_{k-1})\|_2 \|p_k\|_2 dt$$

$$\leq \hat{\kappa} \int_0^1 t\|p_{k-1}\zeta_k^{(2)} + p_{k-2}\zeta_k^{(1)}\|_2 \|p_k\|_2 dt$$

$$\leq \frac{\hat{\kappa}}{2}c_p\|\Delta x_{k-1}\|_2 \|p_k\|_2 = \hat{\kappa}\|\Delta r_{k-1}\|_2 \mathcal{O}(\|\Delta r_{k-1}\|_2),$$

where the last inequality is due to the condition $\|P_{k-1}\zeta_k\|_2 \leq c_p\|p\|_2$ and inequalities (48a) and (50a). Hence the difference can be absorbed in the Big-$\mathcal{O}$ notation in (58). Thus (51) holds when the condition in Line 7 is true for the $k$-th iteration.

We consider the case where the condition in Line 7 is false for the $k$-th iteration. Then $P_k = P_{k-1}, Q_k = Q_{k-1}$. In other words, the memory recording $p_k, q_k$ keeps unchanged from the $(\pi(k)+1)$-th to $k$-th iteration. Since $\pi(k) < k$ and $\pi(\pi(k)) = \pi(k)$, with the inductive hypothesis, we have

$$p_k + q_k = p_{\pi(k)} + q_{\pi(k)}$$

$$= \int_0^1 g'(x_{\pi(k)} - tp_{\pi(k)})p_{\pi(k)}dt + \hat{\kappa}\|\Delta r_{\pi(k)-1}\|_2 \sum_{j=\pi(k)-m_{\pi(k)}}^{\pi(k)-1} \mathcal{O}(\|\Delta r_j\|_2)$$

$$= \int_0^1 g'(x_{\pi(k)} - tp_k)p_k dt + \hat{\kappa}\|\Delta r_{\pi(k)-1}\| \sum_{j=k-m_k}^{k-1} \mathcal{O}(\|\Delta r_j\|_2). \tag{59}$$

To further obtain (51), note that the difference

$$\left\| \int_0^1 g'(x_k - tp_k)p_k dt - \int_0^1 g'(x_{\pi(k)} - tp_k)p_k dt \right\|_2$$

$$\leq \int_0^1 \|g'(x_k - tp_k) - g'(x_{\pi(k)} - tp_k)\|_2 \|p_k\|_2 dt$$

$$\leq \hat{\kappa} \|x_k - x_{\pi(k)}\|_2 \|p_k\|_2$$

$$= \hat{\kappa} \sum_{j=\pi(k)}^{k-1} \|\Delta x_j\|_2 \|p_k\|_2 = \hat{\kappa} \|\Delta r_{\pi(k)-1}\|_2 \sum_{j=\pi(k)}^{k-1} \mathcal{O}(\|\Delta r_j\|_2)$$

can be absorbed into the Big-$\mathcal{O}$ notation in (59). Thus (51) also holds when the condition in Line 7 is false for the $k$-th iteration.

As a result, we complete the induction in the $k$-th iteration. $\square$

### C.3 REGULARIZED SHORT-TERM RECURRENCE ANDERSON MIXING

#### C.3.1 CHECK OF POSITIVE DEFINITENESS

We describe the check of positive definiteness in RST-AM, which follows the same procedure as that of SAM (Wei et al., 2021). From Line 11-13 in Algorithm 1, one update of $x_k$ RST-AM is $x_{k+1} = x_k + H_k r_k$, where $H_k = \beta_k I - \alpha_k Y_k Z_k^\dagger Q_k^T$, $Y_k = P_k + \beta_k Q_k$, $Z_k = Q_k^T Q_k + \delta_k P_k^T P_k$. $H_k$ is generally not symmetric. For the convergence analysis of RST-AM, a critical condition is the *positive definiteness* of $H_k$, i.e.

$$s_k^T H_k s_k \geq \beta_k \mu \|s_k\|_2^2, \quad \forall s_k \in \mathbb{R}^d, \tag{60}$$

where $\mu \in (0,1)$ is a constant. Next, we show how to guarantee it.

Let $\lambda_{min}(\cdot)$ denote the smallest eigenvalue, and $\lambda_{max}(\cdot)$ denote the largest eigenvalue. Since $s_k^T H_k s_k = \frac{1}{2} s_k^T (H_k + H_k^T) s_k$, Condition (60) is equivalent to $\lambda_{min}\left(\frac{1}{2}\left(H_k + H_k^T\right)\right) \geq \beta_k \mu$. By some simple algebraic operations, we obtain $\lambda_{min}\left(\frac{1}{2}\left(H_k + H_k^T\right)\right) = \beta_k - \frac{1}{2}\alpha_k \lambda_{max}(Y_k Z_k^\dagger Q_k^T + Q_k Z_k^\dagger Y_k^T)$. Let $\lambda_k := \lambda_{max}(Y_k Z_k^\dagger Q_k^T + Q_k Z_k^\dagger Y_k^T)$, then Condition (60) is equivalent to

$$\alpha_k \lambda_k \leq 2\beta_k (1 - \mu), \tag{61}$$

namely, (13) in Remark 4. To check Condition (61), note that

$$\lambda_k = \lambda_{max}\left( (Y_k \quad Q_k) \begin{pmatrix} 0 & Z_k^\dagger \\ Z_k^\dagger & 0 \end{pmatrix} \begin{pmatrix} Y_k^T \\ Q_k^T \end{pmatrix} \right) = \lambda_{max}\left( \begin{pmatrix} Y_k^T \\ Q_k^T \end{pmatrix} (Y_k \quad Q_k) \begin{pmatrix} 0 & Z_k^\dagger \\ Z_k^\dagger & 0 \end{pmatrix} \right). \tag{62}$$

Since $\begin{pmatrix} Y_k^T \\ Q_k^T \end{pmatrix} (Y_k \quad Q_k), \begin{pmatrix} 0 & Z_k^\dagger \\ Z_k^\dagger & 0 \end{pmatrix} \in \mathbb{R}^{4\times 4}$, $\lambda_k$ can be computed cheaply. This cost is negligible compared with those to form $P_k^T P_k, Q_k^T Q_k$, which need $\mathcal{O}(d)$ flops. Then, to guarantee the positive definiteness of $H_k$, we check whether $\alpha_k$ satisfies (61) and use a smaller $\alpha_k$ if necessary, e.g. $\alpha_k = 2\beta_k(1-\mu)/\lambda_k$.

#### C.3.2 PROOFS OF THE THEOREMS

We first give the proof of the boundedness of $\|P_{k-1}\zeta_k\|_2$ and $\|Q_{k-1}\zeta_k\|_2$.

**Lemma 3.** *For $P, Q \in \mathbb{R}^{d\times m}(d \geq m), \delta > 0$, and $Z = Q^T Q + \delta P^T P$, we have*

$$\|PZ^\dagger Q^T\|_2 \leq \delta^{-\frac{1}{2}}, \tag{63a}$$

$$\|QZ^\dagger Q^T\|_2 \leq 1. \tag{63b}$$

*Proof.* We first consider the case that $Z$ is nonsingular, then $PZ^\dagger Q^T = PZ^{-1}Q^T$, $QZ^\dagger Q^T = QZ^{-1}Q^T$. It can be seen that $P^T P$ and $Q^T Q$ are symmetric positive semidefinite, and $Z$ is SPD as it is assumed to be nonsingular. Also, we have $\delta P^T P \preceq Z$ and $Q^T Q \preceq Z$, where the notation "$\preceq$"

denotes the *Loewner partial order*, i.e., $A \preceq B$ with $A, B \in \mathbb{R}^{m \times m}$ means that $B - A$ is positive semidefinite. Hence, we have

$$Z^{-\frac{1}{2}} \delta P^{\mathrm{T}} P Z^{-\frac{1}{2}} \preceq I, Z^{-\frac{1}{2}} Q^{\mathrm{T}} Q Z^{-\frac{1}{2}} \preceq I,$$

which implies

$$\|Z^{-\frac{1}{2}} \left(P^{\mathrm{T}} P\right) Z^{-\frac{1}{2}}\|_2 \leq \delta^{-1},$$
$$\|Z^{-\frac{1}{2}} \left(Q^{\mathrm{T}} Q\right) Z^{-\frac{1}{2}}\|_2 \leq 1.$$

Let $\lambda_{max}(\cdot)$ denote the largest eigenvalue, we have

$$\|Q Z^{-1} Q^{\mathrm{T}}\|_2 = \lambda_{max}\left(Q Z^{-1} Q^{\mathrm{T}}\right) = \lambda_{max}\left(Q^{\mathrm{T}} Q Z^{-1}\right) = \lambda_{max}\left(Z^{-\frac{1}{2}} Q^{\mathrm{T}} Q Z^{-\frac{1}{2}}\right) \leq 1,$$

$$\begin{aligned}
\|P Z^{-1} Q^{\mathrm{T}}\|_2^2 &= \lambda_{max}\left(P Z^{-1} Q^{\mathrm{T}} Q Z^{-1} P^{\mathrm{T}}\right) \\
&= \lambda_{max}\left(P^{\mathrm{T}} P Z^{-1} Q^{\mathrm{T}} Q Z^{-1}\right) \\
&= \lambda_{max}\left(Z^{-\frac{1}{2}} \left(P^{\mathrm{T}} P\right) Z^{-\frac{1}{2}} \cdot Z^{-\frac{1}{2}} \left(Q^{\mathrm{T}} Q\right) Z^{-\frac{1}{2}}\right) \\
&\leq \|Z^{-\frac{1}{2}} \left(P^{\mathrm{T}} P\right) Z^{-\frac{1}{2}} \cdot Z^{-\frac{1}{2}} \left(Q^{\mathrm{T}} Q\right) Z^{-\frac{1}{2}}\|_2 \\
&\leq \|Z^{-\frac{1}{2}} \left(P^{\mathrm{T}} P\right) Z^{-\frac{1}{2}}\|_2 \|Z^{-\frac{1}{2}} \left(Q^{\mathrm{T}} Q\right) Z^{-\frac{1}{2}}\|_2 \leq \delta^{-1}.
\end{aligned}$$

Therefore, $\|P Z^{-1} Q^{\mathrm{T}}\| \leq \delta^{-\frac{1}{2}}$ and $\|Q Z^{-1} Q^{\mathrm{T}}\|_2 \leq 1$.

For the case that $Z$ is singular, it can be proved that

$$\ker(Z) := \{x \in \mathbb{R}^m | Z x = 0\} = \{x \in \mathbb{R}^m | P x = 0 \text{ and } Q x = 0\}. \tag{64}$$

In fact, if $P x = 0$ and $Q x = 0$, it is obvious that $Z x = 0$. On the other side, if $Z x = 0$, then $x^{\mathrm{T}} Z x = 0$, i.e. $x^{\mathrm{T}} Q^{\mathrm{T}} Q x + \delta x^{\mathrm{T}} P^{\mathrm{T}} P x = 0$. Since $0 \preceq P^{\mathrm{T}} P, 0 \preceq Q^{\mathrm{T}} Q, \delta > 0$, it follows that $x^{\mathrm{T}} Q^{\mathrm{T}} Q x = 0$ and $\delta x^{\mathrm{T}} P^{\mathrm{T}} P x = 0$, which further implies $Q x = 0$ and $P x = 0$. Hence (64) holds.

Let $U_1$ satisfy $U_1^{\mathrm{T}} U_1 = I$ and $\text{range}(U_1) = \ker(Z)$, i.e. the orthonormal basis of $\ker(Z)$, and $U_2$ satisfy $U_2^{\mathrm{T}} U_2 = I$ and $U_2^{\mathrm{T}} U_1 = 0$, i.e. the orthonormal basis of $\ker(Z)^{\perp}$. With the equality (64), we know $P U_1 = 0, Q U_1 = 0$. Define $U = (U_1, U_2) \in \mathbb{R}^{m \times m}$, then $U^{\mathrm{T}} U = I_m$ and by direct computation, we have

$$U^{\mathrm{T}} Z U = \begin{pmatrix} 0 & 0 \\ 0 & U_2^{\mathrm{T}} Z U_2 \end{pmatrix},$$

where $U_2^{\mathrm{T}} Z U_2 = (Q U_2)^{\mathrm{T}} Q U_2 + \delta (P U_2)^{\mathrm{T}} P U_2$ is nonsingular according to the definition of $U_2$. So

$$Z^{\dagger} = U \begin{pmatrix} 0 & 0 \\ 0 & (U_2^{\mathrm{T}} Z U_2)^{-1} \end{pmatrix} U^{\mathrm{T}}.$$

As a result, we can further compute $P Z^{\dagger} Q^{\mathrm{T}}$ and $Q Z^{\dagger} Q^{\mathrm{T}}$ as

$$P Z^{\dagger} Q^{\mathrm{T}} = (P U_2)(U_2^{\mathrm{T}} Z U_2)^{-1}(Q U_2)^{\mathrm{T}}, \ Q Z^{\dagger} Q^{\mathrm{T}} = (Q U_2)(U_2^{\mathrm{T}} Z U_2)^{-1}(Q U_2)^{\mathrm{T}}.$$

Then let $\hat{P} = P U_2, \hat{Q} = Q U_2$, and $\hat{Z} = \hat{Q}^{\mathrm{T}} \hat{Q} + \delta \hat{P}^{\mathrm{T}} \hat{P}$, and noting that $\hat{Z}$ is nonsingular, we can obtain $\|P Z^{\dagger} Q^{\mathrm{T}}\|_2 = \|\hat{P} \hat{Z}^{-1} \hat{Q}^{\mathrm{T}}\|_2 \leq \delta^{-\frac{1}{2}}, \|Q Z^{\dagger} Q^{\mathrm{T}}\|_2 = \|\hat{Q} \hat{Z}^{-1} \hat{Q}^{\mathrm{T}}\|_2 \leq 1$. $\qquad \square$

As a result of Lemma 3, $\|P_{k-1} \zeta_k\|_2, \|Q_{k-1} \zeta_k\|_2$ in Algorithm 1 are bounded by $\mathcal{O}(\|\Delta r_{k-1}\|_2)$, as shown in the following corollary.

**Corollary 2.** $\|P_{k-1} \zeta_k\|_2, \|Q_{k-1} \zeta\|_2$ *in Algorithm 1 are bounded, i.e.*

$$\|P_{k-1} \zeta_k\|_2 \leq (\delta_k^{(1)})^{-\frac{1}{2}} \|\Delta r_{k-1}\|_2, \ \|Q_{k-1} \zeta_k\|_2 \leq \|\Delta r_{k-1}\|_2, \tag{65}$$

*where* $\zeta_k = (Q_{k-1}^{\mathrm{T}} Q_{k-1} + \delta_k^{(1)} P_{k-1} P_{k-1})^{\dagger} Q_{k-1}^{\mathrm{T}} \Delta r_{k-1}$.

Now, we turn to the proofs of the theorems about RST-AM in Section 3.4. For brevity, we use $\delta_k$ to denote $\delta_k^{(2)}$, i.e. $\delta_k \equiv \delta_k^{(2)}$. The proofs follow those of SAM (Wei et al., 2021). Nonetheless, since RST-AM uses different historical sequences compared with SAM, we give the detailed proofs for RST-AM for completeness.

From Assumption 2, for the mini-batch gradient $f_{S_k}(x_k) = \frac{1}{n_k} \sum_{i \in S_k} f_{\xi_i}(x_k)$, where $n_k = |S_k|$, the following properties hold:

$$\mathbb{E}[\nabla f_{S_k}(x)|x_k] = \nabla f(x_k), \tag{66a}$$

$$\mathbb{E}[\|\nabla f_{S_k}(x_k) - \nabla f(x_k)\|_2^2 |x_k] \leq \frac{\sigma^2}{n_k}. \tag{66b}$$

Consider the update of RST-AM. From Line 11-13 in Algorithm 1, it can be written as $x_{k+1} = x_k + H_k r_k$, where $r_k = -\nabla f_{S_k}(x_k)$, and for $k \geq 0$,

$$H_k = \beta_k I - \alpha_k \left(P_k + \beta_k Q_k\right) \left(Q_k^T Q_k + \delta_k P_k^T P_k\right)^\dagger Q_k^T. \tag{67}$$

To prove the theorems, the critical points are (i) the positive definiteness of the approximate Hessian $H_k$ and (ii) an adequate suppression of the noise from the gradient estimates in the stochastic case.

We first give a lemma related to the projection step.

**Lemma 4.** *Suppose that $\{x_k\}$ is generated by RST-AM. If $\alpha_k \geq 0, \beta_k > 0$, then for any $v_k \in \mathbb{R}^d$, we have*

$$\|H_k v_k\|_2^2 \leq 2 \left(\beta_k^2 \left(1 + 2\alpha_k^2 - 2\alpha_k\right) + \alpha_k^2 \delta_k^{-1}\right) \|v_k\|_2^2. \tag{68}$$

*Proof.* The result holds when $k = 0$ as $H_0 = \beta_0 I$. For $k \geq 1$,

$$H_k v_k = \beta_k v_k - (\alpha_k P_k + \alpha_k \beta_k Q_k)\Gamma_k, \tag{69}$$

where $\Gamma_k = (Q_k^T Q_k + \delta_k P_k^T P_k)^\dagger Q_k^T v_k$ solves

$$\min_\Gamma \|v_k - Q_k \Gamma\|_2^2 + \delta_k \|P_k \Gamma\|_2^2. \tag{70}$$

By direct computation, $\|v_k - Q_k \Gamma_k\|_2^2 + \delta_k \|P_k \Gamma_k\|_2^2 = \|v_k\|_2^2 - v_k^T Q_k (Q_k^T Q_k + \delta_k P_k^T P_k)^\dagger \cdot Q_k^T v_k$, thus

$$\|v_k - Q_k \Gamma_k\|_2^2 + \delta_k \|P_k \Gamma_k\|_2^2 \leq \|v_k\|_2^2. \tag{71}$$

Therefore,

$$\begin{aligned}
&\|H_k v_k\|_2^2 \\
&= \|\beta_k v_k - (\alpha_k P_k + \alpha_k \beta_k Q_k)\Gamma_k\|_2^2 \\
&= \|\beta_k \left(v_k - \alpha_k Q_k \Gamma_k\right) - \alpha_k P_k \Gamma_k\|_2^2 \\
&= \|\beta_k(1 - \alpha_k)v_k + \beta_k \alpha_k (v_k - Q_k \Gamma_k) - \alpha_k \delta_k^{-\frac{1}{2}} \delta_k^{\frac{1}{2}} P_k \Gamma_k\|_2^2 \\
&\leq \left(\beta_k^2(1-\alpha_k)^2 + \beta_k^2 \alpha_k^2 + \alpha_k^2 \delta_k^{-1}\right) \cdot \left(\|v_k\|_2^2 + \|v_k - Q_k \Gamma_k\|_2^2 + \delta_k \|P_k \Gamma_k\|_2^2\right) \\
&\leq \left(\beta_k^2 \left(1 + 2\alpha_k^2 - 2\alpha_k\right) + \alpha_k^2 \delta_k^{-1}\right) \left(\|v_k\|_2^2 + \|v_k\|_2^2\right) \\
&= 2 \left(\beta_k^2 \left(1 + 2\alpha_k^2 - 2\alpha_k\right) + \alpha_k^2 \delta_k^{-1}\right) \|v_k\|_2^2. \tag{72}
\end{aligned}$$

In the above, the first inequality uses the inequality

$$\|\sum_{i=1}^n a_i \mathbf{x_i}\|_2^2 \leq \left(\sum_{i=1}^n |a_i| \|\mathbf{x_i}\|_2\right)^2 \leq \left(\sum_{i=1}^n a_i^2\right) \left(\sum_{i=1}^n \|\mathbf{x_i}\|_2^2\right), \tag{73}$$

where $a_i \in \mathbb{R}, x_i \in \mathbb{R}^d$. The second inequality is based on inequality (71). $\square$

With Lemma 4, we can prove the deterministic case of RST-AM.

***Proof of Theorem 3.*** Since $1 + 2\alpha_k^2 - 2\alpha_k \le 1$ for $0 \le \alpha_k \le 1$, and $\delta_k^{-1} \le C^{-1}\beta^2$, with Lemma 4, we have

$$\|H_k r_k\|_2^2 \le 2\beta^2(1 + C^{-1})\|r_k\|_2^2.$$

Since $\alpha_k$ satisfies (61), we have

$$r_k^{\mathrm{T}} H_k r_k \ge \beta\mu\|r_k\|_2^2.$$

Then, under Assumption 1, we have

$$
\begin{aligned}
f(x_{k+1}) &\le f(x_k) + \nabla f(x_k)^{\mathrm{T}}(x_{k+1} - x_k) + \frac{L}{2}\|x_{k+1} - x_k\|_2^2 \\
&= f(x_k) - r_k^{\mathrm{T}} H_k r_k + \frac{L}{2}\|H_k r_k\|_2^2 \\
&\le f(x_k) - \beta\mu\|r_k\|_2^2 + L\beta^2(1 + C^{-1})\|r_k\|_2^2 \\
&= f(x_k) - \beta\left(\mu - \beta L(1 + C^{-1})\right)\|r_k\|_2^2 \\
&\le f(x_k) - \frac{1}{2}\beta\mu \cdot \|\nabla f(x_k)\|_2^2,
\end{aligned}
\tag{74}
$$

where the last inequality is due to $0 < \beta \le \frac{\mu}{2L(1+C^{-1})}$. Thus, $f(x_{k+1}) - f(x_k) \le -\frac{1}{2}\beta\mu\|\nabla f(x_k)\|_2^2$.

Summing both sides of this inequality for $k \in \{0, \dots, N-1\}$ and recalling $f(x) > f^{low}$ in Assumption 1 gives

$$f^{low} - f(x_0) \le f(x_N) - f(x_0) \le -\frac{1}{2}\beta\mu\sum_{k=0}^{N-1}\|\nabla f(x_k)\|_2^2.$$

Rearranging and dividing further by $N$ yields (15). $\qquad\square$

The next lemmas and proofs are about the stochastic case.

**Lemma 5.** *Suppose that Assumption 2 holds for $\{x_k\}$ generated by RST-AM. In addition, if $\beta_k > 0$, and $\alpha_k \ge 0$ and satisfies (13), then*

$$\mathbb{E}_{S_k}[\|H_k r_k\|_2^2] \le 2\left(\beta_k^2\left(1 + 2\alpha_k^2 - 2\alpha_k\right) + \frac{\alpha_k^2}{\delta_k}\right) \cdot \left(\|\nabla f(x_k)\|_2^2 + \frac{\sigma^2}{n_k}\right), \tag{75a}$$

$$\nabla f(x_k)^{\mathrm{T}}\mathbb{E}_{S_k}[H_k r_k] \le -\frac{1}{2}\beta_k\mu\|\nabla f(x_k)\|_2^2 + \frac{1}{2}\frac{\alpha_k^2(\delta_k^{-\frac{1}{2}} + \beta_k)^2}{\beta_k\mu} \cdot \frac{\sigma^2}{n_k}, \tag{75b}$$

*where $\mu > 0$ is the constant introduced in Remark 4.*

*Proof.* (i) From Lemma 4, we have

$$\mathbb{E}_{S_k}[\|H_k r_k\|_2^2] \le 2\left(\beta_k^2\left(1 + 2\alpha_k^2 - 2\alpha_k\right) + \frac{\alpha_k^2}{\delta_k}\right)\mathbb{E}_{S_k}[\|r_k\|_2^2]. \tag{76}$$

From Assumption 2, we have

$$\mathbb{E}_{S_k}[\|r_k\|_2^2] = \mathbb{E}_{S_k}[\|r_k - \mathbb{E}_{S_k}[r_k]\|_2^2] + \|\mathbb{E}_{S_k}[r_k]\|_2^2 \le \|\nabla f(x_k)\|_2^2 + \sigma^2/n_k. \tag{77}$$

With (76), (77), we obtain (75a).

(ii) The result holds for $k = 0$ since $H_0 = \beta_0 I$. Consider $k \ge 1$. Define $\epsilon_k = \nabla f_{S_k}(x_k) - \nabla f(x_k) = -r_k - \nabla f(x_k)$. Then $H_k r_k = H_k\left(-\epsilon_k - \nabla f(x_k)\right)$. Since $\alpha_k$ satisfies (13), it has $\lambda_{min}\left(\frac{1}{2}\left(H_k + H_k^{\mathrm{T}}\right)\right) \ge \beta_k\mu$. Thus

$$\nabla f(x_k)^{\mathrm{T}} H_k \nabla f(x_k) = \frac{1}{2}\nabla f(x_k)^{\mathrm{T}}\left(H_k + H_k^{\mathrm{T}}\right)\nabla f(x_k) \ge \beta_k\mu\|\nabla f(x_k)\|_2^2,$$

which implies

$$\mathbb{E}_{S_k}[\nabla f(x_k)^{\mathrm{T}} H_k \nabla f(x_k)] \ge \beta_k\mu\|\nabla f(x_k)\|_2^2. \tag{78}$$

Let $M_k = \alpha_k (P_k + \beta_k Q_k) (Q_k^T Q_k + \delta_k P_k^T P_k)^\dagger Q_k^T$, then $H_k = \beta_k I - M_k$. With the assumption (66a), i.e. $\mathbb{E}_{S_k}[\epsilon_k] = 0$, we have

$$\mathbb{E}_{S_k}[\nabla f(x_k)^T H_k \epsilon_k] = \mathbb{E}_{S_k}[\nabla f(x_k)^T (\beta_k \epsilon_k - M_k \epsilon_k)]$$
$$= \beta_k \nabla f(x_k)^T \mathbb{E}_{S_k}[\epsilon_k] - \mathbb{E}_{S_k}[\nabla f(x_k)^T M_k \epsilon_k] = -\mathbb{E}_{S_k}[\nabla f(x_k)^T M_k \epsilon_k].$$

Using the *Cauchy-Schwarz inequality with expectations*, we obtain

$$|\mathbb{E}_{S_k}[\nabla f(x_k)^T H_k \epsilon_k]| = |\mathbb{E}_{S_k}[\nabla f(x_k)^T M_k \epsilon_k]| \leq \sqrt{\mathbb{E}_{S_k}[\|\nabla f(x_k)\|_2^2]} \sqrt{\mathbb{E}_{S_k}[\|M_k \epsilon_k\|_2^2]}$$
$$\leq \|\nabla f(x_k)\|_2 \sqrt{\mathbb{E}_{S_k}[\|M_k \epsilon_k\|_2^2]}. \tag{79}$$

We now bound $\|M_k \epsilon_k\|_2^2$. For brevity, let $Z_k = Q_k^T Q_k + \delta_k P_k^T P_k$, and $N_1 = P_k Z_k^\dagger Q_k^T$, $N_2 = \beta_k Q_k Z_k^\dagger Q_k^T$, then

$$\|M_k\|_2 = \|\alpha_k (N_1 + N_2)\|_2 \leq \alpha_k (\|N_1\|_2 + \|N_2\|_2) \leq \alpha_k (\delta_k^{-\frac{1}{2}} + \beta_k), \tag{80}$$

where the last inequality is from Lemma 3.

With (80), we have $\|M_k \epsilon_k\|_2 \leq \alpha_k(\delta_k^{-\frac{1}{2}} + \beta_k)\|\epsilon_k\|_2$, which implies

$$\mathbb{E}_{S_k}[\|M_k \epsilon_k\|_2^2] \leq \alpha_k^2 (\delta_k^{-\frac{1}{2}} + \beta_k)^2 \mathbb{E}_{S_k}[\|\epsilon_k\|_2^2] \leq \alpha_k^2 (\delta_k^{-\frac{1}{2}} + \beta_k)^2 \frac{\sigma^2}{n_k}, \tag{81}$$

where the last inequality is due to (66b). Now we bound $|\mathbb{E}_{S_k}[\nabla f(x_k)^T H_k \epsilon_k]|$ as follows (cf. (79)):

$$|\mathbb{E}_{S_k}[\nabla f(x_k)^T H_k \epsilon_k]|$$
$$\leq \|\nabla f(x_k)\|_2 \sqrt{\mathbb{E}_{S_k}[\|M_k \epsilon_k\|_2^2]}$$
$$\leq \alpha_k (\delta_k^{-\frac{1}{2}} + \beta_k)\|\nabla f(x_k)\|_2 \sqrt{\mathbb{E}_{S_k}[\|\epsilon_k\|_2^2]}$$
$$\leq \alpha_k (\delta_k^{-\frac{1}{2}} + \beta_k)\frac{\sigma}{\sqrt{n_k}}\|\nabla f(x_k)\|_2$$
$$= \sqrt{\beta_k \mu}\|\nabla f(x_k)\|_2 \cdot \frac{\alpha_k(\delta_k^{-\frac{1}{2}} + \beta_k)}{\sqrt{\beta_k \mu}}\frac{\sigma}{\sqrt{n_k}}$$
$$\leq \frac{1}{2}\beta_k \mu \|\nabla f(x_k)\|_2^2 + \frac{1}{2}\frac{\alpha_k^2(\delta_k^{-\frac{1}{2}} + \beta_k)^2}{\beta_k \mu} \cdot \frac{\sigma^2}{n_k}. \tag{82}$$

With the inequality (78) and (82), we obtain

$$\nabla f(x_k)^T \mathbb{E}_{S_k}[H_k r_k]$$
$$= -\nabla f(x_k)^T \mathbb{E}_{S_k}[H_k (\epsilon_k + \nabla f(x_k))]$$
$$= -\mathbb{E}_{S_k}[\nabla f(x_k)^T H_k \nabla f(x_k)] - \mathbb{E}_{S_k}[\nabla f(x_k)^T H_k \epsilon_k]$$
$$\leq -\mathbb{E}_{S_k}[\nabla f(x_k)^T H_k \nabla f(x_k)] + |\mathbb{E}_{S_k}[\nabla f(x_k)^T H_k \epsilon_k]|$$
$$\leq -\beta_k \mu \|\nabla f(x_k)\|_2^2 + \frac{1}{2}\beta_k \mu \|\nabla f(x_k)\|_2^2 + \frac{1}{2}\frac{\alpha_k^2(\delta_k^{-\frac{1}{2}} + \beta_k)^2}{\beta_k \mu} \cdot \frac{\sigma^2}{n_k}$$
$$= -\frac{1}{2}\beta_k \mu \|\nabla f(x_k)\|_2^2 + \frac{1}{2}\frac{\alpha_k^2(\delta_k^{-\frac{1}{2}} + \beta_k)^2}{\beta_k \mu} \cdot \frac{\sigma^2}{n_k}. \tag{83}$$

$\square$

By imposing one more restriction on $\alpha_k$, we can obtain a convenient corollary:

**Corollary 3.** *Suppose that Assumption 2 holds for $\{x_k\}$ generated by RST-AM. $C > 0$ is the constant in (12). If $\beta_k > 0, 0 \leq \alpha_k \leq \min\{1, \beta_k^{\frac{1}{2}}\}$ and satisfies (61) , then*

$$\mathbb{E}_{S_k}[\|H_k r_k\|_2^2] \leq 2\beta_k^2 \left(1 + C^{-1}\right) \cdot \left(\|\nabla f(x_k)\|_2^2 + \frac{\sigma^2}{n_k}\right), \tag{84a}$$

$$\nabla f(x_k)^T \mathbb{E}_{S_k}[H_k r_k] \leq -\frac{1}{2}\beta_k \mu \|\nabla f(x_k)\|_2^2 + \beta_k^2 \cdot \mu^{-1} \left(1 + C^{-1}\right) \frac{\sigma^2}{n_k}. \tag{84b}$$

*Proof.* The first result (84a) is clear by considering (75a) and noticing that $1 + 2\alpha_k^2 - 2\alpha_k \le 1$ when $\alpha_k \in [0,1]$ and $\delta_k^{-1} \le C^{-1}\beta_k^2$. Since $\alpha_k \le \beta_k^{\frac{1}{2}}, \delta_k \ge C\beta_k^{-2}$ and $(1 + C^{-\frac{1}{2}})^2 \le 2(1 + C^{-1})$ we have

$$
\begin{aligned}
\frac{1}{2} \frac{\alpha_k^2 (\delta_k^{-\frac{1}{2}} + \beta_k)^2}{\beta_k \mu} \cdot \frac{\sigma^2}{n_k} &\le \frac{1}{2} \frac{\beta_k (C^{-\frac{1}{2}}\beta_k + \beta_k)^2}{\beta_k \mu} \cdot \frac{\sigma^2}{n_k} \\
&= \frac{1}{2}\mu^{-1}(C^{-\frac{1}{2}} + 1)^2 \beta_k^2 \cdot \frac{\sigma^2}{n_k} \\
&\le \beta_k^2 \mu^{-1}(1 + C^{-1})\frac{\sigma^2}{n_k}.
\end{aligned}
$$

Substituting it into (75b), we obtain (84b). □

With Corollary 3, we establish the descent property of RST-AM:

**Lemma 6.** *Suppose that Assumptions 1 and 2 hold for $\{x_k\}$ generated by RST-AM. $C > 0$ is the constant in (12). If $0 < \beta_k \le \frac{\mu}{4L(1+C^{-1})}, 0 \le \alpha_k \le \min\{1, \beta_k^{\frac{1}{2}}\}$ and satisfies (61), then*

$$
\mathbb{E}_{S_k}[f(x_{k+1})] \le f(x_k) - \frac{1}{4}\beta_k \mu \|\nabla f(x_k)\|_2^2 + \beta_k^2 \left( (L + \mu^{-1})(1 + C^{-1}) \right) \frac{\sigma^2}{n_k}. \tag{85}
$$

*Proof.* According to Assumption 1, we have

$$
\begin{aligned}
f(x_{k+1}) &\le f(x_k) + \nabla f(x_k)^{\mathrm{T}}(x_{k+1} - x_k) + \frac{L}{2}\|x_{k+1} - x_k\|_2^2 \\
&= f(x_k) + \nabla f(x_k)^{\mathrm{T}} H_k r_k + \frac{L}{2}\|H_k r_k\|_2^2. \tag{86}
\end{aligned}
$$

Taking expectation with respect to the mini-batch $S_k$ on both sides of (86) and using Corollary 3 we obtain

$$
\begin{aligned}
&\mathbb{E}_{S_k}[f(x_{k+1})] \\
&\le f(x_k) + \nabla f(x_k)^{\mathrm{T}}\mathbb{E}_{S_k}[H_k r_k] + \frac{L}{2}\mathbb{E}_{S_k}\|H_k r_k\|_2^2 \\
&\le f(x_k) - \frac{1}{2}\beta_k \mu \|\nabla f(x_k)\|_2^2 + \beta_k^2 \mu^{-1}(1 + C^{-1})\frac{\sigma^2}{n_k} + L\beta_k^2(1 + C^{-1})\left(\|\nabla f(x_k)\|_2^2 + \frac{\sigma^2}{n_k}\right) \\
&= f(x_k) - \beta_k \left(\frac{1}{2}\mu - \beta_k L(1 + C^{-1})\right)\|\nabla f(x_k)\|_2^2 + \beta_k^2(L + \mu^{-1})(1 + C^{-1})\frac{\sigma^2}{n_k}. \tag{87}
\end{aligned}
$$

Then (87) combined with the assumption $\beta_k \le \frac{\mu}{4L(1+C^{-1})}$ implies (85). □

Lemma 6 suggests that the term related to the noise from gradient estimates is bounded as a second-order term (i.e. $\mathcal{O}(\beta_k^2)$). Thus, with the diminishing stepsize, the effect of noise also diminishes. To establish the global convergence, we introduce the definition of a *supermartingale* following the proofs in (Wang et al., 2017; Wei et al., 2021).

**Definition 1.** *Let $\{\mathcal{F}_k\}$ be an increasing sequence of $\sigma$-algebras. If $\{X_k\}$ is a stochastic process satisfying (i) $\mathbb{E}[|X_k|] < \infty$, (ii) $X_k \in \mathcal{F}_k$ for all k, and (iii) $\mathbb{E}[X_{k+1}|\mathcal{F}_k] \le X_k$ for all k, then $\{X_k\}$ is called a supermartingale.*

**Proposition 3** (Supermartingale convergence theorem, see, e.g., Theorem 4.2.12 in (Durrett, 2019)). *If $\{X_k\}$ is a nonnegative supermartingale, then $\lim_{k\to\infty} X_k \to X$ almost surely and $\mathbb{E}[X] \le \mathbb{E}[X_0]$.*

Now, we prove the convergence theory of RST-AM in the nonconvex stochastic case.

***Proof of Theorem 4.*** (i) Define $\phi_k := \frac{\beta_k \mu}{4}\|\nabla f(x_k)\|_2^2$ and $\tilde{L} := (L + \mu^{-1})(1 + C^{-1})$, $\gamma_k := f(x_k) + \tilde{L}\frac{\sigma^2}{n}\sum_{i=k}^{\infty}\beta_i^2$. Let $\mathcal{F}_k$ be the $\sigma$-algebra measuring $\phi_k, \gamma_k$, and $x_k$. From (85) we know that

for any $k$,

$$\mathbb{E}[\gamma_{k+1}|\mathcal{F}_k] = \mathbb{E}[f(x_{k+1})|\mathcal{F}_k] + \tilde{L}\frac{\sigma^2}{n}\sum_{i=k+1}^{\infty}\beta_i^2$$

$$\leq f(x_k) - \frac{1}{4}\beta_k\mu\|\nabla f(x_k)\|_2^2 + \tilde{L}\frac{\sigma^2}{n}\sum_{i=k}^{\infty}\beta_i^2 = \gamma_k - \phi_k, \tag{88}$$

which implies that $\mathbb{E}[\gamma_{k+1} - f^{low}|\mathcal{F}_k] \leq \gamma_k - f^{low} - \phi_k$. Since $\phi_k \geq 0$, we have $0 \leq \mathbb{E}[\gamma_k - f^{low}] \leq \gamma_0 - f^{low} < +\infty$. As the diminishing condition (14) holds, we obtain $\mathbb{E}[f(x_k)] \leq M_f$ for some constant $M_f > 0$. According to Definition 1, $\{\gamma_k - f^{low}\}$ is a supermartingale. Therefore, Proposition 3 indicates that there exists a $\gamma$ such that $\lim_{k\to\infty}\gamma_k = \gamma$ with probability 1, and $\mathbb{E}[\gamma] \leq \mathbb{E}[\gamma_0]$. Note that from (88) we have $\mathbb{E}[\phi_k] \leq \mathbb{E}[\gamma_k] - \mathbb{E}[\gamma_{k+1}]$. Thus,

$$\mathbb{E}\left[\sum_{k=0}^{\infty}\phi_k\right] \leq \sum_{k=0}^{\infty}(\mathbb{E}[\gamma_k] - \mathbb{E}[\gamma_{k+1}]) < +\infty,$$

which further yields that

$$\sum_{k=0}^{\infty}\phi_k = \frac{\mu}{4}\sum_{k=0}^{\infty}\beta_k\|\nabla f(x_k)\|_2^2 < +\infty \text{ with probability 1.} \tag{89}$$

Since $\sum_{k=0}^{\infty}\beta_k = +\infty$, it follows that (16) holds.

(ii) If the noisy gradient is bounded, i.e.,

$$\mathbb{E}_{\xi_k}[\|\nabla f_{\xi_k}(x_k)\|_2^2] \leq M_g, \tag{90}$$

where $M_g > 0$ is a constant, then a stronger result can be obtained.

For any give $\epsilon > 0$, according to (16), there exist infinitely many iterates $x_k$ such that $\|\nabla f(x_k)\|_2 \leq \epsilon$. Then if (17) does not hold, there must exist two infinite sequences of indices $\{s_i\}$, $\{t_i\}$ with $t_i > s_i$, such that for $i = 0, 1, \ldots, k = s_i + 1, \ldots, t_i - 1$,

$$\|\nabla f(x_{s_i})\|_2 \geq 2\epsilon, \|\nabla f(x_{t_i})\|_2 < \epsilon, \|\nabla f(x_k)\|_2 \geq \epsilon. \tag{91}$$

Then from (89) it follows that

$$+\infty > \sum_{k=0}^{\infty}\beta_k\|\nabla f(x_k)\|_2^2 \geq \sum_{i=0}^{+\infty}\sum_{k=s_i}^{t_i-1}\beta_k\|\nabla f(x_k)\|_2^2 \geq \epsilon^2\sum_{i=0}^{+\infty}\sum_{k=s_i}^{t_i-1}\beta_k \text{ with probability 1,}$$

which implies that

$$\sum_{k=s_i}^{t_i-1}\beta_k \to 0 \text{ with probability 1, as } i \to +\infty. \tag{92}$$

According to (77) and (72), we have

$$\mathbb{E}[\|x_{k+1} - x_k\|_2|x_k]$$
$$= \mathbb{E}[\|H_k r_k\|_2|x_k]$$
$$\leq \sqrt{2\left(\beta_k^2(1 + 2\alpha_k^2 - 2\alpha_k) + \alpha_k^2\delta_k^{-1}\right)}\mathbb{E}[\|r_k\|_2|x_k]$$
$$\leq \beta_k\sqrt{2(1 + C^{-1})}\mathbb{E}[\|r_k\|_2|x_k]$$
$$\leq \beta_k\sqrt{2(1 + C^{-1})}(\mathbb{E}[\|r_k\|_2^2|x_k])^{\frac{1}{2}}$$
$$\leq \beta_k\sqrt{2(1 + C^{-1})}M_g^{\frac{1}{2}}, \tag{93}$$

where the last inequalities are due to *Cauchy-Schwarz inequality* and (90). Then it follows from (93) that

$$\mathbb{E}[\|x_{t_i} - x_{s_i}\|_2] \leq \sqrt{2(1 + C^{-1})}M_g^{\frac{1}{2}}\sum_{k=s_i}^{t_i-1}\beta_k,$$

which together with (92) implies that $\|x_{t_i} - x_{s_i}\|_2 \to 0$ with probability 1, as $i \to +\infty$. Hence, from the Lipschitz continuity of $\nabla f$, it follows that $\|\nabla f(x_{t_i}) - \nabla f(x_{s_i})\|_2 \to 0$ with probability 1 as $i \to +\infty$. However, this contradicts (91). Therefore, the assumption that (17) does not hold is not true. □

***Proof of Theorem 5***. According to (87) in Lemma 6, we have

$$\sum_{k=0}^{N-1} \beta_k \left(\frac{1}{2}\mu - \beta_k L(1 + C^{-1})\right) \mathbb{E}\|\nabla f(x_k)\|_2^2$$

$$\leq f(x_0) - f^{low} + \sum_{k=0}^{N-1} \beta_k^2 (L + \mu^{-1})(1 + C^{-1})\frac{\sigma^2}{n_k}, \tag{94}$$

where the expectation is taken with respect to $\{S_j\}_{j=0}^{N-1}$. Define

$$P_R(k) \stackrel{\text{def}}{=} Prob\{R = k\} = \frac{\beta_k \left(\frac{1}{2}\mu - \beta_k L(1 + C^{-1})\right)}{\sum_{j=0}^{N-1} \beta_j \left(\frac{1}{2}\mu - \beta_j L(1 + C^{-1})\right)}, \quad k = 0, \ldots, N-1, \tag{95}$$

then

$$\mathbb{E}\left[\|\nabla f(x_R)\|_2^2\right] = \frac{\sum_{k=0}^{N-1} \beta_k \left(\frac{1}{2}\mu - \beta_k L(1 + C^{-1})\right) \mathbb{E}\left[\|\nabla f(x_k)\|_2^2\right]}{\sum_{j=0}^{N-1} \beta_j \left(\frac{1}{2}\mu - \beta_j L(1 + C^{-1})\right)}$$

$$\leq \frac{D_f + \sigma^2 (L + \mu^{-1})(1 + C^{-1}) \sum_{k=0}^{N-1} \beta_k^2/n_k}{\sum_{j=0}^{N-1} \beta_j \left(\frac{1}{2}\mu - \beta_j L(1 + C^{-1})\right)}. \tag{96}$$

Let $\tilde{D}$ be a problem-independent constant. If we choose $\beta_k = \beta := \min\{\frac{\mu}{4L(1+C^{-1})}, \frac{\tilde{D}}{\sigma\sqrt{N}}\}$, and $n_k = n$, then the definition of $P_R$ simplifies to $P_R(k) = 1/N$. From (96) we have

$$\mathbb{E}[\|\nabla f(x_R)\|_2^2] \leq \frac{D_f + \sigma^2(L + \mu^{-1})(1 + C^{-1})\frac{N\beta^2}{n}}{\sum_{j=0}^{N-1} \beta(\frac{1}{2}\mu - \frac{\mu}{4})}$$

$$= \frac{D_f + \sigma^2(L + \mu^{-1})(1 + C^{-1}) \cdot \frac{N\beta^2}{n}}{N\beta \cdot \frac{1}{4}\mu}$$

$$= \frac{4D_f}{N\beta\mu} + \frac{\sigma^2(L + \mu^{-1})(1 + C^{-1}) \cdot \beta}{\frac{1}{4}n\mu}$$

$$\leq \frac{4D_f}{N\mu} \max\left\{\frac{4L(1 + C^{-1})}{\mu}, \frac{\sigma\sqrt{N}}{\tilde{D}}\right\} + \frac{4\sigma^2(L + \mu^{-1})(1 + C^{-1})}{n\mu} \cdot \frac{\tilde{D}}{\sigma\sqrt{N}}$$

$$\leq \frac{4D_f}{N\mu}\left(\frac{4L(1 + C^{-1})}{\mu} + \frac{\sigma\sqrt{N}}{\tilde{D}}\right) + \frac{4\sigma(L + \mu^{-1})(1 + C^{-1})\tilde{D}}{n\mu\sqrt{N}}$$

$$= \frac{16D_f L(1 + C^{-1})}{N\mu^2} + \frac{\sigma}{\mu\sqrt{N}}\left(\frac{4D_f}{\tilde{D}} + \frac{4(L + \mu^{-1})(1 + C^{-1})\tilde{D}}{n}\right).$$

Therefore, to ensure $\mathbb{E}[\|\nabla f(x_R)\|_2^2] \leq \epsilon$, the number of iterations is $\mathcal{O}(1/\epsilon^2)$. $\qquad\square$

# D EXPERIMENTAL DETAILS

Our main codes were written based on the PyTorch framework [1] and one GeForce RTX 2080 Ti GPU was used for the tests in training neural networks. Our methods are ST-AM (MST-AM) and the regularized version RST-AM.

## D.1 EXPERIMENTAL DETAILS ABOUT ST-AM/MST-AM

The experiments about ST-AM were conducted to verify the main theorems, i.e. Theorem 1, Corollary 1 and Theorem 2. Four types of problems were used for the experiments. The conjugate residual (CR) method is a short-term recurrence version of the full-memory GMRES and needs matrix-vector products to fulfill the algorithm. We give the pseudocode of the CR method (Algorithm 6.20 in (Saad, 2003)) here for readers who are not familiar with this numerical algorithm.

---

[1] Information about this framework can be found in https://pytorch.org.

---

**Algorithm 4** CR Algorithm for solving a SPD linear system $Ax = b$.

---

**Input**: $x_0 \in \mathbb{R}^d$.
**Output**: $x \in \mathbb{R}^d$
  1:  $r_0 = b - Ax_0, p_0 = r_0$
  2: **for** $k = 0, 1, \dots,$ until convergence, **do**
  3:    $\alpha_k = \frac{r_k^{\mathrm{T}} A r_k}{(Ap_k)^{\mathrm{T}} Ap_k}$
  4:    $x_{k+1} = x_k + \alpha_k p_k$
  5:    $r_{k+1} = r_k - \alpha_k Ap_k$
  6:    $\beta_k = \frac{r_{k+1}^{\mathrm{T}} A r_{k+1}}{r_k^{\mathrm{T}} A r_k}$
  7:    $p_{k+1} = r_{k+1} + \beta_k p_k$
  8:    $Ap_{k+1} = A r_{k+1} + \beta_k Ap_k$
  9: **end for**
10: **return** $x_k$

---

According to Theorem 1, the CR method is essentially equivalent to ST-AM for solving strongly convex quadratic optimization. However, it should be pointed out that the CR method needs to directly assess the matrix $A$, and the residual update (Line 5) is based on the linear assumption, which makes it inapplicable for nonlinear problems or the case that $A$ is unavailable. Also, even though finite difference technique can be used to construct the matrix-vector products, the number of gradient evaluations is twice of that of ST-AM.

### D.1.1 STRONGLY CONVEX QUADRATIC OPTIMIZATION

To construct a case of (5), we considered the least squares problem:

$$\min_{x \in \mathbb{R}^d} f(x) := \frac{1}{2} \|Ax - b\|_2^2, \tag{97}$$

where $A \in \mathbb{R}^{\ell \times d}, b \in \mathbb{R}^{\ell}$, which can be reformulated as a form of (5).

We generated a dense random matrix $A \in \mathbb{R}^{500 \times 100}$ and a dense random vector $b \in \mathbb{R}^{500}$ for the test. The gradient descent used a fixed stepsize $\eta \leq \frac{1}{\|A\|_2^2}$ that can guarantee convergence, and the same $\eta$ was used as the $\beta_k$ for the full-memory AM and ST-AM.

We also conducted additional tests about solving the problem (97) with different condition numbers, where the eigenvalues of $A$ were set to be uniformly distributed. The results with different condition numbers ($\mathrm{cond}(A^{\mathrm{T}}A) = 10^2, 10^4, 10^6$) are shown in Figure 4. It can be observed that the curves of CR, AM and ST-AM nearly coincide, which verify the correctness of Theorem 1. Also, since the eigenvalues are uniformly distributed, the superlinear convergence behaviour may not happen. Nonetheless, CR, AM and ST-AM are still much faster then the GD method.

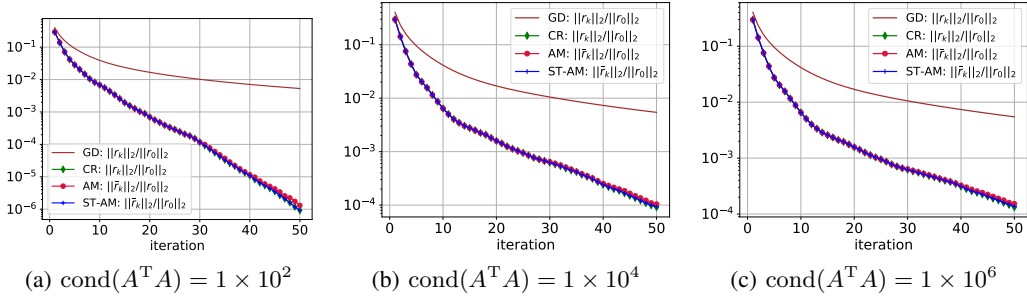

(a) $\mathrm{cond}(A^{\mathrm{T}}A) = 1 \times 10^2$     (b) $\mathrm{cond}(A^{\mathrm{T}}A) = 1 \times 10^4$     (c) $\mathrm{cond}(A^{\mathrm{T}}A) = 1 \times 10^6$

Figure 4: Solving (97) with different condition numbers: $\mathrm{cond}(A^{\mathrm{T}}A) = \lambda_{max}(A^{\mathrm{T}}A)/\lambda_{min}(A^{\mathrm{T}}A)$.

### D.1.2 SOLVING A NONSYMMETRIC LINEAR SYSTEM

For the solution of a nonsymmetric linear system

$$Ax = b, \tag{98}$$

where $A \in \mathbb{R}^{d \times d}, b \in \mathbb{R}^d$, the fixed-point iteration (FP) is $x_{k+1} = g(x_k) := x_k + \eta(b - Ax_k)$. Theorem 2 requires $g$ to be a contractive map, i.e. $\|I - \eta A\|_2 < 1$. We used a test matrix "fidap029" from the Matrix Market [2]. This matrix is a banded matrix and not symmetric. We used Jacobi preconditioner for all the tested iterative methods. Since solving linear systems is not the main focus of this work, we did not do a thorough test of applying ST-AM to solve various nonsymmetric linear systems.

### D.1.3 CUBIC-REGULARIZED QUADRATIC MINIMIZATION

The concerned cubic-regularized quadratic minimization is

$$\min_{x \in \mathbb{R}^d} f(x) := \frac{1}{2}\|Ax - b\|_2^2 + \frac{M}{3}\|x\|_2^3, \tag{99}$$

where $A \in \mathbb{R}^{\ell \times d}, b \in \mathbb{R}^\ell$, and $M \geq 0$ is the regularization parameter. It can be computed that

$$\nabla f(x) = A^{\mathrm{T}}(Ax - b) + M\|x\|_2 x = (A^{\mathrm{T}}A + M\|x\|_2 I)x - A^{\mathrm{T}}b.$$

Hence, for the gradient descent $x_{k+1} = g(x_k) := x_k - \eta \nabla f(x_k)$, the Jacobian of $g$ is $I - \eta A^{\mathrm{T}}A - \eta M(\|x\|_2^{-1}xx^{\mathrm{T}} + \|x\|_2 I)$, which has $\hat{\kappa} > 0$ in the local region $\mathcal{B}(\rho)$.

We generated a dense random matrix $A \in \mathbb{R}^{500 \times 100}$ and a dense vector $b \in \mathbb{R}^{500}$ for the test. The official implementation of L-BFGS in PyTorch was used for comparison. The historical length $m$ of L-BFGS was 50, i.e., BFGS was actually used in the test.

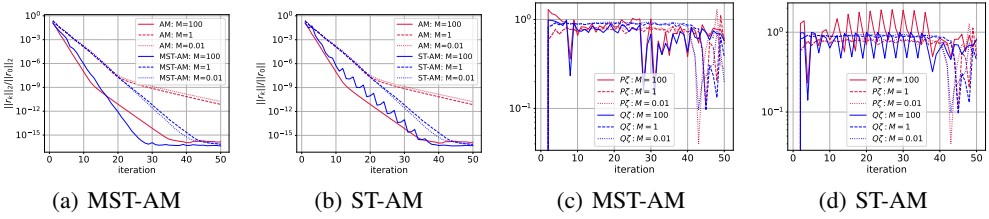

(a) MST-AM      (b) ST-AM      (c) MST-AM      (d) ST-AM

Figure 5: Solving (99) with different $M$. (a) $\|r_k\|_2/\|r_0\|_2$ of MST-AM; (b) $\|r_k\|_2/\|r_0\|_2$ of ST-AM; (c) $\|P_{k-1}\zeta_k\|_2/\|\Delta x_{k-1}\|_2$ and $\|Q_{k-1}\zeta_k\|_2/\|\Delta r_{k-1}\|_2$ of MST-AM; (d) $\|P_{k-1}\zeta_k\|_2/\|\Delta x_{k-1}\|_2$ and $\|Q_{k-1}\zeta_k\|_2/\|\Delta r_{k-1}\|_2$ of ST-AM.

We also conducted the tests related to different regularization parameters $M$, and the cases that $M = 0.01, 1, 100$ are shown in Figure 5. Figure 5 shows that with the large $M = 100$, both AM and MST-AM converge faster. An ablation study was also conducted about the boundedness restriction of $\|P_{k-1}\zeta_k\|_2, \|Q_{k-1}\zeta_k\|_2$ in Theorem 2. In Figure 5(b), we show the convergence behaviour of ST-AM without the boundedness check. In the case of the rather large regularization $M = 100$, ST-AM does not show a monotone decrease of the residual. To further investigate the cause, we plot the magnitude of $\|P_{k-1}\zeta_k\|_2/\|\Delta x_{k-1}\|_2, \|Q_{k-1}\zeta_k\|_2/\|\Delta r_{k-1}\|_2$ in Figure5(c) and Figure 5(d). It can be observed that the evolutions of $\|P_{k-1}\zeta_k\|_2/\|\Delta x_{k-1}\|_2, \|Q_{k-1}\zeta_k\|_2/\|\Delta r_{k-1}\|_2$ are quite oscillatory in ST-AM, while being roughly bounded below 1 in MST-AM. This phenomenon may accounts for the more stable convergence behaviour of MST-AM and verifies the necessity of the changes of MST-AM compared with ST-AM. In our experiments, we also found the restarting period can be set quite large and has little effect on ST-AM.

---

[2] https://math.nist.gov/MatrixMarket/.

### D.1.4 ROOT-FINDING PROBLEMS IN THE MULTISCALE DEEP EQUILIBRIUM MODEL

The multiscale deep equilibrium (MDEQ) model is a recent extension of the deep equilibrium (DEQ) model (Bai et al., 2019) for computer vision. One of the central engines for these DEQ models is the root-finding problem:

$$f_\theta(z; x) = g_\theta(z; x) - z \Rightarrow \text{find } z^* \text{ s.t. } f_\theta(z^*; x) = 0, \tag{100}$$

where $\theta$ and $x$ are parameters and the input representation, respectively. In (Bai et al., 2020), the Broyden's method is employed to solve this problem. Since MST-AM is suitable for solving non-linear systems of equations, we can apply MST-AM to solve (100).

We implemented the MST-AM method and integrated it into the MDEQ framework [3] . The task was image classification on CIFAR-10 and we used the small model for test. We followed the suggested experimental setting of the framework. The optimizer was Adam with learning rate of 0.001 and the weight-decay was $2.5 \times 10^{-6}$. The batch size was 128 and the number of epochs was 50. The tested fixed-point solver was used for the forward root-finding process and for the backward root-finding process. The threshold of the steps for the forward process was 18 and the threshold of the steps for the backward process was 20.

We used the built-in AM method and Broyden's method as the baseline methods. The $m$ for AM was 20, i.e. using the full-memory AM.

### D.2 EXPERIMENTAL DETAILS ABOUT RST-AM

Our experiments on RST-AM focused on training neural networks. Since ST-AM can be regarded as a special case of RST-AM with $\delta_k^{(1)} = \delta_k^{(2)} = 0$, the basic ST-AM is also covered. In the Line 10 in Algorithm 1, $\alpha_k$ should be adjusted to meet the positive definiteness check (13), which can be simplified to $(\Delta x_k)^T r_k \geq \beta_k \mu \|r_k\|_2^2$ in practice. The adjustment of $\alpha_k$ can be (i) $\alpha_k = \min\{\alpha_k, 2\beta_k(1-\mu)/\lambda_k\}$, or (ii) $\alpha_k = 0$. We used the option (ii) since the violation of (13) seldom happened in our tests and option (ii) is more simple to apply.

In the experiments on MNIST, Penn Treebank, we incorporated preconditioning (described in Appendix A.3) into the baseline method SAM and our method. Preconditioning is found to be effective for difficult problems, e.g. mini-batch training with very small batch sizes and the scaling of the model's parameters being important.

#### D.2.1 HYPERPARAMETER SETTING OF RST-AM

The hyperparameters of RST-AM are easy to tune. The only hyperparameters that need to be carefully tuned are the regularization parameters $c_1, c_2$ in $\delta_k^{(1)}$ and $\delta_k^{(2)}$. We found RST-AM is more sensitive to $c_1$, possibly due to the fact that it influences the construction of secant equations. $c_2$ can be quite small in our tests. The hyperparameter $C$ in $\delta_k^{(2)}$ was set to be very small ($C = 1 \times 10^{-16}$) so as to ensure $\delta_k^{(2)} = \frac{c_2 \|r_k\|_2^2}{\|p_k\|_2^2 + \epsilon_0}$ almost all the time. $\epsilon_0$ is introduced to prevent the denominators from being zero. We found $\|p_k\|_2 > 0$ and $\|\Delta x_{k-1}\|_2 > 0$ always held in the tests, so $\epsilon_0$ can be omitted. In the experiments except for deterministic optimization on MNIST, we kept the setting $c_1 = 1, c_2 = 1 \times 10^{-7}$ unchanged and found such setting is quite robust.

The other hyperparameters are $\alpha_k, \beta_k$, which can be always initially set as 1. So RST-AM has the same number of hyperparameters to tune as SGDM, since SGDM needs to tune learning rate and the momentum.

#### D.2.2 EXPERIMENTS ON MNIST

The experiments on MNIST [4] aimed to validate the effectiveness of RST-AM in deterministic optimization, so we were only concerned about the training loss by regarding it as a nonlinear function to be optimized. To facilitate the full-batch training, we used a subset of the training dataset by randomly selecting 10k images from the total 60k images.

---

[3] https://github.com/locuslab/mdeq.

[4] Based on the official PyTorch implementation https://github.com/pytorch/examples/blob/master/mnist.

The baselines were SGDM, Adam, Adagrad, RMSprop and SAM. We tried our best to ensure that the baselines had the best performance in the tests. We tuned the learning rates by log-scale grid-searches from $10^{-3}$ to 100. For SGDM, Adam, Adagrad, and RMSprop, the learning rates were 0.2, 0.001, 0.01, 0.001, respectively. For SAM, we used the hyperparameter setting recommended in (Wei et al., 2021).

For RST-AM, we set $c_1 = 0.05, c_2 = 1 \times 10^{-7}, \alpha_k = \beta_k = 1$. When $(\Delta x_k)^\mathrm{T} r_k \leq 0$ occurs, $x_{k+1} = x_k + 0.2 r_k$ was used as the new update.

For the preconditioned RST-AM, we set $c_1 = 1 \times 10^{-2}, c_2 = 1 \times 10^{-7}$ for the Adagrad-preconditioned RST-AM and $c_1 = 1 \times 10^{-2}, c_2 = 1 \times 10^{-8}$ for the RMSprop-preconditioned RST-AM.

For all these tests, we trained the model for 200 epochs.

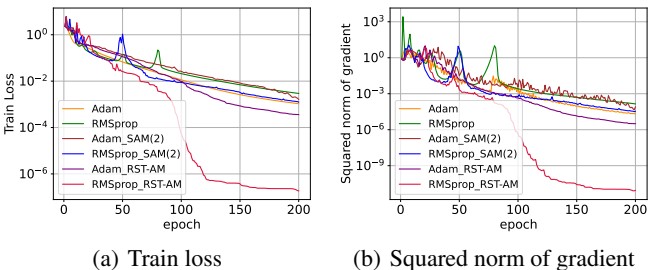

(a) Train loss      (b) Squared norm of gradient

Figure 6: Training on MNIST with the Adam/RMSprop preconditioner.

In Figure 2 in the main paper, we report the RMSprop/Adagrad preconditioned SAM/RST-AM. We give the result about using Adam as the preconditioner in Figure 6. It suggests that Adam does not perform as well as the RMSprop method to serve as a preconditioner for SAM/RST-AM in this task. Nevertheless, Adam_RST-AM is still better than Adam and Adam_SAM(2), which demonstrates the effect of our proposed short-term recurrence scheme.

### D.2.3 EXPERIMENTS ON CIFAR

This group of experiments were the same as those in (Wei et al., 2021) so that we can have a direct comparison between RST-AM and SAM. We still give the details here for completeness.

We followed the same way of training ResNet (He et al., 2016). The batchsize was set to be 128. For $N$ iterations of training, the learning rate of each optimizer was decayed at the $(\lfloor \frac{N}{2} \rfloor)$-th and the $(\lfloor \frac{3}{4} N \rfloor)$-th iterations. Here, for SAM and RST-AM, the $\alpha_k$ and $\beta_k$ serve as the learning rates, so the learning rate decay denotes decaying $\alpha_k, \beta_k$ simultaneously. The experiments were run with 3 random seeds.

The baseline optimizers were SGDM, Adam, AdaBound (Luo et al., 2018), AdaBelief (Zhuang et al., 2020), Lookahead (Zhang et al., 2019) and AdaHessian (Yao et al., 2021). Adam and the recently proposed optimizers AdaBound and AdaBelief are adaptive learning rate methods which use different learning rates for different model parameters. Lookahead is a $k$-step method and has an inner optimizer. In each cycle of Lookahead, the inner-optimizer $optim$ is used to iterate for $k$ steps and then the first iterate and the last iterate are interpolated to obtain the starting point of the next cycle. Compared to Adam, AdaHessian uses Hessian-vector products to construct a diagonal approximation of the Hessian.

For fair comparison, the hyperparameters of all the optimizers (including RST-AM) were tuned through experiments on CIFAR-10/ResNet20. For each optimizer, the hyperparameter setting that attained the highest final test accuracy on CIFAR-10/ResNet20 was kept unchanged for training the other networks on CIFAR. We found the results of hyperparameter tuning were consistent with those reported in (Wei et al., 2021). For completeness, we list the settings of hyperparameters here (learning rate is abbreviated as lr, and "*" indicates the same setting as that in (Wei et al., 2021)).

- **SGDM**\*: lr = 0.1, momentum = 0.9, weight-decay = $5 \times 10^{-4}$, lr-decay = 0.1.

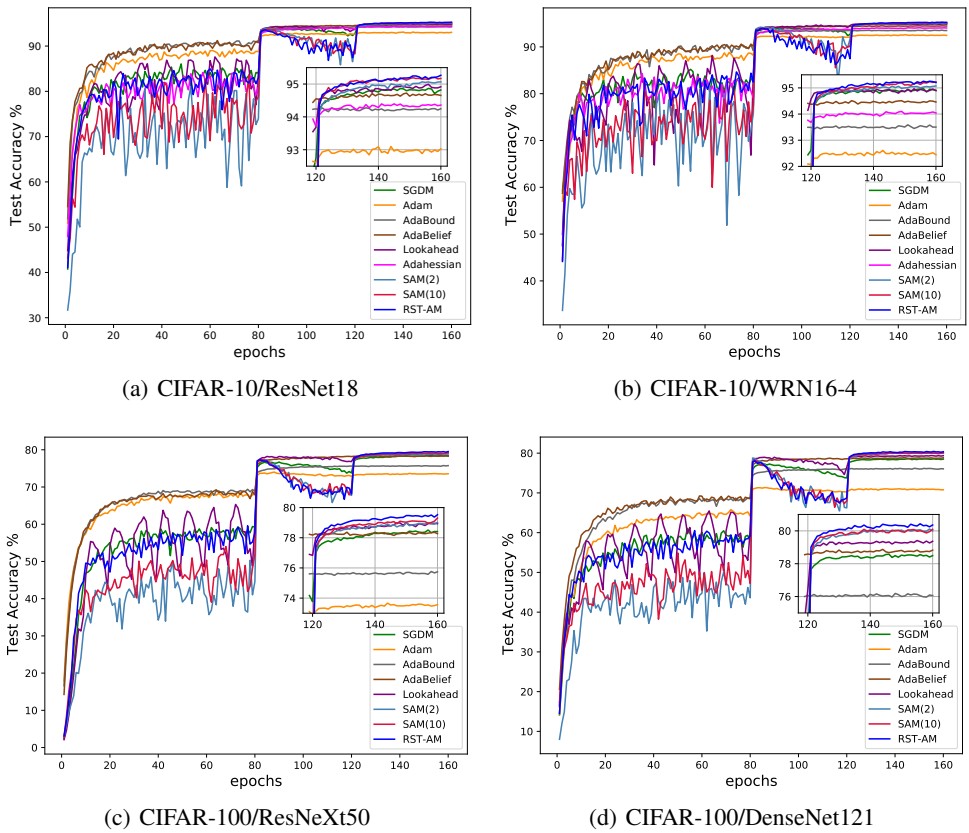

(a) CIFAR-10/ResNet18

(b) CIFAR-10/WRN16-4

(c) CIFAR-100/ResNeXt50

(d) CIFAR-100/DenseNet121

Figure 7: Test accuracy of training ResNet18 and WideResNet16-4 on CIFAR-10 and training ResNeXt50 and DenseNet121 on CIFAR-100.

- **Adam**[*]: lr = 0.001, $(\beta_1, \beta_2) = (0.9, 0.999)$, weight-decay = $5 \times 10^{-4}$, lr-decay = 0.1.
- **AdaBound**: lr = 0.001, $(\beta_1, \beta_2) = (0.9, 0.999)$, final_lr = 0.1, gamma = 0.001, weight-decay = $5 \times 10^{-4}$, lr-decay = 0.1.
- **AdaBelief**[*]: lr = 0.001, $(\beta_1, \beta_2) = (0.9, 0.999)$, eps = $1 \times 10^{-8}$, weight-decay = $5 \times 10^{-4}$, lr-decay = 0.1.
- **Lookahead**[*]: $optim$: SGDM (lr = 0.1, momentum = 0.9, weight-decay = $1 \times 10^{-3}$), $\alpha = 0.8$, steps = 10, lr-decay = 0.1.
- **AdaHessian**[*]: lr = 0.15, $(\beta_1, \beta_2) = (0.9, 0.999)$, eps=$1 \times 10^{-4}$, hessian-power: 1, weight-decay = $5 \times 10^{-4}/0.15$, lr-decay = 0.1.
- **SAM(2)**: $optim$: SGDM (lr = 0.1, momentum = 0, weight-decay = $1.5 \times 10^{-3}$), $\alpha_k = 1.0, \beta_k = 1.0, c_1 = 0.01, p = 1, m = 2$, weight-decay = $1.5 \times 10^{-3}$, lr-decay = 0.06.
- **SAM(10)**[*]: $optim$: SGDM (lr = 0.1, momentum = 0, weight-decay = $1.5 \times 10^{-3}$), $\alpha_k = 1.0, \beta_k = 1.0, c_1 = 0.01, p = 1, m = 10$, weight-decay = $1.5 \times 10^{-3}$, lr-decay = 0.06.
- **RST-AM**: $c_1 = 1, c_2 = 1 \times 10^{-7}, \alpha_0 = \beta_0 = 1$, weight-decay = $1 \times 10^{-3}$, lr-decay = 0.1.

For the tests of SGDM, Adam, AdaBelief, Lookahead, AdaHessian and SAM(10), we also had consistent numerical results with those reported in (Wei et al., 2021), so we reported their results of these methods in the main paper for reference.

Figure 7 shows the curves of test accuracy for training ResNet18 and WRN16-4 on CIFAR-10 and training ResNeXt50 and DenseNet121 on CIFAR-100. The numerical results of final test accuracy can be found in Table 1(a) in the main paper. It can be found in Figure 7 that the convergence behaviour of RST-AM is less erratic than SAM(10) and SAM(2). In the first 80 epochs, RST-AM

converges faster than SAM, partly due to using a smaller weight decay. The learning rate schedule has a large impact on the convergence of each optimizer. Similar to SAM(10), RST-AM can always climb up and stabilize to a higher test accuracy in the final 40 epochs. It is found that RST-AM is comparable to SAM(10), while improving the short-memory SAM(2).

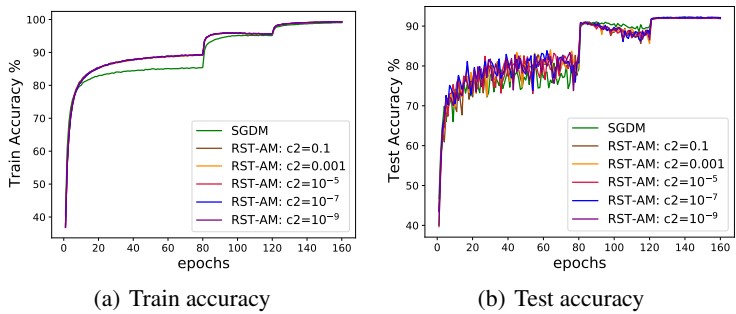

(a) Train accuracy        (b) Test accuracy

Figure 8: Train accuracy and test accuracy of RST-AM with different $c_2$.

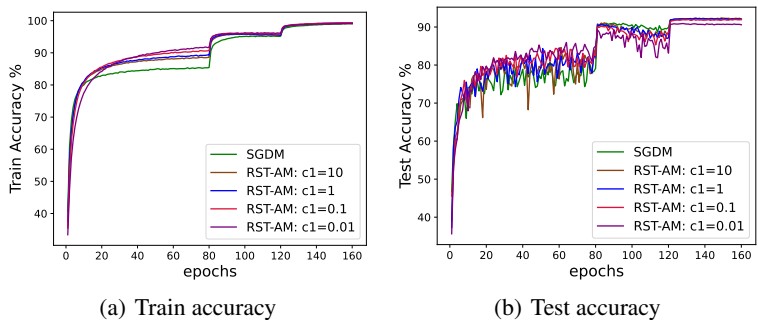

(a) Train accuracy        (b) Test accuracy

Figure 9: Train accuracy and test accuracy of RST-AM with different $c_1$.

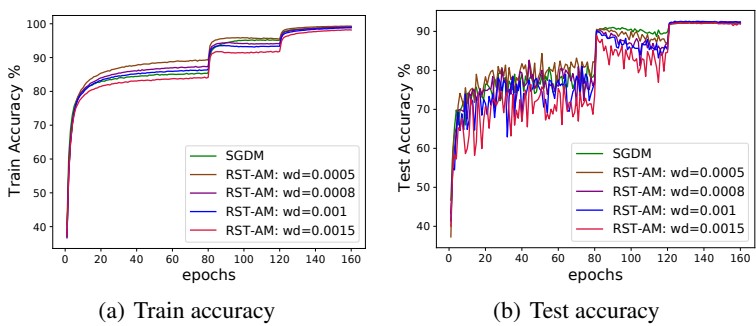

(a) Train accuracy        (b) Test accuracy

Figure 10: Train accuracy and test accuracy of RST-AM with different weight-decays.

Since our hyperparameter tuning was conducted on CIFAR10/ResNet20, we give some results about the hyperparameters on this model.

**Effect of $c_2$ in RST-AM**. Figure 8 shows the effect of $c_2$, where we kept $c_1 = 1$, weight-decay=$5 \times 10^{-4}$ fixed, and $c_2 = 10^{-1}, 10^{-3}, 10^{-5}, 10^{-7}, 10^{-9}$. It indicates that RST-AM is not sensitive to $c_2$.

**Effect of $c_1$ in RST-AM**. Figure 9 shows the effect of $c_1$, where we kept $c_2 = 10^{-7}$, weight-decay=$5 \times 10^{-4}$ fixed, and $c_1 = 0.01, 0.1, 1, 10$. It suggests that with smaller $c_1$, RST-AM converges faster in terms of train accuracy, but the test accuracy is worse.

**Effect of weight decay in RST-AM**. Weight-decay is a common hyperparameter that can affect the generalization of each optimizer. In Figure 10, we show the convergence behaviour of RST-AM with different weight-decays. It suggests that with a too small weight-decay, RST-AM tends to be overfitting in the test dataset.

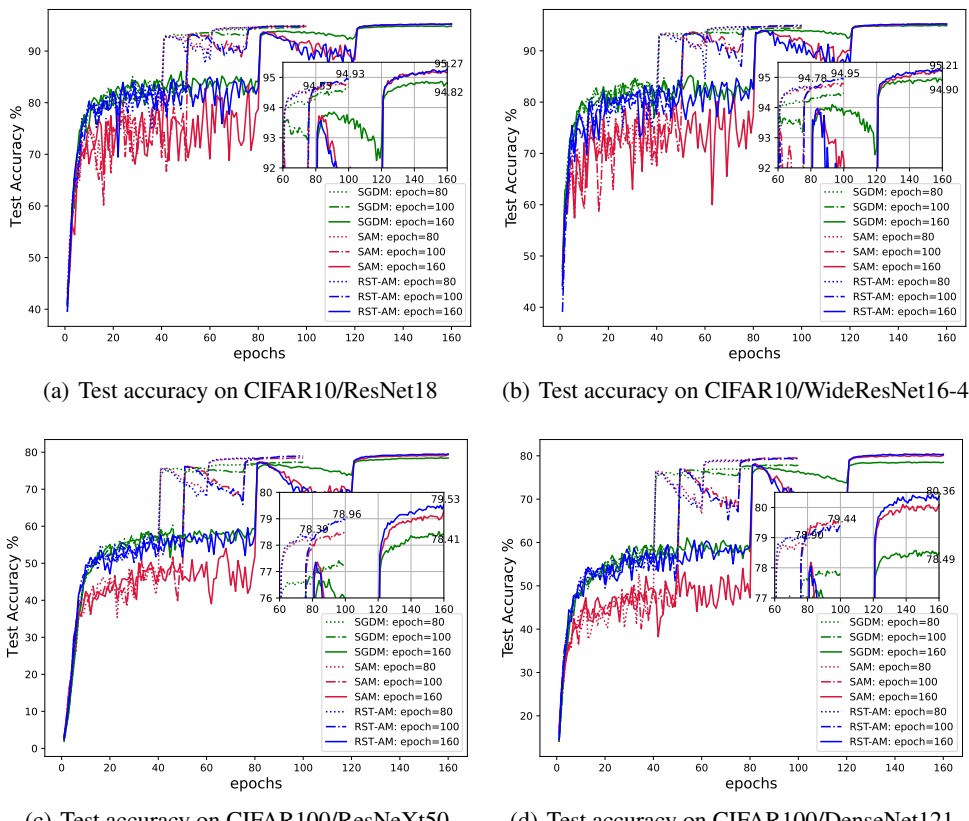

(a) Test accuracy on CIFAR10/ResNet18     (b) Test accuracy on CIFAR10/WideResNet16-4

(c) Test accuracy on CIFAR100/ResNeXt50     (d) Test accuracy on CIFAR100/DenseNet121

Figure 11: Training deep neural networks for 80,100,160 epochs. The results of final test accuracy of RST-AM for training 80,100,160 epochs and the final test accuracy of SGDM for training 160 epochs are shown in the nested figures for comparison.

Table 5: The cost and final test accuracy compared with SGDM. The notations "m","t/e", "e", "t" and "a" are abbreviations of memory, per-epoch time, training epochs, total running time, and accuracy, respectively. "*" indicates numbers published in (Wei et al., 2021).

| Cost (× SGDM) & accuracy | CIFAR10/ResNet18 | | | | | CIFAR10/WRN16-4 | | | | |
|---|---|---|---|---|---|---|---|---|---|---|
| | m | t/e | e | t | a(%) | m | t/e | e | t | a(%) |
| SGDM* | 1.00 | 1.00 | 1.00 | 1.00 | 94.82 | 1.00 | 1.00 | 1.00 | 1.00 | 94.90 |
| SAM(10)* | 1.73 | 1.78 | 0.56 | 1.00 | 94.81 | 1.26 | 1.28 | 0.63 | 0.80 | 94.94 |
| RST-AM | 1.05 | 1.46 | 0.56 | 0.82 | 94.84 | 1.03 | 1.14 | 0.63 | 0.71 | 94.95 |
| Cost (× SGDM) & accuracy | CIFAR100/ResNeXt50 | | | | | CIFAR100/DenseNet121 | | | | |
| | m | t/e | e | t | a(%) | m | t/e | e | t | a(%) |
| SGDM* | 1.00 | 1.00 | 1.00 | 1.00 | 78.41 | 1.00 | 1.00 | 1.00 | 1.00 | 78.49 |
| SAM(10)* | 1.30 | 1.16 | 0.50 | 0.58 | 78.37 | 1.16 | 1.19 | 0.50 | 0.60 | 78.84 |
| RST-AM | 1.04 | 1.07 | 0.50 | 0.54 | 78.39 | 1.01 | 1.11 | 0.50 | 0.55 | 78.90 |

Table 1(a) in the main paper reports the test accuracy of each optimizer when training for the same epochs. In fact, as shown in Figure 11, within 100 epochs, RST-AM can achieve a better test

accuracy than SGD. So if the running time of the training process matters, it is expected that RST-AM can use less total running time due to the large number of reduction in training epochs. In Table 1(b), we set SGD as the baseline, and compare the computation and memory cost with SGD. In Table 5, we give more details about the saving in training epochs and the final test accuracy. It can be seen that RST-AM can achieve a comparable or better test accuracy than SGDM with less computation time, while reducing the memory footprint of SAM.

We also tried using Adam as a preconditioner for RST-AM but found the final test accuracy was often worse. For example, the test accuracy of Adam_RST-AM for CIFAR-10/ResNet20 is only 90.79%. We suppose it is the worse generalization ability of Adam (Luo et al., 2018) that makes Adam not suitable as a preconditioner for RST-AM in the image classification task.

### D.2.4    EXPERIMENTS ON PENN TREEBANK

This group of experiments were the same as those in (Wei et al., 2021) for direct comparison. Results in Table 2 were measured with 3 random seeds. The parameter settings of the LSTM models were the same as those in (Zhuang et al., 2020; Wei et al., 2021). The baseline optimizers were SGDM, Adam, AdaBelief, and SAM. The validation dataset was used for tuning hyperparameters.

For SGDM, the learning rate (abbr. lr) was tuned via grid-search in $\{1, 10, 30, 100\}$. We set lr = 10, momentum = 0.9 for the 2-layer/3-layer LSTM, and lr = 30, momentum = 0 for the 1-layer LSTM.

For Adam, the learning rate was tuned via grid-search in $\{1\times10^{-3}, 2\times10^{-3}, 5\times10^{-3}, 8\times10^{-3}, 1\times10^{-2}, 2\times10^{-2}\}$. We found the setting that lr = $5 \times 10^{-3}$ performs best.

For AdaBelief, we set lr = $5 \times 10^{-3}$ which is better than the recommended settings of the official implementation[5].

For SAM, we used the recommended setting in (Wei et al., 2021) and used the baseline Adam as the preconditioner. The cases $m = 2$ and $m = 20$ are denoted as SAM(2) and SAM(10), respectively.

For our method RST-AM, we kept $c_1 = 1, c_2 = 1 \times 10^{-7}$ unchanged and used the same preconditioner (Adam) as SAM.

The batch size was 20. We trained the model for 500 epochs and the learning rate was decayed by 0.1 at the 250th epoch and the 375th epoch.

Table 6:    The cost to achieve comparable results of Adam. The notations "m","t/e" and "t" are abbreviations of memory, per-epoch time and total running time, respectively.

| Cost | 1-Layer | | | 2-Layer | | | 3-Layer | | |
|---|---|---|---|---|---|---|---|---|---|
| ($\times$ Adam) | m | t/e | t | m | t/e | t | m | t/e | t |
| Adam | 1.00 | 1.00 | 1.00 | 1.00 | 1.00 | 1.00 | 1.00 | 1.00 | 1.00 |
| SAM(10) | 1.20 | 1.84 | 0.74 | 1.36 | 1.88 | 0.75 | 1.90 | 1.67 | 0.67 |
| RST-AM | 1.06 | 1.73 | 0.69 | 1.15 | 1.78 | 0.71 | 1.11 | 1.53 | 0.61 |

Table 7: Test perplexity of training 1,2,3-layer LSTM on Penn Treebank for 200 epochs. Lower is better. "*" indicates numbers published in (Wei et al., 2021).

| Method | 1-Layer | 2-Layer | 3-Layer |
|---|---|---|---|
| Adam* | 80.88$\pm$.15 | 64.54$\pm$.18 | 60.34$\pm$.22 |
| SAM(2) | 81.82$\pm$.09 | 66.62$\pm$.26 | 61.55$\pm$.11 |
| SAM(10) | **79.30$\pm$.12** | **63.21$\pm$.02** | 59.47$\pm$.07 |
| RST-AM | 79.49$\pm$.11 | 63.61$\pm$.26 | **59.34$\pm$.23** |

In Table 6, we report the memory footprint and per-epoch running time of SAM(10) and RST-AM compared with Adam. It indicates that RST-AM also largely reduces the memory overhead of the

---

[5]https://github.com/juntang-zhuang/Adabelief-Optimizer/tree/update_0.2.0/PyTorch_Experiments/LSTM.

long-memory SAM(10) in the language modeling task. Since the batch size is very small, the cost of stochastic gradient evaluation is quite cheap, which makes the additional cost of matrix computation in RST-AM and SAM(10) become considerable. However, if we consider achieving a comparable validation/test perplexity to that of Adam, RST-AM does not need to train for the same number of epochs as Adam. Table 7 shows the test perplexity of Adam, SAM(2), SAM(10), and RST-AM for training 200 epochs. By comparing the results of Table 2 and Table 7, we see RST-AM is better than Adam with much fewer training epochs, thus RST-AM can save a large amount of training time, as shown in Table 6.

### D.2.5 Experiments on adversarial training

The adversarial training considers the *empirical adversarial risk minimization* (EARM) problem:

$$\min_{x \in \mathbb{R}^d} \frac{1}{T} \sum_{i=1}^{T} \max_{\|\bar{\xi}_i - \xi_i\|_2 \le \epsilon} f_{\bar{\xi}_i}(x), \tag{101}$$

where $\bar{\xi}_i$ is the $i$-th adversarial data within the $\epsilon$-ball centered at $\xi_i$. We followed the standard PGD adversarial training in (Madry et al., 2018), using projection gradient descent (PGD) to solve the inner maximization problem and the tested optimizers (SGD and RST-AM) to solve the outer minimization problem. The experiments were conducted on CIFAR10/ResNet18, CIFAR10/WRN16-4, CIFAR100/ResNet18, and CIFAR100/DenseNet121. We trained the neural networks for 200 epochs and decayed the learning rate at the 100th and 150th epoch.

Since adversarial training is much more time consuming than the ordinary training in Section D.2.3, we tuned the hyperparameters in CIFAR10/ResNet20, and applied the same hyperparameters to other models. We found the setting that $c_1 = 1, c_2 = 1 \times 10^{-7}$ and weight-decay = 0.001 is still suitable for this task.

The CIFAR-10 (CIFAR-100) dataset contains 50K images for training and 10K images for testing. Since it is found that the phenomenon of overfitting is severer (Rice et al., 2020) in adversarial training, we randomly selected 5K images from the total 50K training dataset as the validation dataset (the other 45K images remained as the training dataset), and chose the best checkpoint model in the validation set to evaluate on the test dataset. We consider two types of test accuracy:
(i) the clean test accuracy, where clean data was used for model evaluation;
(ii) the robust test accuracy, where corrupted data was used for model evaluation. The attacking methods are FGSM (Goodfellow et al., 2014), PGD-20 (Madry et al., 2018), and C&W$_\infty$ attack (Carlini & Wagner, 2017).

Table 8: Test accuracy (%) for adversarial training.

| Optimizer | CIFAR10/ResNet18 | | | | CIFAR100/DenseNet121 | | | |
|---|---|---|---|---|---|---|---|---|
| | Clean | FGSM | PGD-20 | C&W$_\infty$ | Clean | FGSM | PGD-20 | C&W$_\infty$ |
| SGD | 82.16 | 63.23 | 51.91 | 50.22 | 59.45 | 39.76 | 30.92 | 29.00 |
| RST-AM | 82.53 | 63.78 | 52.43 | 50.52 | 60.48 | 40.41 | 31.20 | 29.52 |
| Optimizer | CIFAR10/WRN16-4 | | | | CIFAR100/ResNet18 | | | |
| | Clean | FGSM | PGD-20 | C&W$_\infty$ | Clean | FGSM | PGD-20 | C&W$_\infty$ |
| SGD | 80.84 | 60.97 | 49.29 | 47.62 | 55.42 | 36.17 | 28.18 | 26.31 |
| RST-AM | 81.36 | 61.38 | 49.93 | 47.95 | 56.49 | 37.00 | 28.50 | 26.66 |

In Tabel 8, we report the average results of tests with three different random seeds. It shows that RST-AM can achieve both higher clean test accuracy and higher robust test accuracy across various models on CIFAR10/CIFAR100. To see the convergence behaviour of SGD and RST-AM, we plot the curves of the clean validation accuracy and the PGD-10 attacked validation accuracy in Figure 12. We can observe the phenomenon of robust overfitting (Rice et al., 2020) from these curves, which justifies our experimental setting with validation set for the checkpoint selection. Nonetheless, the numerical results suggest that RST-AM can still be better than SGDM in adversarial training.

It is also found that due to the heavy cost of gradient evaluations in PGD adversarial training, the additional computational cost incurred by RST-AM is negligible and the per-epoch running time of SGD and RST-AM is roughly the same. So we do not report it here.

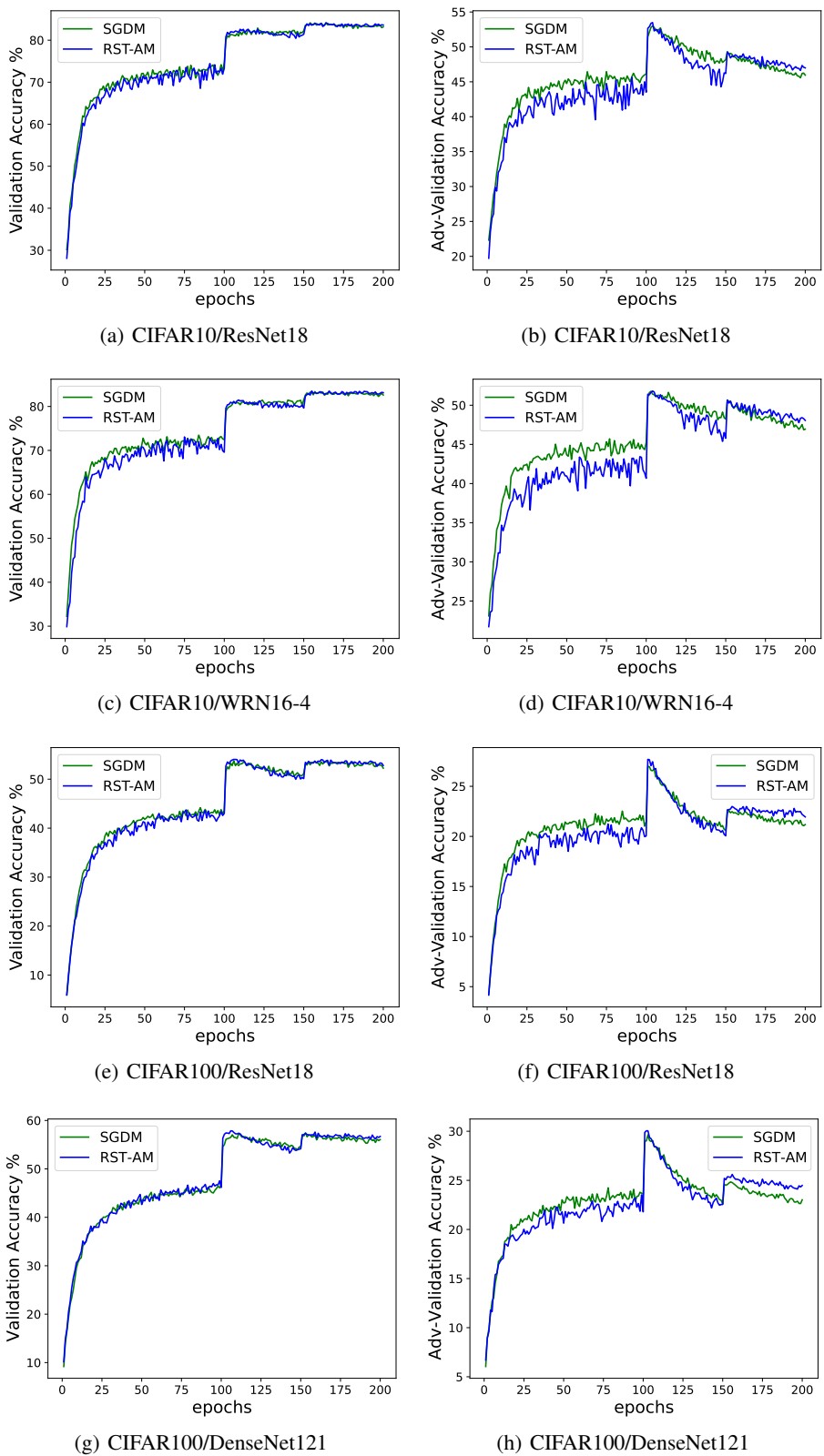

(a) CIFAR10/ResNet18

(b) CIFAR10/ResNet18

(c) CIFAR10/WRN16-4

(d) CIFAR10/WRN16-4

(e) CIFAR100/ResNet18

(f) CIFAR100/ResNet18

(g) CIFAR100/DenseNet121

(h) CIFAR100/DenseNet121

Figure 12: Clean accuracy and PGD-10 attacked accuracy on the validation set in training different neural networks.

### D.2.6 EXPERIMENTS ON TRAINING A GENERATIVE ADVERSARIAL NETWORK

We describe our setting of training a generative adversarial network (GAN) here. Like the adversarial training, the GAN training process is also a min-max problem. The stability of an optimizer is critical for the training process. To demonstrate the applicability of RST-AM, we conducted experiments on a GAN which was equipped with *spectral normalization* (Miyato et al., 2018) (SN-GAN). The experimental setting was the same as that of AdaBelief (Zhuang et al., 2020): the dataset was CIFAR-10; ResNets were used as the generator and the discriminator networks; the steps for optimization in the discriminator and the generator per iteration were 5 and 1, respectively; the minibatch size was 64 and the total iteration number was 100000. The Frechet Inception Distance (FID) (Heusel et al., 2017) was used as the evaluation metric: lower FID score means better accuracy.

The baseline optimizer were Adam and AdaBelief as they perform well in this task (Zhuang et al., 2020). We also used the recommended hyperparameter settings for the two optimizers. For our RST-AM method, due to the ill-conditioning of the min-max problem, we used the AdaBelief as the preconditioner and set the damping parameter $\alpha_k = 0.6$. The regularization parameters $c_1 = 1$ and $c_2 = 1 \times 10^{-7}$ were still kept unchanged just as the previous experiments.

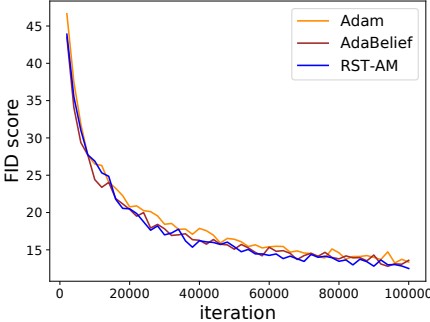

Figure 13: FID score for training SN-GAN on CIFAR-10.

In Figure 13, we show the curve of FID score for each optimizer, which is the average of three independent runs. It indicates that the RST-AM method is stable for this min-max optimization problem.

Table 9: The effect of $\alpha_k$ for RST-AM.

| Method | Adam | AdaBelief | $\alpha = 0.8$ | $\alpha = 0.6$ | $\alpha = 0.5$ | $\alpha = 0.4$ | $\alpha = 0.2$ | $\alpha = 0.1$ |
|---|---|---|---|---|---|---|---|---|
| FID score | 13.07 | 12.80 | 12.48 | **12.05** | 12.75 | 13.13 | 13.07 | 12.59 |

Since we only tuned the damping parameter $\alpha_k$ for RST-AM, we report the FID scores of other choices of $\alpha_k$ in Table 9 during our experiment. It shows that even with the suboptimal choices of $\alpha_k$, e.g. $\alpha_k = 0.1, 0.5, 0.8$, RST-AM can still outperform the baselines.

### D.3 ADDITIONAL EXPERIMENTS

To further test the performance of our method in training neural networks on larger datasets or different models, we conducted additional experiments of the image classification task in ImageNet (Deng et al., 2009) and the Transformer (Vaswani et al., 2017) based neural machine translation task in the IWSTL14 DE-EN (Cettolo et al., 2014) dataset.

### D.3.1 EXPERIMENTS ON IMAGENET

We trained ResNet50 on ImageNet with SGDM and RST-AM. We used the built-in ResNet50 model in PyTorch. We ran the tests of each optimizer with three random seeds and four GeForce RTX 2080

Ti GPUs were used for each test. The hyperparameters of SGDM were set as the recommended setting in PyTorch. For RST-AM, the hyperparameters were kept the same as those in the CIFAR experiments. The weight-decay was $1 \times 10^{-4}$. The number of the total training epochs is 90. The learning rate decay of SGD was at the 30th and 60th epochs. For RST-AM, since the experiments on CIFAR show it can often converge faster to an acceptable solution than SGDM, we adopted the early learning rate decay strategy recommended in (Zhang et al., 2019): decay the $\alpha_k$ and $\beta_k$ for RST-AM at the 30th, 50th and 70th epochs.

Table 10: TOP 1 test accuracy (%) w.r.t. epoch, the best TOP1 test accuracy (%), and the cost. The memory, per-epoch time and total time of SGDM are set as the units. The total time is the time to first achieve the accuracy $\geq 75.90\%$.

| Method | epoch = 72 | epoch = 88 | epoch = 90 | best | memory | per-epoch time | total time |
|--------|-----------|-----------|-----------|------|--------|----------------|------------|
| SGDM   | 75.75     | 75.90     | 75.81     | 75.93±.15 | 1.00 | 1.00 | 1.00 |
| RST-AM | 75.90     | 75.95     | 75.98     | 76.04±.06 | 1.06 | 1.01 | 0.83 |

In Table 10, we report the TOP1 accuracy in the validation dataset during training. Note that we also report the epoch number for each optimizer to achieve the accuracy equal or exceeding 75.90%. It shows RST-AM needs 72 epochs while SGDM needs 88 epochs. The curves of the training accuracy and test accuracy are shown in Figure 14. The results suggest that RST-AM is still a competitive optimizer in training a larger model in a larger dataset.

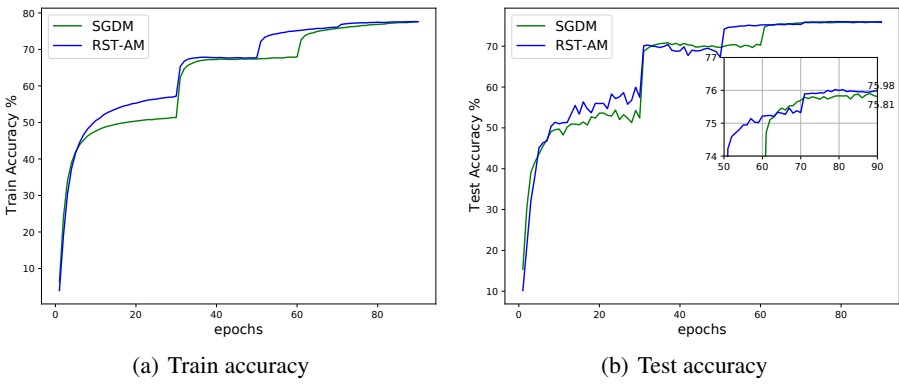

(a) Train accuracy  (b) Test accuracy

Figure 14: Train and test accuracy for training ImageNet/ResNet50.

### D.3.2 TRAINING TRANSFORMER

We conducted the neural machine translation task with Transformer. We implemented our RST-AM method and integrated it into the fairseq framework [6]. The basic experimental setting was set as the recommended setting in (Yao et al., 2021). We trained the model for 50 epochs and the BLEU (Papineni et al., 2002) score was calculated using the average model of the last five checkpoints. We added a baseline optimizer RAdam (Liu et al., 2019) which was inspired by the warmup procedure in training Transformer.

Table 11: The BLEU score of training Transformer on IWSLT14.

| Method | SGD | Adam | AdaBelief | RAdam | RST-AM |
|--------|-----|------|-----------|-------|--------|
| BLEU score | 28.14±.08 | 35.71±.03 | 35.15±.14 | 35.60±.06 | **35.89±.02** |

---

[6] https://github.com/pytorch/fairseq.

The numerical results reported in Table 11 show that Adam is well-suited for this neural machine translation task, though it does not perform well in the image classification task. Also, the results demonstrate that RST-AM can still outperform Adam in this task.

Table 12: BLEU score evaluated at the 40/45/50-th epoch, and the cost. The memory, per-epoch time and total time of Adam are set as the units. The total time is the time of RST-AM to achieve a BLEU score matching the final BLEU of Adam: $|$ BLEU(RST-AM) $-$ BLEU(Adam) $| \leq 0.03$, where 0.03 is the standard deviation of the results of Adam.

| Method | epoch = 40 | epoch = 45 | epoch = 50 | memory | per-epoch time | total time |
|---|---|---|---|---|---|---|
| Adam | 35.42±.10 | 35.54±.08 | 35.71±.03 | 1.00 | 1.00 | 1.00 |
| RST-AM | **35.59±.14** | **35.69±.06** | **35.89±.02** | 1.16 | 1.00 | 0.90 |

In Table 12, we report the BLEU scores evaluated in the test dataset at the 40th, 45th and 50th epochs. The results show the better performance of RST-AM over Adam and the per-epoch computational cost is nearly the same. To achieve a comparable solution to Adam, RST-AM can save 10% training time.

