# OpenReview forum: "A Class of Short-term Recurrence Anderson Mixing Methods and Their Applications"
_ICLR.cc/2022/Conference — ICLR 2022 Poster_

### Official Review · Reviewer_i7Ws · 2021-10-19

**Correctness:** 3
**Technical Novelty And Significance:** 4
**Empirical Novelty And Significance:** 3
**Recommendation:** 8
**Confidence:** 2

**Main Review:**

I begin by noting that I am not an expert on the finer details of the theory of (non)linear solvers, so my review is primarily focused on the application side of things.

This appears to be an excellent paper. The quality of the writing is uniformly high throughout. I appreciated the various remarks throughout, used to elucidate the occasional mathematical detail.

My first comment is that I think it would be really beneficial to more explicitly state the delta between this work and previous work. When is it worth the effort to implement the introduced methods as compared to some off-the-shelf competing method? (LU solvers for linear problems; quasi-Newton for nonlinear problems; Adam for supervised learning etc.) For the audience interested in applications, rather than theory, I believe a section describing this would be especially valuable.

I would describe the experiments as still having room to improve. (And I apologise for being "that reviewer" who asks for more experiments, in a paper that already features several.) Numerical methods for the solution of (non)linear systems are applied widely across disparate fields, so in this case I think it is particularly important to check how broadly applicable any new technique is. I would regard the following as being particularly important extra comparisons:

- Section 4.1: how does the proposed method compare to Newton and standard quasi-Newton methods (BFGS, L-BFGS, chord, Levenberg–Marquardt, etc.)? For those problems for which the Jacobian is available -- an increasingly common case following the popularisation of autodifferentiation -- these are still the go-to technique in many/most regimes.
- Section 4.1: the test problems considered are all very simple. What about more complicated test problems? In particular I would be interested in finding the fixed point of "complicated" nonlinear functions (e.g. a neural network, or the system arising from an implicit ODE solver).
  - Focusing on the implicit ODE case: at present this is ubiquitously done via Newton's method or the (quasi-Newton) chord method. (And in particular not via Picard iterations, which can introduce an undesirable loss of stability to the overall numerical method.)
- Training GANs. The choice of optimiser can have a *very* dramatic impact when training GANs: easily making the difference between convergent and divergent training, despite the same change in optimiser having minimal-to-no effect in the supervised learning case. A few examples:
  - [1] consider a case in which GAN training with SGD/Adam fails completely, but training with Adadelta produces stable/convergent results.
  - [2] derive optimisers by solving the gradient flow ODE. [Usually not a great idea in the supervised learning case, but the unreliability of other optimisers in the adversarial case motivates a return to this idea.]
  - [3, 4] consider using negative or complex momentum within the optimiser.
  - [5] consider taking Cesàro means over the parameters being optimised, during training. (This is commonly also done in theoretical studies of optimisers.)

An in addition the following would be nice as extra comparisons, but I think they are less important than those suggested above:

- Section 4.2: MNIST: SAM and RST-AM preconditioned with Adam is noticeably absent.
- Section 4.2: CIFAR: SAM and RST-AM with Adadelta/Adagrad/Adam/RMSprop preconditioners. These seemed to help a lot on MNIST. (Figure 2(c))

## Minor points / typos

Equation (1): I think specifying $X_k \in \mathbb{R}^{m \times d}$, $R_k \in \mathbb{R}^{m \times d}$ would improve readability slightly.

Remark 1: I'd suggest including here that the mixing step is essentially a gradient descent step ($\bar{r}_k$ is some linear combination of $\nabla f(x_j)$; c.f. also the standard condition given by equation (14)).

Top of page 4: both $\bar{x}_k^G$ and $x_k^G$ are used. Neither notation is really defined precisely, but I am guessing that $\bar{x}_k^G$ is meant to be $x_k^G$.

Top of page 4: it is stated that $\bar{x}_k = \bar{x}_k^F = x_k^G$ . This would seem to indicate that the $\bar{x}_k$ iterates of ST-AM compute exactly the same values as AM (with the corollary that the actual output $x_k$ is almost identical), and as such ST-AM is a strict improvement  over AM due to its lower memory requirements. This seems at odds with the rest of the text, which seems to suggest that the use of ST-AM represents a trade-off.

Theorem 2: "constant $\kappa$..." -> "constants $\kappa$..."

Section 3.4, end of second paragraph: $\nabla_{S_k} f(x_k)$ -> $\nabla f_{S_k}(x_k)$.

Equation (14): some space (\qquad) between the equations would be nice.

End of page 8: "less training epochs" -> "fewer training epochs"

Page 9: "less epochs" -> "fewer epochs"

## References

[1] Kidger et al. "Neural SDEs as Infinite-Dimensional GANs" NeurIPS 2021 https://arxiv.org/abs/2105.13493

[2] Qin et al. "Training Generative Adversarial Networks by Solving Ordinary Differential Equations" NeurIPS 2020 https://arxiv.org/abs/2010.15040

[3] Gidel et al. "Negative Momentum for Improved Game Dynamics" AISTATS 2019 https://arxiv.org/abs/1807.04740

[4] Lorraine et al. "Complex Momentum for Optimization in Games" 2021 https://arxiv.org/abs/2102.08431

[5] Yazici et al. "The Unusual Effectiveness of Averaging in GAN Training" ICLR 2019 https://arxiv.org/abs/1806.04498

**Summary Of The Paper:**

The paper introduces three variants of Anderson acceleration: short-term Anderson mixing (ST-AM), modified short-term Anderson mixing (MST-AM), and regularised short-term Anderson mixing (RST-AM). These may be used to obtain the solution to linear systems, nonlinear systems, and optimisation problems.

**Summary Of The Review:**

I recommend acceptance due to the high quality of the paper. My main two concerns are (a) readability wrt applications and (b) experiments across a broader range of regimes.

---

> ### Author Response · Authors · 2021-11-20
> **Response to Reviewer i7Ws**
>
> Thank you for your support and comments. We hope our response can clarify our contributions and address your concerns.
>
> Q1. The delta between this work and previous work.
> A1. Thanks for your suggestion. We briefly discuss it here and a detailed discussion has been given in Appendix B.2 in the revision.
> 1. ST-AM versus LU solvers for linear systems. The LU solvers can incur overwhelming memory and computation cost for large sparse linear systems, while ST-AM is very efficient and economical owing to the convergence theory and the short-term recurrence.
> 2. MST-AM versus quasi-Newton methods for nonlinear problems. Based on short-term recurrences, MST-AM is more suitable for large-scale and high-dimensional problems.
> 3. RST-AM versus first-order methods for optimization. RST-AM is a second-order method but with memory footprint close to first-order methods. It can be applied to improve any slowly convergent black-box iterative process by viewing the latter as a fixed-point iteration. It can also efficiently incorporate preconditioning. Any optimizer, even an optimizer built upon neural networks, can be used as a preconditioner for RST-AM, which largely enhances the applicability of RST-AM for various applications.
>
> Q2. The comparison of MST-AM to other Newton-like methods.
> A2. Newton method. In each step, the Newton method needs to compute the Hessian and solve a linear system which is prohibitive in large problems.
> Limited quasi-Newton methods. Many historical iterates/gradients are needed to form the secant equations. Instead, the MST-AM uses the short-term recurrences.
> In summary, the ST-AM can be seen as an efficient and economical method for solving high-dimensional and large-scale problems.
>
> Q3. MST-AM as a fixed-point solver for the root-finding problem.
> A3. Thanks for your suggestions.
> (1) As guaranteed by our Theorem 1, Corollary 1 and Theorem 2, MST-AM can be very efficient to solve the sparse linear system arising from an implicit method for solving ODE.
> (2) We have applied the MST-AM to a complex root-finding problem: training mutiscale deep equilibrium model (MDEQ) [1].  In this experiment, we choose Broyden’s method (default method in [1]) and the full-memory AM for comparison. The results are shown in Figure 1 in Section 4.1 in revision. The results show that MST-AM can achieve competitive results (0.79% higher),  and is faster than Broyden’s method in both forward and backward process. Compared to full-memory AM, MST-AM is comparable in forward process and faster in backward process.
>
> Q4. The results of RST-AM for training GAN.
> A4.  Thanks for providing the excellent references and detailed explanations in training GAN. Indeed, training GAN is a min-max problem that is hard to solve and SGD often exhibits instability for the training.
> Although we do not explicitly give a theoretical analysis of applying RST-AM to the min-max problems due to space limit, we applied the RST-AM to train the spectrally normalized GAN (SG-GAN) [2] empirically. The results were added to the Section 4.2 in the revision, and more experimental details are given in Appendix D.2.6.
>
> The FID scores of RST-AM compared with other optimizers are given in the following table:
>
> Method| Adam| AdaBelief | RST-AM
> :-: | :-: | :-: | :-:
> Best FID|13.07|12.80|12.05
> Final FID|13.34|13.59|12.50
>
> It indicates that RST-AM is applicable and consistently outperforms the Adam optimizer. Since RST-AM is a general solver, it may not leverage the structures of the min-max problem. We will also give deep investigation to the min-max problem and develop more efficient solvers based on our ST-AM methods for the min-max problem in future works.
>
> Q5. Adam as the preconditioner for SAM/RST-AM on MNIST experiments.
> A5. We added the comparison in Appendix D.2.2. The results suggest that Adam is not suitable to serve as a preconditioner for SAM/RST-AM in this task. Nevertheless, the Adam preconditioned RST-AM still outperforms Adam and the Adam preconditioned SAM.
>
> Q6. Adam as the preconditioner for SAM/RST-AM on CIFAR experiments.
> A6. Our experience shows that the adaptive learning rate methods such as Adam are not suitable to serve as a preconditioner in this task, which may come from the bad generalization ability. We conducted a test of using Adam preconditioned RST-AM to train CIFAR10/ResNet20, the final accuracy is much worse than that of without preconditioning, as also discussed at the end of Section D.2.3. In fact, the AM/SAM/RST-AM can be viewed as being preconditioned by SGD with the mixing parameter $\beta_k$ being the stepsize of SGD.
>
> Q7. Minor points and typos.
> A7. Thanks for pointing out these issues. We have corrected them in revision.
> ST-AM is equivalent to the full-memory AM in solving (5), but may not in general cases.
>
> References
> [1] Shaojie Bai, et al. Multiscale deep equilibrium models. NeurIPS 2020.
> [2] Takeru Miyato, et al. Spectral normalization for generative adversarial networks. ICLR 2018.

---

> > ### Comment · Reviewer_i7Ws · 2021-11-20
> > **Response**
> >
> > Thank you to the authors for their response, and in particular for the additional experiments.
> >
> > Overall my belief is that this is an excellent paper; correspondingly I have updated my score from a 6 to an 8. I regret that I was not able (nor were any of the other reviewers) to provide a meaningful review of the theoretical aspects of this paper, so my confidence is low at merely a 2. (Meta-reviewer: perhaps an expert can be found to review this aspect of the paper?)
> >
> > I do note that:
> > - There are still no experimental comparisons to standard (quasi-)Newton methods. (Which would be feasible for some of the problems considered.)
> > - For completeness, quite general linear and nonlinear problems can arise from implicit ODE solvers, not merely "sparse linear system[s]".
> > - The intended audience of the paper is clearly a theoretical audience, rather than an applied audience. That's totally fine, but I'd encourage the authors to write a blog post/etc. discussing just how to practically implement their techniques.

---

> > > ### Author Response · Authors · 2021-11-21
> > > **Thank you for your support**
> > >
> > > Thanks a lot for your support and raising the score. We add some clarifications, hoping them can further address your concerns.
> > >
> > > Q1. Experimental comparisons to standard (quasi-)Newton methods.
> > > A1. The compared quasi-Newton methods in our experiments are listed as follows.
> > > (1) Cubic-regularized optimization: Figure 1(c) contains the BFGS method.
> > > (2) Root-finding problem: Figure 1(d-f) contains the Broyden’s method which is a default solver in MDEQ.
> > > Both the above two methods belong to the standard quasi-Newton methods: BFGS is for unconstrained optimization and Broyden's method is for nonlinear equations [1].
> > >
> > > In large-scale problems, the memory issue of standard Newton's method can be prohibitive. For example, the tested MDEQ model in our experiments has 170K parameters, then the size of the Jacobian is $(1.7\times 10^{5})^2 = 28.9$ G, which is larger than the GPU memory in our device. Thus, we did not compare ST-AM with standard Newton's method in this case.
> > >
> > > In the task of image classification, we have tested the performance of L-BFGS ( a built-in quasi-Newton method in PyTorch). But it is found that the training process is not stable and exhibits a divergence behaviour. Besides, we have tested the SdLBFGS [2] which is a stochastic version of L-BFGS.  The final accuracy is less than 80% in CIFAR-10/ResNet20 that is significantly below  our baselines ($\geq $ 90% accuracy).  Thus, in our paper, we choose the AdaHessian [3], a recent second-order method, as the baseline. From the results in Table 1, it is shown that  though AdaHessian benefits a lot from the automatic differentiation of PyTorch to calculate Hessian-vector products, the memory and computation cost is still very large. Indeed, as shown in Table 1(b)
> > > in Section 4.2, AdaHessian significantly increases the memory footprint ($\geq$2.2 times that of RST-AM) and runs out of the GPU memory for training ResNeXt50 and DenseNet121 on CIFAR-100.  Moreover, the test accuracy of RST-AM is better than AdaHessian (1.00% in average).
> > >
> > >
> > > Q2. The linear or nonlinear systems arising from implicit ODE solvers.
> > > A2. Thanks for pointing it out. We agree with your suggestion.
> > > The proposed ST-AM is applicable for general linear or nonlinear systems, not merely the sparse linear systems. In the MDEQ experiment, every iteration we need to solve complex nonlinear equations both in the forward process and in the backward process.
> > > A comprehensive test of ST-AM on solving PDE/ODE arising from scientific computing is an important problem, but this may significantly increase the length of the paper (current version: 48 pages) and may be out of the scope of this paper. We will systematically study this problem in future.
> > >
> > > Q3. The implementation of the ST-AM methods.
> > > A3. Thanks for your suggestion. We will post a blog to show the implementation details of ST-AM after the review process.
> > >
> > > References
> > > [1] Jorge Nocedal and Stephen Wright. Numerical optimization. Springer Science & Business Media, 2006.
> > > [2] Xiao Wang et al. Stochastic quasi-Newton methods for nonconvex stochastic optimization. SIAM Journal on Optimization, 2017.
> > > [3] Zhewei Yao et al. Adahessian: An adaptive second order optimizer for machine learning. AAAI 2021.

---

### Official Review · Reviewer_Axgn · 2021-11-04

**Correctness:** 3
**Technical Novelty And Significance:** 3
**Empirical Novelty And Significance:** Not applicable
**Recommendation:** 6
**Confidence:** 3

**Main Review:**

I think overall this is a paper with solid theory and experiments. The theoretical and experiments can be separated into two parts, one part is for the basic version of ST-AM designed mostly for nice problems like strongly convex quadratic ones, the other part is for RST-AM designed for neural network training. In my perspective, the first part may not be of too much interest to the audience of ICLR and is more suitable for optimization journals. Yet, the part of the MST-AM algorithm could be of great interest since it is applicable to training deep models. For the weakness, I feel the discussion on improvement of MST-AM over SAM (Wei et al., 2021) could be improved. The paper mentioned MST-AM is motivated by SAM  and talked about which steps are different. It could be better if the authors could provide a discussion on intuitions why MST-AM outperforms SAM in the paper. Also, the content of the paper is quite dense, it might be better to move some less important results into appendix and provide more discussion on intuitions in the main paper.

**Summary Of The Paper:**

The paper proposes a new class of memory-efficient Anderson mixing (AM) methods. Compared with classical Anderson mixing which requires saving m historical iterates, the new variants require only storing two historical iterates while keeping good performance. Convergence analyses are given to the proposed variants showing a variant for strongly convex quadratic problem enjoys the same guarantee as full-memory AM, and another variant converges with a similar convergence rate as SGD (O(1/\sqrt{number of iterations})) on nonconvex problems. Experiments on MNIST, CIFAR-10, and PENN TREEBANK with some popular deep models validates the superior performance of one of the proposed method.

**Summary Of The Review:**

In summary, I think this is a relatively strong paper due to the technical depth and the empirical studies, though some parts could be improved.

---

> ### Author Response · Authors · 2021-11-20
> **Response to Reviewer Axgn**
>
> Thank you for your support and comments. We hope the following responses can address your concern.
>
> Q1. The first part of the experiments.
> A1. Our proposed modified ST-AM (MST-AM) (see Section 3.3) is a fixed-point solver that can be useful in machine learning. To show the applicability, we added an experiment of applying the  MST-AM method to solve the root-finding problem in multiscale deep equilibrium (MDEQ) model [1], which is a recent proposed implicit neural network model. The results have been added to Figure 1 in Section 4.1. The MST-AM can be a competitive fixed-point solver compared with the built-in quasi-Newton methods while the memory and computation cost can be reduced since MST-AM uses short-term recurrences.
>
> Q2. The improvement of ST-AM over SAM.
> A2. Our work has fundamental improvements over SAM:
> (1) In theory, SAM is based on the limited-memory Anderson mixing which means that SAM is not equivalent to full-memory AM when solving strongly convex quadratic optimization (or solving SPD linear systems). On contrary, it is proved that ST-AM is equivalent to the full-memory AM in this case.
> (2) In practice, setting the number of historical iterations is heuristic in SAM, while RST-AM only needs to store two previous iterations. Compared with SAM, the reduced memory requirement in RST-AM makes it applicable for training more challenging problems in machine learning.
>  We have added the discussion in Remark 3 in Section 3.3 and Remark 5 in Section 3.4 in the revision.
>
> Q3. The intuition of the improvement by the ST-AM methods.
> A3. The key difference between the ST-AM methods and the limited-memory AM (including SAM) is the usage of the historical information. Unlike the limited-memory AM that simply discards the oldest iteration to make space for $\Delta x_{k-1}$ and $ \Delta r_{k-1} $, the ST-AM methods carefully incorporate historical information with some form of linear combination (e.g. orthogonalization in the basic ST-AM). In the ideal case, i.e. strongly convex quadratic optimization, the basic ST-AM is equivalent to the full-memory AM which means there is no loss of historical information. For general nonlinear problem, since a smooth function can be approximated  via a quadratic function in a small region around the optima, it is expected that the ST-AM method can still roughly match the performance of the full-memory AM, as also demonstrated in our experiments.
>
> The discussions about the intuitions have been added in the revision (see Remark 3 in Section 3.3 and Remark 5 in Section 3.4).
>
> References
> [1] Shaojie Bai, Vladlen Koltun, and J. Zico Kolter. Multiscale deep equilibrium models. In Advances in Neural Information Processing Systems (NeurIPS), 2020.

---

### Official Review · Reviewer_xxcF · 2021-11-07

**Correctness:** 3
**Technical Novelty And Significance:** 3
**Empirical Novelty And Significance:** 3
**Recommendation:** 6
**Confidence:** 3

**Main Review:**

The paper tackles an ambitious goal, which is to design better optimizers for machine learning workloads than the first order methods which have come to dominate the field. It makes an interesting contribution with several well-motivated new algorithms, establishing convergence proofs and running detailed experiments to check whether the theory matches the practice.
The paper is generally well-written and nice to read, with special care taken to cite appropriate earlier works.

Here are a few outstanding issues, however.
1. The paper is perhaps not as self-contained as one could hope for, and assumes a strong familiarity with many topics: e.g. Anderson Mixing, the properties of first order methods on fixed point problems, preconditioning (which is only made explicit far in the Appendix). It would make the paper more readable and easier to understand to add more context to it.

2. A claim is made that the memory footprint of the algorithms is close to that of GD or SGD, but a more precise theoretical comparison would be helpful. For instance, we see that the memory footprint seems to only increase by 5% over SGD in some experiments, but are bigger than Adam in others which leaves the reader wondering how these seemingly inconsistent results came about.

3. Finally, the biggest issue with the paper is its experimental section. While thorougly reported (which is appreciated), it relies on toyish datasets which make the claims of the paper more difficult to support. Another issue is that the authors chose RNNs for language tasks while transformers have obtained SOTA results for years, and Adam seems to be more critical to good transformer performance than good RNN performance. All told, it seems like the experimental section, while well-executed, misses key points to make it really relevant to the community.

All told, this is a nice paper which clears the bar for publication at ICLR this year, but which could be much improved with a revamped, more ambitious experimental section.

Details
- section 3.4, first paragraph: optimizaiton --> optimization
- fig 1 caption : Probrem --> Problem
- page 34 precondtioner --> preconditioner

**Summary Of The Paper:**

The paper introduces 3 new variants of Anderson Mixing methods relying on very limited short term memory. This makes these methods more attractive for usual machine learning workloads.
The paper also provides detailed analysis of the performance of each of these AM algorithms, as well as thorough experiments, showcasing improved performance of select neural network training.

**Summary Of The Review:**

Interesting new algorithms, well-motivated, delivered with good theoretical analysis. The experimental section is a bit lacking, but that's not sufficiently concerning not to recommend the paper for publication.

---

> ### Author Response · Authors · 2021-11-20
> **Response to Reviewer xxcF**
>
> Thank you for your support and comments. We hope our replies can address your concerns.
>
> Q1. The preliminaries of reading this paper.
> A1. Thank you for your advice. We add more context about the preliminaries including Anderson mixing in Appendix A.2, fixed-point iterations in Appendix A.1, and preconditioning in Appendix A.3.
>
> Q2. The analysis of the memory footprint of the ST-AM methods.
> A2.  (1) We give a theoretical analysis in Appendix B.1 about the memory footprint. Since ST-AM methods only need to store two previous iterations, the additional memory cost compared with SGD is $4d$, where $d$ is the model parameter size. It can be a great reduction of memory requirement compared with the $2md$ additional memory cost of AM($m$) where $m$ is usually greater than 5.
> (2) Since Adam uses moving average which incorporates one previous iteration, the memory footprint of Adam is smaller than RST-AM which stores two previous iterations. Also, since the experiments on language model used a very small batch size and the model parameter size is considerably large, the additional memory overhead is larger than that in the image classification task. Moreover, since the codes were written in Python, the results may not exactly reflect the theoretical analysis. We empirically found that even with different version of PyTorch, the results can  be different. For example, for training 2-layer LSTM, with PyTorch1.7.0, the memory footprint of RST-AM is 1.07 times that of Adam, while with PyTorch1.8.0, it is 1.15 times that of Adam. We will give further investigation into this issue in the future work.
>
> Q3. The experiments on larger datasets and training more advanced neural network models.
> A3. (1) Tests on larger datasets.
> To test the performance of our RST-AM method in larger datasets, we conducted experiments on training ResNet50 in ImageNet. For SGD, we used the official recommended setting of PyTorch. The hyperparameter setting of RST-AM was kept the same as the CIFAR experiments.
>
> Method |  epoch = 72  |  epoch = 88 | epoch = 90  |    best  | memory | per-epoch time | total time
> :-: | :-: | :-: | :-: | :-: | :-: | :-: | :-:
> SGD |       75.75          |   75.90         |  75.81          |  75.93  |   1.00      |   1.00                | 1.00
> RST-AM | 75.90          |   75.95        |   75.98         |   76.04  |   1.06      |   1.01              |   0.83
>
> We report the TOP1 accuracy w.r.t. epoch, the best accuracy, and the memory/computation cost in the table (also can be found in Appendix D.3.1).  The cost of SGD is set as the unit. The total time is the running time to first achieve an acceptable 75.90% accuracy. It indicates that RST-AM can still show promising results in a larger dataset.
>
> (2) Tests on Transformers.
> We implemented RST-AM and integrated it into the fairseq framework, and then applied it to train Transformers on the IWSLT14 DE-EN dataset. We used the official recommended experimental setting. The BLEU scores are listed as follows:
>
> Method |  SGD  |  Adam | AdaBelief  |    RAdam  | RST-AM
> :-: | :-: | :-: | :-: | :-: | :-:
> BLEU |    28.14  |   35.71 |  35.15       |  35.60       |   35.89
>
> The results show that Adam is well-suited for training Transformer, but still cannot outperform RST-AM. We also report the BLEU scores evaluated in the test datasets at the 40th, 45th, and 50th epochs, and the memory/computation cost as follows (the cost of Adam is set as the unit):
>
> Method |  epoch = 40  |  epoch = 45 | epoch = 50    | memory | per-epoch time | total time
> :-: | :-: | :-: | :-: | :-: | :-: | :-:
> Adam |       35.42±.10         |   35.54±.08         |  35.71±.03           |   1.00      |   1.00     | 1.00
> RST-AM |  35.59±.14          |   35.69±.06        |   35.89±.02         |   1.16      |   1.00     |   0.90
>
> The results indicate that to achieve a comparable solution within the standard deviation of the final result of Adam (35.68~35.74), RST-AM can still save 10% training time.
>
>
> Q4. Improvement of the experimental section.
> A4. Thank you for your advice about our experimental section. We hope that our additional experiments (please see two more experiments in deep equilibrium model (DEQ) and GAN in our general response) can further verify the advantages of the proposed ST-AM.
>
> Q5. Language issues.
> A5. Thanks for pointing them out. We have corrected them.

---

### Author Response · Authors · 2021-11-20
**Response to all the reviewers**

Dear reviewers,

We sincerely thank your support and the constructive suggestions. We tried our best to revise the paper and hope these results can demonstrate the advantages of ST-AM. We summarize the main changes.
1. Experimental perspective. To demonstrate the applicability of ST-AM, we have added the following experiments according to the reviews to test the performance.
(1.1) Root finding problems. The MST-AM has been applied to train the mutiscale deep equilibrium (MDEQ) model [1] which is a recent proposed implicit neural network model (see Section 4.1).
(1.2) Training GAN. We have applied the RST-AM to train spectrally normalized generative adversarial networks (SN-GAN) [2] (see Section 4.2).
(1.3) Training ResNet50 in ImageNet (see Appendix D.3.1).
(1.4) Neural machine translation with Transformer (see Appendix D.3.2).

In summary, the current experiments related to deep learning contain
(a) the root-finding problem in MDEQ model;
(b) training CNN on MNIST;
(c) image classification with various deep neural networks on CIFAR-10/CIFAR-100;
(d) language model on Penn Treebank;
(e) adversarial training on CIFAR-10/CIFAR-100;
(f) training SN-GANs on CIFAR-10;
(g) training ResNet50 on ImageNet;
(h) training Transformers on IWSLT14 DE-EN.
Our experiments consistently show that ST-AM can achieve the comparable results with less training time.

2. Writing perspective.
(2.1) More discussions about the intuition of ST-AM have been added (In Remark 3 and Remark 5).
(2.2) Preliminaries about fixed-point iteration, (preconditioned) Anderson mixing (AM) have  been added in Appendix A to make our paper more self-contained.
(2.3) The theoretical analysis of the memory and computational cost of ST-AM has been added in Appendix B.1.
(2.4) We give comparisons of our methods with the LU solvers for linear systems, quasi-Newton methods for nonlinear problems and first-order methods for optimization (see Appendix B.2).

Finally, we would like to highlight the theoretical analysis of ST-AM, which is the main focus of our paper, by briefly introducing the development of AM. Although AM is proposed in 1965 and has been widely applied in scientific computing, there is no systematic convergence results until 2015 [3]. The recent paper by Mai & Johansson [4] has constructed a counter-example that the AM fails. Thus, to establish the convergence theory, it needs several modifications without losing its original efficiency.
Based on the above analysis, the modifications of AM have two difficulties: (i) the convergence theory; (ii) setting the number of historical iterations needed to be stored.  In our paper, the convergence results of the proposed ST-AM consist of the strongly convex quadratic problem, nonlinear fixed-point problem, and stochastic optimization problems. Moreover, rather than setting the number of historical iterations heuristically in practice, the ST-AM methods only need to store two previous iterations.

We hope the revision and the following answers can address your concerns and help you better evaluate our work.

Thank you!

Yours faithfully,

The Authors.

References
[1] Shaojie Bai, Vladlen Koltun, and J. Zico Kolter. Multiscale deep equilibrium models. In Advances in Neural Information Processing Systems (NeurIPS), 2020.
[2] Takeru Miyato, Toshiki Kataoka, Masanori Koyama, and Yuichi Yoshida. Spectral normalization for generative adversarial networks. In International Conference on Learning Representations, 2018.
[3] Alex Toth and CT Kelley. Convergence analysis for Anderson acceleration. SIAM Journal on Numerical Analysis, 53(2):805–819, 2015.
[4] Vien Mai and Mikael Johansson. Anderson acceleration of proximal gradient methods. In International Conference on Machine Learning, pp. 6620–6629. PMLR, 2020.

---

### Decision · Program_Chairs · 2022-01-20

**Decision:**

Accept (Poster)

**Comment:**

The reviewers found this work well-motivated and the additional experiments conducted during the response phase were greatly appreciated. Anderson's acceleration appears to be a simple device that may be of great value to this field, and therefore this work is a timely contribution. The presented theoretical results justify the authors' modifications, although at times it felt more comparisons would be welcome: (a) Section 3.2 could have compared to a lot of three-term recurrences that lead to the optimal dependence on the condition number, including Chebyshev's polynomial and conjugate gradient, as well as the results in Brezinski et al. (2018); (b) Section 3.3 would benefit from some comparison with "Evans, Claire, Sara Pollock, Leo G. Rebholz, and Mengying Xiao (2020). “A Proof That Anderson Acceleration Improves the Convergence Rate in Linearly Converging Fixed-Point Methods (But Not in Those Converging Quadratically)”. SIAM Journal on Numerical Analysis, vol. 58, no. 1, pp. 788–810." (c) the results in Section 3.4 seem to be a bit preliminary and it would be great if the authors could compare to standard rates of SGD.

Overall we believe this work will generate more interest on memory-based optimization techniques in deep learning, and we encourage the authors to thoroughly polish their draft by incorporating the reviewers' comments and the responses during the discussion phase.